# Demethylation C–C coupling reaction facilitated by the repulsive Coulomb force between two cations

Xiaoping Zhang[1,7], Keke Huang[2,7], Yanlin Fu [3,7], Ni Zhang[4], Xianglei Kong [5], Yuanyuan Cheng[1], Mingyu Zheng[1], Yihao Cheng[1], Tenggao Zhu[4], Bina Fu [3,6] ✉, Shouhua Feng [2] ✉ & Huanwen Chen [1,4] ✉

Carbon chain elongation (CCE) is normally carried out using either chemical catalysts or bioenzymes. Herein we demonstrate a catalyst-free approach to promote demethylation C–C coupling reactions for advanced CCE constructed with functional groups under ambient conditions. Accelerated by the electric field, two organic cations containing a methyl group (e.g., ketones, acids, and aldehydes) approach each other with such proximity that the energy of the repulsive Coulomb interaction between these two cations exceeds the bond energy of the methyl group. This results in the elimination of a methyl cation and the coupling of the residual carbonyl carbon groups. As confirmed by high-resolution mass spectrometry and isotope-labeling experiments, the C–C coupling reactions (yields up to 76.5%) were commonly observed in the gas phase or liquid phase, for which the mechanism was further studied using molecular dynamics simulations and stationary-point calculations, revealing deep insights and perspectives of chemistry.

The formation of carbon–carbon bond (C–C), specifically carbon chain elongation (CCE) along with the construction of advanced functional groups, is fundamental to organic and/or biological synthesis utilizing easily accessible carbon resources[1,2]. Biologically, CCE processes are efficiently conducted through enzyme-driven pathways in living organisms[3]. Chemically, classical chemical reactions such as Friedel-Crafts alkylation reaction[4], Wurtz reaction[5], Aldol reaction[6], and Michael addition reaction[7] contribute to the formation of advanced molecular frameworks. Typically, the addition of one or two carbon atoms to a specific site can be readily accomplished using a diverse range of reagents, including $CO_2$, CO, HCN, NaCN, $CH_2I_2$, $CHCl_3$, $CH_2Cl_2$, $CH_3OH$, $(CH_3)_2LiCu$, HCHO, $CH_3MgI$, $Ph_3P = CH_2$, $(CH_3)_2SO_4$, CuCN, $CO(NH_2)_2$, and semicarbazide. The cross-coupling reactions developed by Heck, et al.[8,9], typically offer a versatile platform for C–C

bond formation. These reactions are characterized by a sophisticated process involving metal-catalyzed oxidative addition-reductive elimination[10,11] and/or the radical addition-elimination process[12,13]. While advanced CCE reactions have expanded the boundaries of synthesis, they often necessitate chemical catalysts with harsh steps or biological enzymes with complex biological substrates, posing challenges for active group protection, the resultant harvest and the subsequent yield.

Previously, Coote et al.[14], documented that the Diels-Alder reaction experienced acceleration in the presence of an electric field, as the flow of electrons from the highest occupied molecular of dienes to the lowest unoccupied molecular orbital of dienes was influenced by the electrostatic field. More recently, chemical reactions accelerated up to 6 orders of magnitude[15–20] were rationalized by the ultrahigh electric

[1]Jiangxi Key Laboratory for Mass Spectrometry and Instrumentation, East China University of Technology, Nanchang 330013, P. R. China. [2]State Key Laboratory of Inorganic Synthesis and Preparative Chemistry, Jilin University, Changchun 130012, P. R. China. [3]State Key Laboratory of Molecular Reaction Dynamics, Center for Theoretical and Computational Chemistry, Dalian Institution of Chemical Physics, Chinese Academy of Sciences, Dalian 116023, P. R. China. [4]School of Pharmacy, Jiangxi University of Chinese Medicine, Nanchang 330004, P. R. China. [5]State Key Laboratory of Elemento-organic Chemistry, College of Chemistry, Nankai University, Tianjin 300071, P. R. China. [6]Hefei National Laboratory, Hefei 230088, China. [7]These authors contributed equally: Xiaoping Zhang, Keke Huang, Yanlin Fu. ✉e-mail: bina@dicp.ac.cn; shfeng@jlu.edu.cn; chw8868@gmail.com

fields (approximately $10^9\,V\,m^{-1}$[21,22], showing that the electric field enables unusual chemical reactions[15-20] and the formation of polypeptides[23] instead of ionizing the neutral substrates. However, few efforts were made to investigate whether the repulsive effect of positive charges could be used to induce the CCE process.

In this work, an original strategy was presented using Coulombic repulsion between two positive charges to induce CCE reactions (Fig. 1) that can occur between two positively charged organic species (i.e., two sodiated acetone, R1a, R1b) in either the liquid or gas phases. Initially, the distance ($d_0$) between the positive charges sitting on the carbonyl carbon atoms of the cations R1a, R1b was far larger than $d_1$ (Fig. 1a), the repel effect induced by repulsive Coulomb force (RCF) between the R1a, R1b cations was too small (-0 eV) to trigger effective interaction between R1a and R1b. When $d_0$ gradually decreased, the RCF between R1a and R1b continuously increased and became a dominant force. When $d_0 = d_1$, the R1a and R1b cations began to contact on the edge of the reaction cross-section. Because of inertia, R1a ion still moved towards R1b, making $d_0$ continue to decrease. Once $d_0 < d_1$ (Fig. 1b), the repulsive energy generated by RCF might be significantly larger than the C–C bond energy, which induced the elimination of a small cleavage cation (typically methyl, amino, phenyl cation) and the coupling of the two carbonyl carbon atoms, resulting in positively charged CCE products. After electric neutralization, the CCE products were accumulated for further characterization. Thus, the RCF between two positively charged (i.e., protonated or metal ion complexed) organic species R1a and R1b functioned as a facile type of virtual catalyst to facilitate CCE and functional group construction (Fig. 1c).

## Results and discussion
### CCE reaction in liquid-phase

A custom-designed reaction apparatus (Fig. 2a) was comprised of a capillary spray array for electrospraying a 1:1 ($v/v$) sodiated acetone

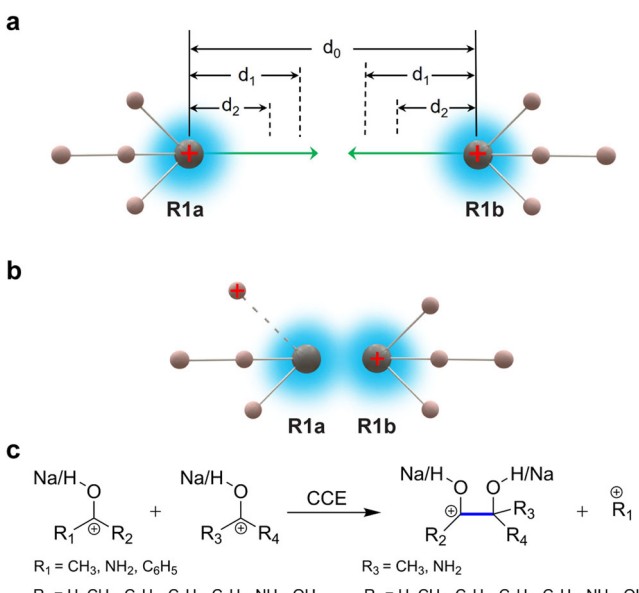

**Fig. 1 | Summary of C–C coupling reactions involving two cations R1a, R1b for carbon chain elongation (CCE). a** The schematic illustration on the initial stage ($d_0 \gg 2d_1$) of two positively charged acetone cations, R1a, R1b, approaching each other. **b** The repelling and coupling stage ($d_2 < d_0 < d_1$) of two positively charged acetone cations, R1a, R1b, approaching to each other. **c** The investigated C–C coupling reactions between two positively charged molecules, R1a, R1b. Note: R1a, R1b: two sodiated acetone. $d_0$ is the distance between the two positive charges sitting on the carbonyl carbon atoms of R1a and R1b. $d_1$ is the effective radius of the reaction cross-section between R1a and R1b. $d_2$ is the average length of C–C bond.

solution to produce reagent ions (e.g., R1a), a reaction vessel assembly and several electrodes to construct a precisely sequence-controlled voltage system (Fig. 2b, further described in the Methods section). More specifically, several electrodes contained an Al-plate, a coated Si-plate, a Cu-plate, a ground electrode. The Al-plate and the spray array were both applied with a constant voltage (+6 kV over the time slot of $t_0 \sim t_4$) for the acceleration of R1a. The ground electrode was applied with a constant voltage (0 V over the time slot of $t_0 \sim t_4$). The coated Si-plate was applied with a pulsed voltage (+2 kV over the time slot of $t_0 \sim t_1$, $t_2 \sim t_3$; 0 V over the time slot of $t_1 \sim t_2$, $t_3 \sim t_4$) for creation of positively charged ions of acetone (e.g., R1b) on the liquid-air interface. The Cu-plate was applied with a pulsed voltage (0 V over the time slot of $t_1 \sim t_2$, $t_3 \sim t_4$; disconnection over the time slot of $t_0 \sim t_1$, $t_2 \sim t_3$) for eliminating the extra electrostatic charges which might be accumulated up to $10^9\,V\,m^{-1}$[24] during the reaction duty cycles (Supplementary Note 1). This custom-designed reaction apparatus was ultimately applied for CCE reactions (Fig. 1) between the ions of R1a and R1b. While the whole apparatus was electronically grounded, the R1a ions (shown as the charged red-brown plume) produced by positive electrospray ionization (ESI, +6 kV) using the spray array were accelerated by +6 kV applied on the Al-plate to impact the R1b ions (shown as the small gray balls marked with + sign), which were created on the solution-air interface of acetone solution containing acetic acid ($10\,\mu g\,mL^{-1}$) and equivalent sodium salts of fluorine, chlorine ($10\,\mu g\,mL^{-1}$) by using the coated Si-plate with +2 kV immersed in the acetone solution (shown as the green inside the vessel).

It is noted that the primary products of CCE reactions between the ions of R1a, and R1b (Fig. 1) consist of charged species, specifically the methyl cation (i.e., $CH_3^+$) and the organic cation with elongated carbon chain (i.e., the sodium 2-methyl-butane-2,3-diolate-3-ium). Under the experimental conditions, the cations were electronically neutralized in two possible pathways (Fig. 3) to form stable final products. The pathway 1 was conducted by capturing a hydrogen radical from the solvent to yield a radical cation species, which was further directly grounded and neutralized to produce stable MH and $R_1H$ (i.e., $M = R_2C(OH/Na)\text{-}C(OH/Na)R_4R_3$; $R_1$ = methyl). The pathway 2 was conducted by cations coupling with encountered anions to produce stable MX and $R_1X$ (i.e., $M = R_2C(OH/Na)\text{-}C(OH/Na)R_4R_3$; $R_1$ = methyl; X = F, Cl, Br, $CH_3COO$). For example, by subjecting to ESI-MS, the final products of the CCE reaction produced signals at $m/z$ 149, $m/z$ 167 and $m/z$ 183 (Fig. 2c) corresponded to $(MH + H)^+$, $(MF + H)^+$, and $(MCl+H)^+$, respectively. These ion signals were further confirmed by isotope labeling experiments using deuterated acetone, yielding dominant signals at $m/z$ 158, $m/z$ 176 and $m/z$ 192 (Fig. 2d) corresponded to $(MH\text{-}d_9 + H)^+$, $(MF\text{-}d_9 + H)^+$, $(MCl\text{-}d_9 + H)^+$, respectively. This correspondence was attributed to the addition of H radical (pathway 1 in Fig. 3), $F^-$, and $Cl^-$ anions (pathway 2 in Fig. 3) in the solution within the reaction vessel. The ESI-MS (Fig. 2c, d) and collision-induced dissociation mass spectrometry (CID-MS) (Supplementary Fig. 1a, b) results indicated the generation of demethylation C–C coupling products during the CCE reaction between the ions of R1a and R1b.

The small $CH_3^+$ cation, which was produced during CCE process, was effectively captured by such anions as $F^-$, $Cl^-$, $Br^-$ and $CH_3COO^-$ to yield $CH_3F$, $CH_3Cl$, $CH_3Br$, and $CH_3COOCH_3$, respectively, in the reaction solution (pathway 2 in Fig. 3). The product signals at $m/z$ 93, $m/z$ 109, $m/z$ 153 and $m/z$ 133, which resulted from the formation of $[CH_3X + Ac^-]$ (X = F, Cl, Br, and $CH_3COO$) adducts, were also identifiable in the negative ion detection mode (Supplementary Fig. 1c). These findings were subsequently validated through CID-MS analysis (Supplementary Fig. 1d−g).

Under the optimized conditions (Supplementary Fig. 2), the CCE strategy demonstrated efficacy across a diverse range of chemical substrates (Supplementary Table 1) such as acetone, butanone, 2-pentanone, 2-hexanone, acetamide, acetic acid, acetophenone, acetaldehyde, and urea. The final products

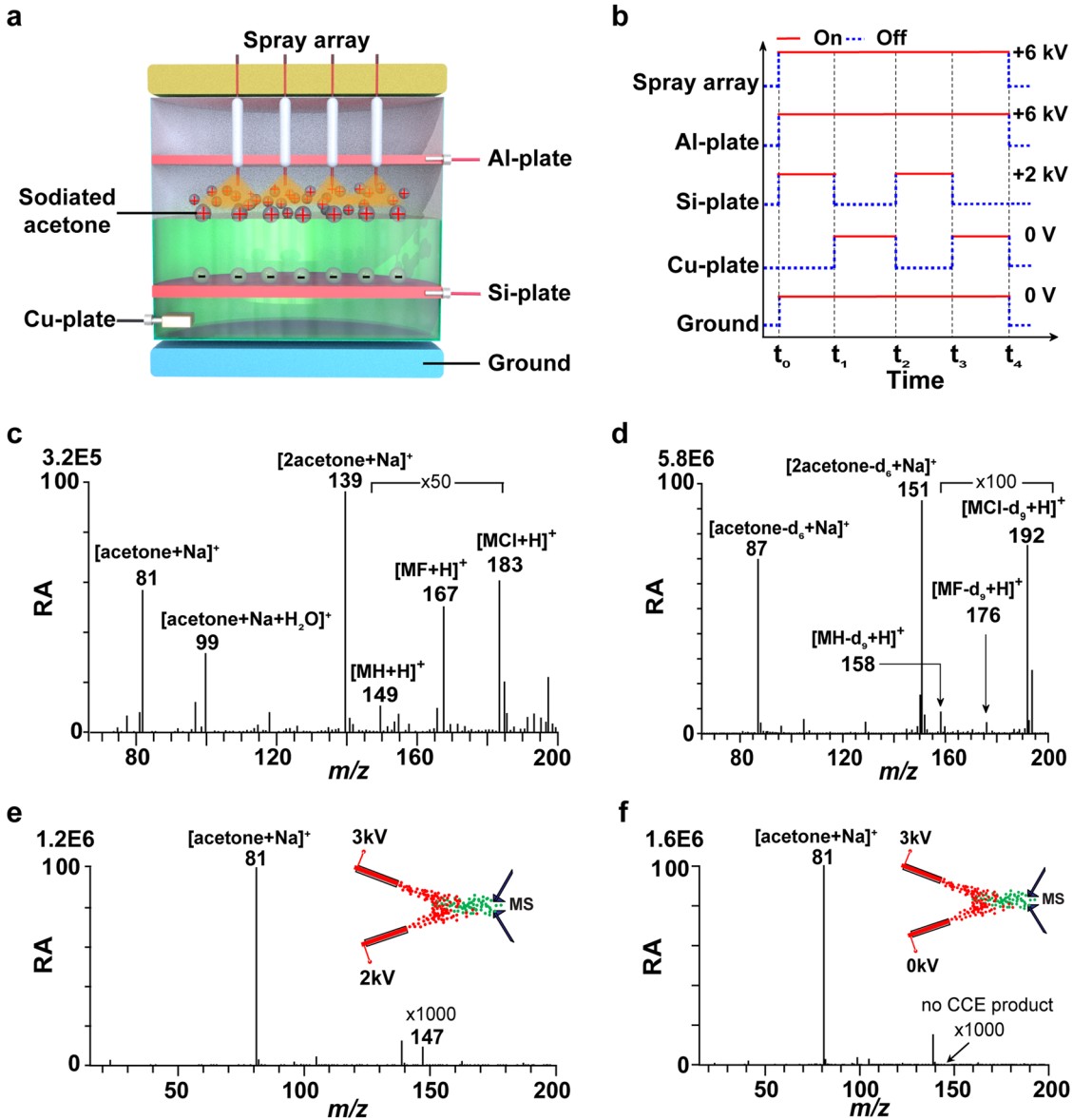

**Fig. 2 | The liquid-phase carbon chain elongation (CCE) reaction occurred between two sodiated acetone cations R1a, R1b. a** Schematic illustration of the experimental apparatus. **b** Diagram of potential variation of each electrode over time. **c** The abundant signals of CCE products recorded by electrospray ionization mass spectrometry (ESI-MS). **d** The isotope-labeled CCE products detected by ESI-MS. **e** The online-MS spectrum of CCE reaction between two sodiated acetone cations using a two-channel electrospray apparatus when spray 1 and spray 2 were both charged. **f** The online-MS spectrum of CCE reaction between two sodiated acetone cations using a two-channel electrospray apparatus when spray 1 was charged while the spray 2 was not charged. RA denotes Relative Abundance. **M** = R₂C(OH/Na)-C(OH/Na)R₄R₃. R₁, R₂, and R₃ all denote -CH₃ group.

enumerated in Supplementary Table 1 were confirmed through CID-MS measurements (Supplementary Fig. 3), and if applicable, deuterium labeling experiments (illustrated in Supplementary Fig. 4) were conducted using isotope-labeled reagents. No significant advantage was observed in facilitating the demethylation C−C coupling reaction for the tested compounds, whether they were sodiated or protonated forms (Supplementary Fig. 5). Similarly, the deamination C−C coupling and dephenylation C−C coupling processes resulted in the formation of products, as evidenced by Supplementary Fig. 6. These products included protonated/ sodiated urea and protonated/sodiated benzaldehyde, respectively. These findings necessitate a deep understanding of the chemistry underlying the reactions to be uncovered. The yields of CCE products in the liquid phase ranging from <1% to approximately 76.5%, were calculated based on the ratio of the intensity between the product and the sum of the intensities of the reactant,

and product (Supplementary Note 2). While this approach is frequently employed to estimate reaction yields in MS-based microsynthetic studies[17,25,26] because of its simplicity and effectiveness in assessing microscale products. In addition to MS techniques, ultraviolet-visible (UV-Vis) and nuclear magnetic resonance (NMR) spectroscopy were also employed to further characterize CCE products. 2-Methylbutane-2,3-diol standard and the CCE product formed by two protonated acetone cations exhibited similar UV-Vis spectra. Both of them resulted in the decrease or even disappearance of the characteristic absorption peak of $KMnO_4$. The characteristic spectra of $KMnO_4$ were in the range of 500 - 600 nm, and the maximum absorption peak is 525 nm (Supplementary Fig. 7). The characteristic peaks observed in ¹H NMR spectrum of CCE product closely corresponded to the peaks in ¹H NMR spectrum of the 2-methylbutane-2,3-diol standard (Supplementary Fig. 8). The characterization results suggested that CCE products

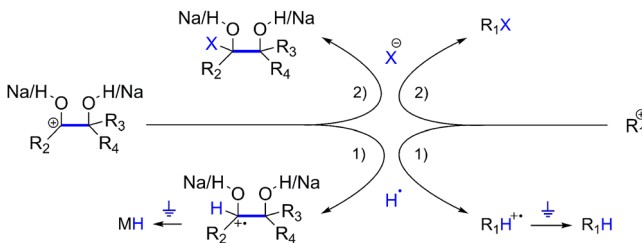

**Fig. 3 | Direct grounding and anion coupling in solvent to obtain the final product of carbon chain elongation reactions as depicted in Fig. 1.** M = R$_2$C(OH/Na)-C(OH/Na)R$_4$R$_3$.

can be produced in the liquid phase by employing the apparatus depicted in Fig. 2a.

As demonstrated in the experiment (Supplementary Table 2), CCE reactions only occurred between two positively charged species. For instance, the control experiments demonstrated that no CCE products (Supplementary Fig. 9a and d) were detected when only the electrospray was charged but no voltage was applied to the coated Si-plate. However, when both the spray and coated Si-plate were properly charged, the corresponding CCE products (m/z 149 and m/z 171 for sodiated acetone; m/z 105 for protonated acetone) were observed (Supplementary Fig. 9b and e). It was noted that all the CCE products were also validated through the corresponding CID-MS experiments (Supplementary Fig. 9c and f).

Similarly, ionic CCE products at m/z 147 can also be observed with a low intensity by using online MS with two-channel charged sprays (spray1, +3 kV, spray2, +2 kV) (Fig. 2e). The characterization of m/z 147 was confirmed by tandem MS experiment (Supplementary Fig. 10). No ionic CCE product ion was observed by using online MS with two-channel sprays, when only one-channel spray was charged (spray1, +3 kV, spray2, 0 kV) (Fig. 2f). These experiment results were similar with those obtained using the apparatus in Fig. 2a (Supplementary Fig. 9). Two-channel electrospray is in effect an easy apparatus to implement. However, on the one hand, the cation-cation collision in the two-channel electrospray is not a forward collision but an oblique collision at a certain angle. Due to the dispersion of three-dimensional space and the influence of atomizing gas flow, the forward collision probability is low, and the corresponding yield of CCE product ions is also low (~ 0.01% for sodiated/protonated acetone). The use of spray array apparatus can better improve the yield of CCE product ions (~ 14% for sodiated/protonated acetone). On the other hand, for the two-channel electrospray apparatus, the lack of product collection apparatus makes it difficult to collect the forming CCE product for subsequent analysis. While the use of spray array apparatus can facilitate the collection of products for subsequent characterization analysis. The design of a spray array-planar interface cell aims to promote the development of mass spectrometry preparation towards the bulk-scale synthesis. In addition, the low yield of the product, the difficulty of product collection, and the waste of the substrates result in the low utilization rate of the substrates. Therefore, a spray array-planar interface cell apparatus was designed to study the CCE reaction between two cations. In this case where two different cations were presented in the spray and the reaction vessel, the hetero-dicationic reaction products could be observed (Supplementary Fig. 11). The CCE products, corresponding fragment ions, ions associated with electrochemical redox reactions, and C−C bond cleavage reactions were not detected during the high-voltage spraying of sodiated acetone solution or acetone aqueous solution (Supplementary Fig. 12).

## CCE reaction in gas-phase
Gas-phase chemistry, due to the absence of a solvent, allows for the observation of the intrinsic characteristics and fundamental reactivity

of chemical species[27–30]. Consequently, the mass-selected sodiated acetone (R1, referring to gas-phase R1a, R1b ions of m/z 81) was confined in an ion trap (with a vacuum of approximately 10$^{-5}$ Torr) to replicate RCF-induced CCE reaction in the gas phase (Supplementary Fig. 13). This process could be readily optimized by modifying collision energy (CE), activation time, and presence of neutral chemicals, offering feasible approach to investigate the chemical reactivity of the precursor ions and ionic products in the gas phase[31–34]. Under the specified experimental parameters, the sodiated acetone R1 (m/z 81) did not yield any fragments or adducts in the CID-MS spectrum (Supplementary Fig. 14) with low CE (CE < 1.6 eV). However, it did generate abundant product ions (Fig. 4a, b, Supplementary Fig. 14) at m/z 41, m/z 99, and m/z 147 when exposed to relatively high CE (CE ≥ 3.6 eV). The prominent peak observed at m/z 41 (Fig. 4b) corresponded to the dissociation products of R1 (m/z 81), indicating a combination of the allyl cation (P1) and hydrated sodium cation (P2) through the elimination of NaOH and propadiene from the dissociation of R1, as illustrated in Fig. 4a, Supplementary Figs. 15 and 16a. The proposed structures of the allyl cation and hydrated sodium cation were validated through isotope labeling experiments (Fig. 4d, Supplementary Fig. 16b), a kinetic isotope effect of $k_H/k_D = 5$, and theoretical calculations (detailed in the subsequent calculation section). The dominance of the allyl cation (P1) and the hydrated sodium cation (P2) may be attributed to the stability of the product ions, which are more stable than P4 carbocation. The detailed information is illustrated in the following calculation results section.

Interestingly, the anticipated product P4 resulting from CCE reaction was observed at m/z 147, exhibiting an exact mass-to-charge ratio of m/z 147.03918. This value closely corresponded to the theoretical mass of $C_5H_9Na_2O_2^+$ (m/z 147.03925, Supplementary Fig. 17) with a deviation of 0.5 ppm. Upon CID, the precursor ion at m/z 147 (P4) generated distinctive fragments at m/z 146 (P5), m/z 131 (P6), m/z 107 (P7/P7'), m/z 91 (P8), and m/z 67 (P9) (Fig. 4c) through the loss of a H radical, CH$_4$, NaOH, (NaOH+CH$_4$), and two molecules of NaOH, respectively. The data from the fragmentation (Supplementary Fig. 18a) confirmed that P4 was a covalently bound product formed in the gas phase. As a result, P4 (m/z 147) was determined to be the (sodium 2-methyl-butane-2,3-diolate)−3-ium with a relatively high yield (≥ 25%, Supplementary Note 2). P4 was the CCE product observed as a charged intermediate (Supplementary Table 3) prior to electrical neutralization. The effects of experimental conditions including CE, CID reaction time, and water vapor content within the ion trap on the generation of P4 products were systematically investigated (Supplementary Fig. 19). The data on the other product ion CH$_3^+$ (m/z 15) of CCE reaction was not recorded, because the default setting of the mass spectrometer commenced recording from m/z 20, thereby excluding the lower mass range.

Isotope-labeling experiments also confirmed gas-phase CCE reactions. For instance, the tandem mass spectrum of sodiated acetone-d$_6$ (m/z 87) resulted in a mass shift of 9 Da for P4-d$_9$, as evidenced by the increase from m/z 147 to m/z 156 following the demethylation C−C coupling reaction. This confirmed the presence of three -CD$_3$ groups in P4-d$_9$ (Fig. 4d). Under the experimental conditions, P4-d$_9$ (m/z 156) exhibited characteristic fragments at m/z 154 (P5-d$_8$), m/z 136 (P6-d$_5$), m/z 115 (P7-d$_8$), m/z 95 (P8-d$_4$), and m/z 74 (P9-d$_7$) due to the elimination of D radical, CD$_4$, NaOD, (NaOD+CD$_4$), and 2NaOD, as illustrated in Fig. 4e. As a result of the extended activation period (≥100 ms) within the ion trap, D radical may combine with the precursor ions, leading to the formation of the peak at m/z 158 (Fig. 4e). The pathways for the fragmentation of P4-d$_9$ (m/z 156) were determined in Supplementary Fig. 18b, demonstrating the structural identity of P4 and P4-d$_9$, as confirmed by comprehensive spectral data obtained from CID-MS experiments (Fig. 4e).

Furthermore, the gas-phase reactions of mass-selected protonated urea, protonated 2-pentanone, protonated 2-hexanone, and

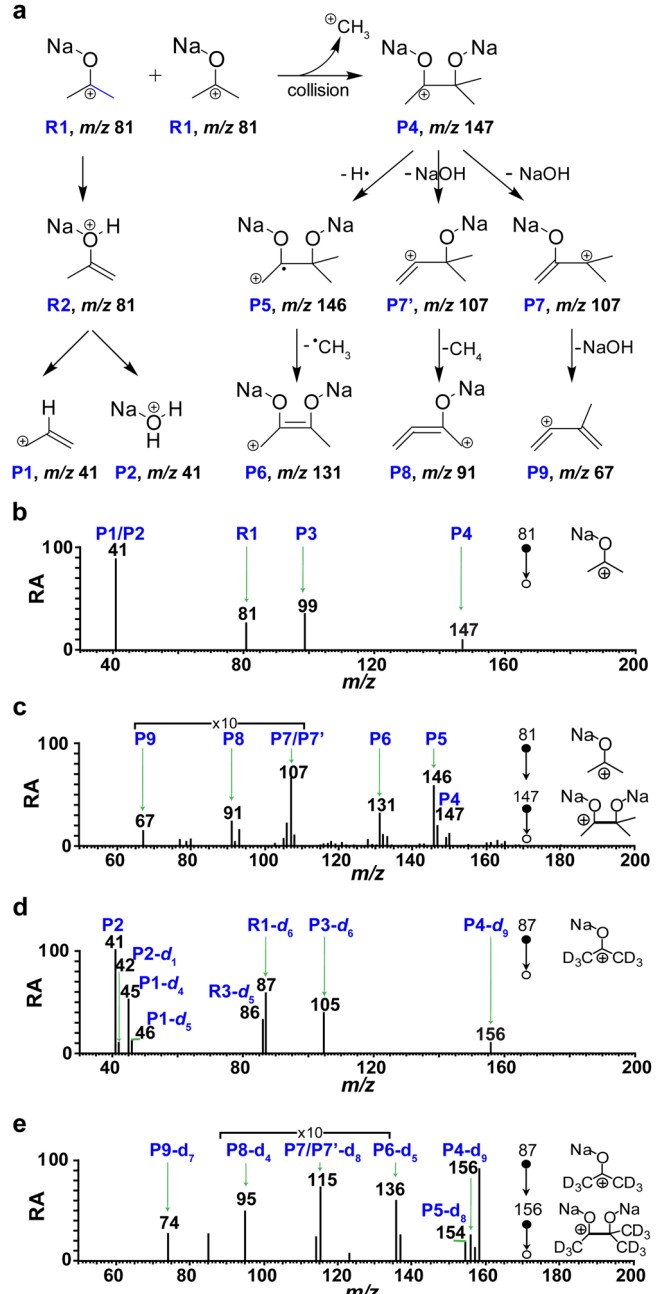

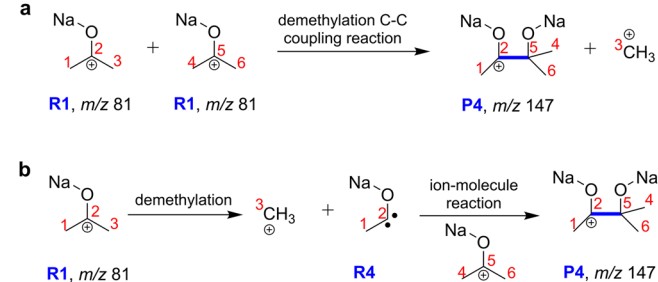

**Fig. 5 | Gas-phase production of sodium 2-methyl-butane-2,3-diolate)−3-ium (P4, *m/z* 147) by collision activation of sodiated acetone (R1, *m/z* 81). a** Route 1 for demethylation C−C coupling reaction through a direct mechanism. **b** Route 2 for ion/molecule reaction by a two-step mechanism.

**Fig. 4 | Reaction and dissociation of sodiated acetone (R1, *m/z* 81) and sodiated acetone-d₆ (R1-d₆, *m/z* 87) in the gas phase. a** The proposed structures for product ions in the dissociation R1. **b** The product ions in the tandem mass spectrum of R1. **c** Tandem mass spectrum of R1 (*m/z* 81 →*m/z* 147 →). **d** The product ions in the tandem mass spectrum of R1-d₆. **e** Tandem mass spectrum of R1-d₆ (*m/z* 87 →*m/z* 156 →). (Note: Signal-to-noise ratio is significantly greater than 3. RA denotes Relative abundance. R1 refers to gas-phase R1a, R1b ions of *m/z* 81).

protonated acetic acid with neutral acetone vapor (approximately $10^{-5}$ Torr) in the ion trap were also conducted. The experimental data confirm that the demethylation C−C coupling reaction occurs exclusively between two cations (Supplementary Fig. 20), but not between cations and neutral compounds (Supplementary Figs. 21 and 22). This observation aligns with the data presented in the liquid phase (Supplementary Table 2, Supplementary Figs. 9 and 10). The gas-phase experimental data strongly indicate that the formation of P4 (*m/z* 147) occurs through the demethylation C−C coupling reaction between two positively charged acetones R1 (*m/z* 81) under specific experimental conditions.

## Gas-phase CCE reaction studied by stationary-point calculations

A further in-depth theoretical examination of the dissociation or coupling of R1 (referring to gas-phase R1a, R1b ions of *m/z* 81) has been conducted through density functional theory (DFT) calculations. As depicted in Fig. 5, there are two potential pathways for the formation of the P4 product (*m/z* 147) involving two sodiated acetone cations. Route 1 involves CCE reaction through the elimination of a methyl cation with a direct mechanism (Fig. 5a). Route 2 involves the ion/molecule reaction through a two-step mechanism, in which a carbene intermediate is initially formed from a sodiated acetone by the direct loss of a methyl cation, and then reacts with another sodiated acetone (Fig. 5b). The stationary point calculations conducted at B3LYP-gCP-D3/6-31 + G(d) level indicate that the product ion at *m/z* 147 through Route 1 (Fig. 6a, Supplementary Fig. 23) is more favorable compared to Route 2 (Supplementary Fig. 24).

In addition to route 1 through TS1 and TS2, an alternative pathway for C−C coupling through TS3 (Supplementary Fig. 23) involving multiple transition states and intermediates was also identified. This pathway involves the initial coupling of the C−C bond, followed by the rotation of the CH₃ group, and ultimately the detachment of CH₃⁺. The energy barrier for TS3 is approximately 7.1 eV. For Route 1, the key transition states of TS2 for P4a formation and TS3 for P4b formation were verified through intrinsic reaction coordinate (IRC) calculations (Supplementary Fig. 25), thereby validating the findings presented in Fig. 6a and Supplementary Fig. 23, respectively. The imaginary vibrational frequency of TS1 and TS2, along with the corresponding eigenvector of these vibrational modes, indicated the correct direction associated with the stretching of one C−C bond and the contraction of the other C−C bond, respectively (as shown in Supplementary Information: The imaginary frequency mode of TS1 and TS2). Our IRC calculations of TS1 and TS2 demonstrate that the correct direction is associated with one C−C bond stretching and the other C−C bond contracting. (Supplementary Fig. 25a).

For Route 2, the energy required for the initial dissociation of a single sodiated acetone and the subsequent ion-molecule reaction between carbene and sodiated acetone is at least 8.9 eV (Supplementary Fig. 24), significantly exceeding that for Route 1. Additionally, the dissociation energies of other compounds with varying R₁ groups were calculated, revealing similarly high energy requirements ranging from 8.9 eV to 14 eV (Supplementary Fig. 26). Compared to other R₁ groups, the methyl group requires the lowest energy. The findings indicate that the initial dissociation of R1 through Route 2, followed by the ion-molecule reaction, is less likely to occur compared to Route 1, which involves a direct demethylation C−C coupling reaction. Consequently, the reaction proceeding through a direct mechanism is a more favorable process in terms of both kinetics and energy. The charge analysis results (Supplementary Fig. 27) obtained with the natural bond orbital method suggest that there exists charge transfer from the carbonyl carbon to the methyl carbon, possibly attributed to the significant RCF

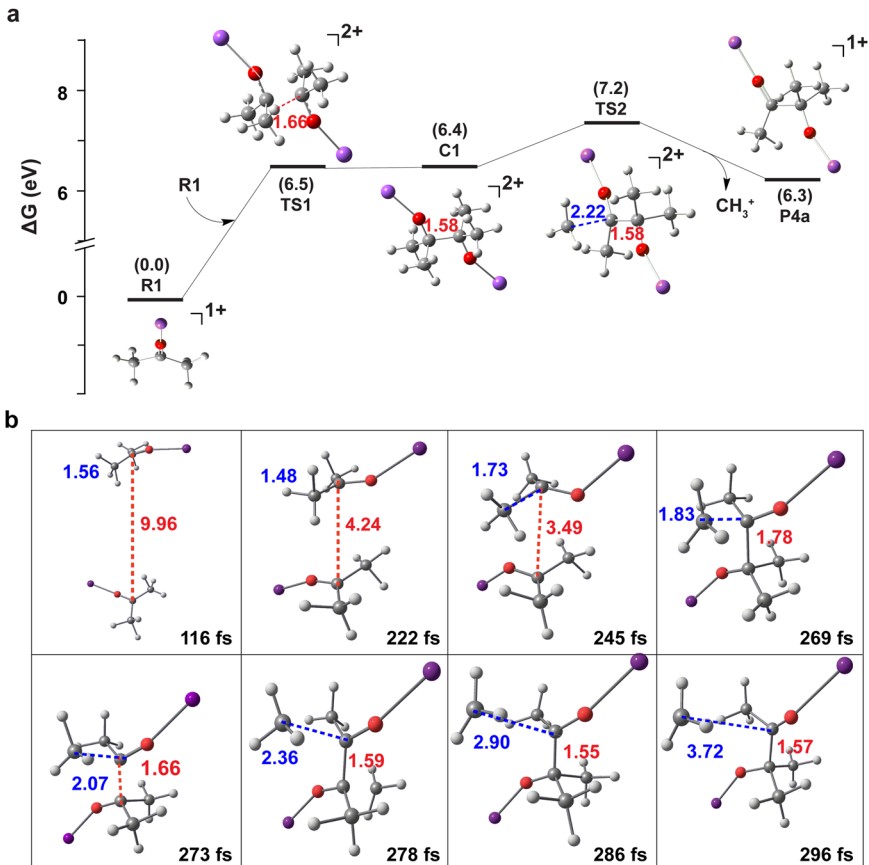

**Fig. 6 | Results of stationary point and ab initio molecular dynamics calculations that support the carbon chain elongation process. a** Relative energy levels for Route 1 for the generation of P4a ion via TS1 and TS2 through the demethylation C−C coupling reaction. **b** Snapshots of one reactive trajectory via TS1 and TS2.

(Note: Each panel depicts the geometry at the specified reaction time (fs). The internuclear distances (Å) between the two C atoms involved in bond formation, and between the two C atoms in detaching $CH_3^+$ and P4a are indicated by red and blue fonts, respectively).

between the two cations. Furthermore, the sequence of energies needed to dissociate R1 (Supplementary Fig. 28a) and P4a (Supplementary Fig. 28b) corresponded with the experimental measurements (Fig. 4b, c). The detailed discussion of relative energies, geometric parameters, and charge variations is presented in Supplementary Figs. 23–28. No carbene intermediate (Supplementary Fig. 29) was detected with greater certainty during CCE reactions conducted with the experimental apparatus (Fig. 2a), wherein the solution within the reaction vessel was supplemented with $CH_3CN$ to potentially react with any carbene intermediate[35]. The findings consistently demonstrate that CCE reaction is likely to occur via Route 1.

**Gas-phase CCE reaction studied by dynamics simulations calculations**

To gain a deeper understanding of the dynamical behavior of the reaction between the two cations, ab initio molecular dynamics (AIMD) calculations were conducted at the B3LYP/6-31 + G(d) level of theory. Extensive dynamic simulations relying on a full-dimensional potential energy surface are considered ideal; however, the creation of a global potential energy surface for a reactive system of this scale remains a formidable challenge[27,36–38]. Figure 6b displays snapshots of a typical trajectory obtained through TS2. The trajectory was initiated with a 16 Å separation between two cations, achieving a record time of 0 fs. The reaction through TS2 was observed to occur efficiently when R1 reactants were initially rovibrationally excited, especially with the excitation of the $CH_3$ group. The evidence suggests that the reaction proceeds through the formation of the C−C bond and the elimination of $CH_3^+$ via TS1 and TS2, respectively. The trajectory animation generated by TS2 during the formation of P4a indicates that the $CH_3$ group

in R1 is vibrationally excited, as depicted in the animation of the Supplementary Movie 1.

To achieve more understanding of the repulsive effect on the reaction caused by the RCF, stationary-point and AIMD calculations were conducted on the potential energy. As illustrated in Fig. 7a, a standard one-dimensional (1D) potential energy curve depicts the relationship with $d_0$ (the distance between the two positive charges on the carbonyl carbon atoms of R1a, R1b cations) while keeping other degrees of freedom constant at the equilibrium geometry. The curve was calculated at the B3LYP level. The potential energy at $d_0 = 2.4$ Å exceeds the energy of the C−C bond (approximately 3.4 eV). Specifically, the potential energy experiences a significant increase when $d_0$ is less than 2.4 Å. When $d_0$ is equal to $d_1$ (where $d_1$ presents the effective radius of the reaction cross-section of R1a, R1b cations, 1.8 Å) and $d_0$ is equal to $d_2$ (where $d_2$ presents C−C bond length, 1.6 Å), the potential energies are approximately 5.6 eV and 6.7 eV, respectively. The findings are additionally corroborated by AIMD calculations.

Figure 7b illustrates the interatomic distances and potential energy over the course of the reaction time for the trajectory depicted in Fig. 6b. The reaction involves the formation and breaking of C−C bonds at a distance of 1.8 Å (the intersection of the blue and green lines in Fig. 7b at 269 fs), and the potential energy at this distance is approximately 8.2 eV, exceeding that of 1D static potential energy curve (approximately 5 eV). This phenomenon is comprehensible due to the rovibrational excitation of the two reactants in AIMD simulations, and the calculation of the 1D static potential curve with the reactants held at the equilibrium geometry (i.e., the ground rovibrational state). AIMD results are generally in agreement with the static potential curve, but they provide a more accurate depiction of the

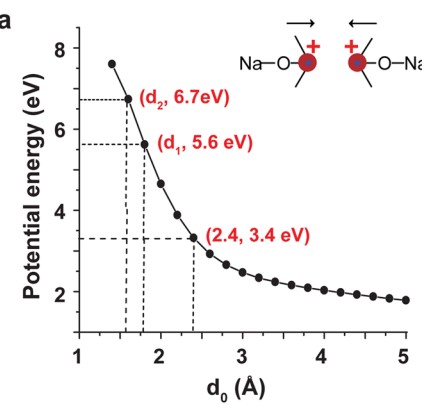

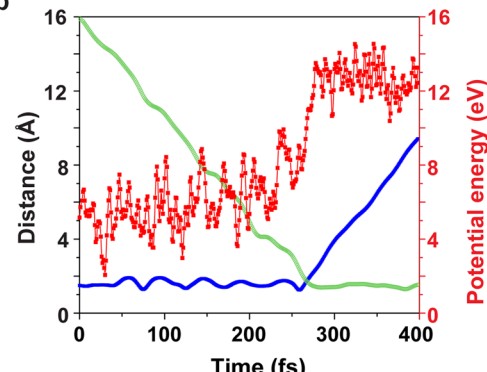

**Fig. 7 | Potential energy of demethylation C–C coupling reaction using stationary point and ab initio molecular dynamics calculations. a** Potential energy as a function of the distance ($d_0$) between the two positive charges on the carbonyl carbon atoms, with other degrees of freedom fixed at the equilibrium geometry, where $d_1$ is the effective radius of the reaction cross-section of the two R1 cations, and $d_2$ is C–C bond length. The potential energy at $d_0 = 2.4$ Å exceeds the energy of the C–C bond (-3.4 eV). **b** Interatomic distances and potential energy as a function of reaction time for one typical trajectory in Fig. 6b. (Note: Green curve represents the trend of the internuclear distance (Å) between the two carbonyl carbon atoms of sodiated acetone and sodiated acetone with bond forming. Blue curve represents the trend of the internuclear distance (Å) between the carbonyl carbon and methyl carbon in detaching $CH_3^+$ and P4. Red curve represents the reaction potential energy. The variation fluctuations in 0–200 fs and 300-400 fs are caused by vibrations of molecules, while the drastic changes in 200−300 fs are due to the occurrence of chemical reactions).

dynamic behavior of the reaction. As depicted in Fig. 7b, the inflection point for the interatomic distance between the two carbonyl carbon atoms of sodiated acetones essentially coincides with that of the interatomic distance between the carbonyl carbon and methyl carbon during the detachment of $CH_3^+$ and P4a. The findings indicate that the formation of the C–C bond and the detachment of $CH_3^+$ occur in the collision process of the two reactant cations.

The potential energies along this trajectory peak at approximately 8 eV, a level that is relatively high but justifiable, given that TS2 for this pathway is already at 7.2 eV. AIMD calculations were initiated for two reactants, incorporating additional rovibrational energies and CE. Consequently, the potential energies for this trajectory may exceed the transition-state energy. Thus, according to the results obtained from AIMD, it is anticipated that the substantial potential energy produced by RCF serves as the primary driving force in promoting the removal of a methyl cation and the linking of the two carbonyl carbon atoms (demethylation C–C coupling), leading to the formation of positively charged CCE products. AIMD findings indicate that the reaction R1 + R1 → P4 + $CH_3^+$ predominantly takes place when R1 is initially in a highly excited state and in its internal degrees of freedom, facilitated by the effective transfer of energy from translational motion to the internal excitations of the reactants. This aligns with the experimental procedure, where the total energy supplied by the electric field for the gas-phase collisional activation of R1 was not less than approximately 5 eV, with a CE of 22% in the ion trap. Consequently, CE (i.e., relative translational energy) for R1 counterparts remains high even when the distance is substantial. When the two reactants (cations) approach each other, RCF between the two sodium acetones R1 increases significantly (Fig. 7). This process initially cools the reactants translationally due to the high RCF, resulting in the rapid deceleration of the two cations and the transfer of substantial energy to the excitation of the $CH_3$ group. In this process, the formation of P4 from the collision of sodium acetones R1 is energetically enabled, as facilitated by electric field activation and RCF promotion.

### The universality of demethylation C–C coupling reaction

The proposed strategy demonstrated high efficiency in increasing the carbon atoms per step for a wide range of chemical substrates under both gas-phase (Supplementary Table 3) and liquid-phase conditions (Supplementary Table 1). This approach offers a straightforward method for obtaining halogenated hydrocarbon derivatives (e.g.,

fluorinated alcohol and ketone) and other functionalized products. $CH_3^+$ species generated in CCE reaction has prompted advanced research into the methylation/demethylation of DNA in the field of epigenetics. CCE reactions are particularly noteworthy for their ability to facilitate the direct introduction of multiple active groups, such as -F, -$CH_3COO$, and -$NH_2$, without the need for complex group protection prior to the reaction. As no chemical catalysts or biological enzymes were utilized throughout the entire process, the ionic species could be readily separated from the neutrals, thereby facilitating the product harvest. Additionally, the absence of polysubstituted products, such as *p,m,o*-dihydroxybenzene, can be attributed to the repulsion effect of positive charges, which hindered further substitution after the initial ionic group was attached to the molecular skeleton. The yields of CCE products in both the gas phase and liquid phase varied from <1% to approximately 76.5%. The disparity in yields between the gas phase and the liquid phase could be associated with the stability of CCE products. For instance, the demethylation C–C coupling products generated in the gas phase existed in cationic form, whereas those produced in the liquid phase were in neutral form due to the capture of H radicals or other anions in the solution.

According to Gomberg–Bachmann[39–41], in the pinacol synthesis, the coordinated ketyl radical generated by the reduction of carbonyl compounds with an Mg/MgI_2 mixture has the capability to undergo C–C bond coupling. C–C coupling mechanisms described in this study differed from those reported in Gomberg–Bachmann[39–41]. This is because the demethylation, deamination, and dephenylation C–C coupling reactions in this study were influenced by both the electric field acceleration and RCF between the two cations. Gas-phase CCE products resulting from demethylation, deamination, and dephenylation C–C coupling reactions were observed with protonated/sodiated forms of 2-pentanone, 2-hexanone, acetic acid, urea, and benzaldehyde (Supplementary Fig. 30). In this research endeavors, the formation of O–O coupling products were detected during the dissociation of sodiated alcohols, concomitant with the elimination of $NaH_2^+$ (Supplementary Fig. 31).

In summary, demethylation, deamination, and dephenylation C–C coupling reactions are facilitated by the strong RCF between two positively charged organic species of various compounds (e.g., ketones, aldehydes, alcohols, acetic acid, aniline, and phenol) in the liquid/gas phase, and have been developed under the influence of a precisely sequence-controlled voltage system. The proposed

mechanism was validated through precise mass measurements, CID-MS, deuterium labeling, NMR experiments, and theoretical calculations. As various compounds with different groups were used, the proposed strategy was demonstrated to minimize operational steps and chemical facilities for advanced molecule construction with active functional groups under mild, environment-friendly conditions. This approach requires neither transition/noble metal catalysts nor protection/deprotection steps. The yields (0.1 ~ 68% in the gas phase and 0.8 ~ 76.5% in the liquid phase) have been reasonably achieved by the concept-proof setup, and further improvements in yields could be realized through the use of specialized apparatus. Therefore, this study may not only potentially introduce possibilities for the direct assembly of products with CCE and sophisticated functional groups, but also have various applications across multiple fields, such as chemistry, natural product research, and drug discovery.

## Methods

### Materials and methods

Methanol of HPLC grade was purchased from Sigma-Aldrich (St. Louis, MO, USA) with a purity > 99.9%. Acetone, butanone, 2-pentanone, 2-hexanone, urea, acetamide, ethanol, propanol, and acetic acid were purchased from J&K Scientific Ltd. (Shanghai, China) with a purity > 99%. Acetophenone, acetaldehyde, benzaldehyde, aniline, and phenol were purchased from Sun Chemical Technology (Shanghai, China) Co, Ltd. with a purity > 99%. Acetone-$d_6$ was purchased from Cambridge Isotope Laboratories, Inc. (Andover, USA) with a purity > 99%. 2-Methylbutane-2,3-diol was purchased from Beijing Innochem Science &Technology Co, Ltd. with a purity > 95%.

### Experimental setup and process

The custom-designed reaction apparatus (Fig. 2a) is comprised of three primary components: (1) a capillary spray array for the generation of R1a ions by electrospraying sodiated acetone solution, (2) reaction vessel for CCE reaction in liquid-phase, (3) several electrodes including Al-plate, coated Si-plate, Cu-plate, in which a precisely sequence-controlled voltage system is constructed. As shown in Fig. 2a, R1a ions were produced by the spray array while the R1b ions were created on the liquid-air interface by the high voltage applied to the coated Si-plate covered by the solution inside the reaction vessel. Thus, the solution interface was the major region for liquid-phase CCE reaction. The ionic products of liquid phase CCE reaction can subsequently capture H radicals or other anions in the solution, resulting in the formation of stable final products of the CCE reaction.

Taking sodiated acetone as an example, the capillary electrospray array was comprised by inserting one end of all the capillaries (inner diameter (ID) 0.19 mm) into a solution of sodiated acetone, which was then biased with high voltage (+1 ~ 6 kV) to generate charged droplets containing R1a ions (Fig. 2a). All the spray tips were neatly stacked together, with a distance of 5 mm from the liquid surface. In order to create cations R1b, a coated Si-plate biased with +1 ~ 2 kV was placed under the solution inside the reaction vessel. A Cu-plate was immersed in the solution and linked to Ground to dissipate the static electricity accumulated during the CCE process. A timing-control system was developed to regulate the high voltage settings for various electrodes such as Al-plate, spray array, coated Si-plate, and Cu-plate (Fig. 2a).

In this study, the voltage values were equally set and maintained constant for the spray array and Al-plate electrodes, whereas the voltage in the coated Si-plate exhibited a pulsed mode. When the cations formed by the spray array collides with the cations formed at the solution surface of the Si-plate, a large amount of static electricity is generated due to the accumulation of charge, which needs to be released or eliminated. The static electricity generated during the reaction is released to the ground by adding a Cu-plate electrode to the solution. The Si-plate and Cu-plate are in the same solution, and their purpose is to form cations and release static electricity, respectively,

resulting in their inability to work at the same time. Therefore, Si-plate and Cu-plate work alternatively in a pulse mode, while the spray array, Al-plate, and Ground maintain a constant voltage.

Following the operation of the timing-control system, R1a cations from capillary produced by the capillary electrospray array reacted with R1b cations present at the liquid-air interface, which were generated from the reaction vessel through the sequence-controlled voltage system, resulting in the formation of CCE products. The formed CCE products underwent additional analysis using offline MS, UV-Vis and NMR techniques. The quantity of acetone reactant utilized was 24 mL, and the reaction took place over a duration of 30 min using the apparatus depicted in Fig. 2a. Subsequently, the reaction solution was diluted by a factor of 100 and subjected to ESI-MS analysis.

### MS for detection CCE products in liquid phase

All MS experiments related to CCE products formed in the liquid phase were performed by a linear ion trap (LTQ)-XL advantage ion-trap mass spectrometer equipped with a custom-made ESI interface (Thermo Fisher Scientific Inc., San Jose, USA). LTQ mass analyzer was utilized in the positive-ion detection mode, and mass spectra were acquired within the $m/z$ range of 15 ~ 400. The formation of $[M + H]^+$ or $[M + Na]^+$ ions was achieved through ESI under the specified typical conditions, which were optimized by adjusting various experimental parameters such as capillary temperature, and tube lens voltage. LTQ mass analyzer was additionally utilized in the negative-ion detection mode, and mass spectra were acquired within the $m/z$ range of 15 ~ 200. The formation of $(M + CH_3COO^-)$ ions was achieved by the classical ESI-MS method. The optimized conditions were established as follows: the spray voltage was maintained at 3 ~ 6 kV, nitrogen gas (purity > 99.999%) was supplied at a pressure of 0.4 MPa, the capillary temperature was set to 320 °C, the capillary voltage was adjusted to 30 V, the tube lens voltage was set at 20 V, and sample solutions were delivered into the ESI source via a syringe at a flow rate of 5 μL min⁻¹. The angle formed between the electrospray beam and the MS inlet of the LTQ-MS instrument was 150°. MS/MS conditions were as follows: the ion gauge reading in the instrument software was maintained at $1 \times 10^{-5}$ torr, CE was ranged from 20 ~ 30%, and the isolation width was set at 2 Da. The system automatically optimized the other LTQ-XL parameters.

### Gas phase reaction by using MS as a reactor

All experiments involving MS of CCE products formed in the gas phase were carried out by an LTQ-XL advantage ion-trap mass spectrometer equipped with a custom-made ESI interface (Thermo Fisher, Scientific Inc., San Jose, USA). Various compounds, including acetone, were dissolved in a methanol solution at typical concentrations ranging from 0.01 mM to 0.2 mM. The solution was supplemented with an appropriate quantity of sodium salt in order to generate sodium adduct ions. Subsequently, the sample solutions underwent ESI-MS analysis. LTQ mass analyzer was utilized in the positive-ion detection mode, and mass spectra were acquired within the $m/z$ range of 15 ~ 200 and 50 ~ 400, respectively. The formation of $[M + H]^+$ or $[M + Na]^+$ ions was achieved through ESI under the following standard conditions, which were optimized by adjusting various experiment parameters such as capillary temperature and tube lens voltage. ESI-MS conditions were set as follows: the spray voltage was maintained at 3 kV; nitrogen gas with a purity exceeding 99.999% was supplied at a pressure of 0.4 MPa; the capillary temperature was set to 320 °C; the capillary voltage was 30 V; the tube lens voltage was 20 V. Sample solutions were introduced into ESI source using a syringe pump at a flow rate of 5 μL min⁻¹. The angle formed between the electrospray beam and the MS inlet of the LTQ-MS instrument was 150°, and the pressure within the ion trap was $1 \times 10^{-5}$ torr.

Mass selection and collisional activation were carried out according to the standard isolation and excitation procedures by using

the advanced scan function of the LTQ software. CID-MS experiments involved applying an excitation alternating current voltage to the end caps of the ion trap to induce collisions of the isolated ions (with an isolation width of 1 - 10 $m/z$) for a duration of 30 - 200 ms and varying excitation amplitudes within the ion-trap. CID-MS spectra of the protonated molecule ions and sodiated complex ions were obtained by activating the precursor ions at normalized collision energies varied from 0% to 50%. The experiments were conducted under conditions of relative humidity of air ranging from 15% to 60%, and pure helium (99.999%) was used as the collision gas. Other LTQ-XL parameters were automatically optimized by the system.

### High-resolution MS experiments
High-resolution MS experiments were performed on an Orbitrap-XL mass spectrometer (Thermo Scientific, San Jose, USA) equipped with a home-made ESI setup. The instrument was operated at a high-resolution up to 200,000. A mass spectrum was averaged for 30 scans obtained over the sampling period of 0.5 min in the range of $m/z$ 15 to 200. Other Orbitrap-XL parameters were in accordance with the LTQ-XL parameters mentioned above.

### Characterization of CCE product by UV-Vis spectroscopy
CCE product and corresponding diols standard were analyzed by using UV-Vis absorption spectroscopy (UV-1900i, Shimadzu, Japan). Potassium permanganate was reported to undergo reduction by diols in an acidic environment, with the maximum absorption peak at 525 nm[42] (Supplementary Fig. 7). Following the reaction with diols, the peak at 525 nm exhibited a decrease or even disappearance.

### CE calculations
Equation below is derived from the conversion equation between CE and normalized CE as provided by Thermo manufacturers. In Equation, settling NCE represents the setting value of CE, isolation center represents the mass-to-charge ratio of the precursor ion, and charge factor represents the charge factor. If a single charge is carried, its value is 1. Given the mass to charge ratio of the precursor ions is $m/z$ 81 and NCE setting value is 22, then the calculated energy in electron volt unit is 22 × 81 / 500 × 1 = 3.6 eV.

$$E_{col} = (\text{settling NCE}) \times (\text{isolation center}) \times (\text{charge factor}) \div (500 m/z)$$

### DFT calculations
The demethylation C–C coupling reactions resulting from the collision of two sodiated acetone cations were investigated through theoretical calculations by using the Gaussian 09 program[43]. DFT calculations are used to examine the electronic structure of molecules in their ground state by analyzing electron density. These calculations require less computational overhead compared to formal ab initio methods, while still incorporating dynamic correlation.

B3LYP-gcp-D3 functional, known for its accurate calculation of reaction barriers, has been successfully utilized in several analogous systems[44–46], and was therefore selected in this study. The structures of the reactants, transition states, intermediates, and dissociation products were optimized with DFT method at B3LYP-gcp-D3/6−31 + G(d) level. All reactants, intermediates, and dissociation products were confirmed as true energy minima due to the absence of imaginary frequencies. All transition states, based on the presence of a single imaginary vibrational frequency and a plausible vibrational mode, were identified. Furthermore, transition states were verified to establish a connection between reactant and product ions through the employment of IRC calculations. The vibrational frequencies of all the principal species were computed with the same theoretical approach. The electronic energies of stationary states were adjusted for zero-point vibrational energy (ZPE), and ZPE corrections were scaled by a factor of 0.9804[47]. Gibbs free energy values were computed under the assumption of a temperature of 298.15 K and a pressure of 1 atm within the collision cell. The complete structural details and energies of all relevant structures can be found in the Supplementary Data File. The energies under consideration encompass both electronic and thermal free energies.

The potential energy curve for two R1 cations (Fig. 7a) is calculated as a function of the distance between the two positive charges on the carbonyl carbon atoms by using a flexible scanning plot with the keyword opt=modredundant. This approach is equivalent to a restrictive optimization by fixing the distance between the two carbonyl carbon atoms.

### AIMD calculations
Direct AIMD calculations were performed at the B3LYP/6-31 + G(d) level. DFT calculation was performed by using the Gaussian 09 program. All trajectories were propagated in the velocity-Verlet integration scheme with a time step of 0.1 fs, for a maximum duration of 10 ps. If the distance between the two fragments reaches 15.0 Å, the trajectory is terminated. Many reactive trajectories via the direct mechanism were obtained, while one typical trajectory was shown to explain the reaction process, as depicted in the animation of Supplementary Movie 1.

Because of the significant long-range interaction between the two cation reactants, the trajectories of the bimolecular collision must be initiated at distances greater than 15 Å, where the interaction becomes negligible. This finding is consistent with the experimental conditions. The process of bringing together the two cations from a significant distance is time-consuming, and it is followed by the determination of a reactive or non-reactive trajectory. Consequently, the computational expense is substantial, rendering direct AIMD calculations for bimolecular collisions of the two large cations impractical with traditional initial condition sampling.

Instead, the subsequent approach was utilized to effectively select the initial condition of two reactant cations. Initially, trajectories were generated from the transition state (TS2) by sampling coordinates and momenta from the normal mode distribution representing the zero-point phase space of TS2, along with additional energy required to overcome the barrier, such as 50 kcal mol⁻¹. Secondly, for trajectories leading to the reactant side, the initial velocities were uniformly assigned with opposite directions, and subsequently simulated to ascertain their potential to reach the product side. Based on this criterion, the initial condition of the reactants has been effectively determined, enabling them to surpass the transition state and ultimately form the product. The approach of selecting initial conditions resulted in the successful determination of the correct trajectory evolution originating from the reactant. The initial condition of the specific trajectory indicates a total energy of 15.57 eV, with 5.90 eV attributed to the rovibrational energy of R1, 3.51 eV for the remaining energy, and 6.16 eV for CE.

AIMD simulations were conducted with supplementary CE and rovibrational energies of the reactants. This is especially accurate due to the high barrier (TS2) of 7.2 eV, which necessitates sufficient energies for the reaction to take place. In AIMD simulations of bimolecular reactions, it is essential to incorporate CE and ZPEs of the reactants. Furthermore, to improv efficiency, supplementary rovibrational energies may also be integrated. As for a product-like transition state (TS2) in this reaction, also known as a late-barrier reaction, it is essential for the initial vibrational excitation to effectively enhance reactivity.

## Data availability
The authors declare that all the data that supports the findings of the study are included in the main text and Supplementary Information Files. The source data are available from the corresponding authors on

request. All the data generated in this study have been deposited in the Figshare database under [https://doi.org/10.6084/m9.figshare.25309150].

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

## Acknowledgements

The authors acknowledge Ms. Pinghua Hu and Mr. Shaohua Wang for their help in performing the MS experiments and drawing the picture. The work was supported by National Natural Science Foundation of China (No. 21520102007, 22104014, 22288201, 22174076, 22364002, 22103011) (H.C. B.F. X.K. X.Z. and Y.C.), Jiangxi Provincial Department of Science and Technology (No. 20232BAB213047, 20203BDH80W010) (H.C. and X.Z.), Jiangxi Province International Cooperation Project (20203BDH80W010, 20232BBH80012) (H.C.), Jiangxi University of Chinese Medicine School-level Science and Technology Innovation Team Development Program (CXTD22005, 2004-5252300403) (H.C.), and the Innovation Program for Quantum Science and Technology (2021ZD0303305) (B.F.).

## Author contributions

X.Z. collected the MS data, interpreted the MS data, and co-wrote the manuscript; K.H. improved the reactor (Fig. 2a), validated the reaction data and co-wrote the manuscript; B.F., Y.F., and Yu.C. provided theoretical calculations and the data interpretation; X.K. validate the gas phase data, co-write the manuscript; T.Z., M.Z., and Yi.C. characterized the versions of the reactor (Fig. 2a) and co-collected the data; T.Z. N.Z. partially interpreted the isotope data, polished the manuscript and compiled all the figures; H.C. conceived the idea, designed the reactor (Fig. 2a) of the research, partially interpreted the MS data and co-prepared the manuscript; S.F. supervised the whole project and prepared the manuscript; X.Z., K.H., X.K. B.F. H.C., and S.F. edited the manuscript.

## Competing interests

The authors declare no competing interests.
