## [Peer Review File · Nature Communications]

Demethylation C–C coupling reaction facilitated by the repulsive Coulomb force between two cationsReviewers' Comments:

Reviewer #1:

Remarks to the Author:

This manuscript reports a method for the study of ion reactions - in particular targeting carbon chain elongation (CCE) reactions.

I found it difficult to understand the experimental and instrumental methods in this manuscript.

The description of the instrumentation and the process were too hard to reconcile. Page 4 contains an intricate mixture of letters ("a, b" distances "d" Figures 1c etc) and the figures do not orientate the reader to the processes. It is confusing.

Supporting information Figure S7 depicts the ions trap - but with a picture of the ion trap that is unclear to me what is going on in that trap - what are the electrodes and what are the purple-shaded areas coming off the electrodes?

And what is shown in Figure 1a - this a custom designed capillary array reactor? And where does that fit into the experiment?

I am sorry but I cannot get my head around the experiment - as there is gas-phase ion chemistry but also mentions of an "impact cation" created on a liquid-air interface - is that at the electrospray or in the reactor?

As we move into the results section, its unclear what is gas-phase chemistry and what is solution phase chemistry. And Fig 2 is an intricate assignment "codes" and labels that is, in my opinion, too onerous to decipher. Some clear reaction schemes are needed to assist the reader.

Ultimately, I think there might be good science in here - but the current manuscript needs significant/major improvement to clearly describe the experimental methods and the results. For a general chemistry journal, I'm not sure many general readers will be able to understand the processes. As an ion chemist and mass spectrometrists, I cannot follow the experimental workflow to then properly evaluate the claims of the paper. (The Abstract could be significantly improved too - to outline clearly the experimental are gas-phase (right?) and conducted using ion trap mass spectrometry).

Reviewer #2:

Remarks to the Author:

See in the attached

The manuscript by Zhang *et al.* entitled “Demethylation C–C coupling reaction facilitated by the repulsive Coulomb force between two cations” presents a combined theoretical and experimental study in facilitating the C-C coupling via the formation of carbon-based cations and the application of high voltage. The manuscript suffers from several major flaws that prevent me from giving a positive review at this stage.

- I believe it is erroneous to cite the use of a programmed electric field when according to Figure 1a, a very high voltage (between 1 and 9 kV is cited throughout the manuscript) is applied to the aluminium electrode to essentially promote electrochemical reactions via electro spraying. It is also very unclear which voltage is used at which stage as multiple numbers between 1 to 9 kV are cited throughout the manuscript. The high voltage applied (up to 9 kV, which is equivalent to over thousands of kJ/mol in energy) is a very high voltage that can generate an array of side reactions by breaking even strong C-C bonds. The strong bonding energy of the C-C bond of approximately 350 kJ/mol can be easily broken under such high voltage. Therefore, the authors do not make a convincing argument that it is the electric field that allows the formed carbon-based cations to move close to each other and form a C-C- bond while cleaving the methenium in the process.
- The final products from the carbon-carbon elongation were analysed using ion trap mass spectroscopy. The MS spectra (e.g. Figure 2 as well as multiple MS spectra in supplementary figures) appear to be very messy producing an array of fragmentation products and it is not convincing whether these fragmentation products (see P5 to P9 structures in Figure 2) are not formed in the original experimental setup using the schematic in Figure 1a, especially considering very high voltage was used to generate the proposed product **P4** (see Schematic 2).
- The DFT calculations presented in Figures 3, S13 and S14 are misleading, especially the investigation of Route 2 (see Scheme 3). The authors are very inconsistent as for Route 2 the demethylation step was not studied properly as I do not understand why two R1 initial reagents were used in plotting the Gibbs free energy profile (Figure S13). What the authors should have done is to systematically study the fragmentation mechanism of the C-C bond in a single carbon-based cation to generate a smaller cation of the R1 substituent and a carbene. All R1 groups used experimentally should have been compared here. From the way the data are presented in S13, the Route 2 was done incorrectly, and this is the reason for very high barriers.
- In terms of Route 1 (Figure 3), it is very hard to believe that there is only one transition state in such complex carbon-carbon elongation reaction. The AIMD simulations clearly indicate that as the two carbon-based cations move close to each other their potential energy rises to nearly 8 eV (Figure 4a), which is a very high energy to overcome as it corresponds to over 770 kJ/mol! This is an unsurmountable amount of energy to overcome! The very unconvincing part of the TS1 structure is the fact that the C-C bond is already formed in the transition state and of course the only way to allow for the C-C bond to form is to have the methenium cation liberated! Essentially, I am convinced that the authors could not find a transition state between the two carbon-based radicals until they forced the formation of the C-C bond (which is easily done as DFT calculations are driven by finding the closest minimum in the potential energy surface, especially the C-C bond was shortened to suggest that it was close to a minimum), thus allowing for the methenium cation to be liberated. As a result, I do not believe that the DFT calculations presented showcase the correct comparison of two potential mechanisms as shown in Scheme 3.

- The way the AIMD simulations were performed was to introduce the bias by starting with the TS1 structure and enforcing the C-C bond formation. I am also not convinced that such hugely positive reaction barrier of over 770 kJ/mol could be overcome unless the methenium cation was encouraged to fragment during the simulation. As the authors pointed out that to imitate the true collision such simulations would be computationally infeasible, but the ones presented can hardly be considered definitive either. Without any input of energy (via significantly increased temperature and inclusion of additional potential energy as regularly done in metadynamics) it is very hard to believe that AIMD simulations could indeed form the C-C bond out of two unoccupied 2p orbitals on the carbon cationic centers! The electrons must come from somewhere and the only convincing argument here is that the CH_3^+ cation is liberated leaving the carbene on one of the cationic centers and this is when the C-C bond can be formed!
- I fail to see the impact of this study in general. The cleavage of the C-C bond can be achieved using a very high voltage and this does not mean that such cleavage represents a practical application. The DFT barriers are way too high (the barriers under 120 kJ/mol are considered acceptable experimentally) to see how the mechanisms presented in Scheme 3 could be achieved under standard laboratory conditions. 9kV of voltage is enough to power a single household! This is an incredibly high amount of energy required to make the proposed carbon-carbon elongation viable.

Reviewer #3:

Remarks to the Author:

The authors in this paper present a study of carbon chain elongation reactions made feasible by the application of an electric field. They have devised a reaction set-up where they show the possible formation of a CCE product from two cationic reactants which are generated by the application of a positive potential to an electrospray and to a liquid-air interface. Upon impact, the repulsive Coulombic force leads to the dissociation of one of the cations and leads to the formation of C-C bond. The evidence of this reaction happening is shown through mass spectrometric studies and the mechanism is corroborated by theoretical calculations.

While the overall results of the paper are interesting, my biggest concern with the paper is that to achieve these results, they apply very high potentials to the system. Most organic molecules will be ionized at this potential and this could reduce the overall applicability of the results. Additionally, the authors rely completely on mass spectrometric data analysis to prove that they are indeed observing a CCE product. Moreover, Zare, Cooks and others have recently shown many chemical reactions which are made feasible/accelerated in charged droplets, but their works have not been referred to. (eg: 10.1002/anie.201507805)

While the results of the paper are interesting and could be of potential use to develop more CCE chemistry, I think that there are several questions which the authors must address before this can be considered for publication in nature comm.

Major points:

1. The electrochemical set up is not very well described. It is not clear why spray has pulsed voltage while the plates have constant voltage. And how this configuration allows the authors to achieve the right magnitudes of electric fields. Moreover, it will be good to provide a value of the electric field generated at the interface.
2. It seems to be that there are two set of experiments conducted in this article. The one with the electrochemical setup and another with mass spec using electrospray ionization. Why are the results from the cell not discussed? What are yields of the CCE product from the cell? What other techniques are used to verify the formation of the final product? If no other techniques were used, I suggest that the authors use NMR, UV-Vis or FTIR techniques to show the final product formation as well as report the yields of the product from the experimental setup.
3. The mass spec experiment for the CCE product formation is essentially a gas phase reaction. Therefore, how these fare in the solution phase in the actual experimental setup will be a good comparison.
4. It is clear from the MS data that the major product of the reaction is an allylic cation species. The authors have given a proposed mechanism for this product in the SI, but it will also be worth mentioning in the main text. It will be good to discuss the origin of this species and why that dominates instead of the CCE product.
5. The authors do not discuss what happens if only the electrospray is charged but not the interface. Do they still form CCE products?
6. If two distinctive cations are generated in the spray and at the interface, do the authors see more of homo-dicationic reaction or hetero-dicationic reaction?
7. In teh SI, some reactions are mentioned which have 76% yield, why do the yields of these reactions vary from less than 1% to 76% should be discussed, ie., what factors control these yields?

Minor points

1. The writing of the paper needs to be revised. There are grammatical errors throughout and some sentences are too convoluted to understand. For example, in the methods section, authors have forgotten to remove the guidelines in the start.
2. On page 4, it is not clear why the second solution (at the interface) has acetic acid and salts but no acetone. As the whole paper is discussed with sodiated acetone cation reaction with another sodiated cation, it will be better to use that for both cation a and b examples.
3. In the methods section, the voltage range mentioned is different than that on page 4.
4. On page 7, scheme 2 is incorrect. To neutralize a cation, we will need an anion, while X⁻ will serve

as an anion, H radical will still leave the final molecule as a cation.

5. Figure 2 b and c are mislabelled.

6. On page 12, last sentence, specify which section of SI is being referred to.

7. Please redo the caption of figure S3 to make it more succinct.

Dear reviewers,

We thank you for encouraging and constructive comments. These concerns are addressed point-by-point below.

For your convenience, we have listed all the changes made in the revised version on the following pages. Please note that all page numbers, line numbers, reference numbers, etc. in this list refer to the *revised manuscript*. We hope that you now find this manuscript acceptable for publication in *Nature Communications*.

Changes made in manuscript in response to referees' comments and questions (All page numbers, line numbers, etc. refer to the *revised version* of the manuscript)

REVIEWER COMMENTS

Reviewer #1 (Remarks to the Author):

Comments to the Author

1. This manuscript reports a method for the study of ion reactions - in particular targeting carbon chain elongation (CCE) reactions. I found it difficult to understand the experimental and instrumental methods in this manuscript.

⇒ We thank the reviewer for pointing out these questions. We apologized for the unclear description in the experimental and instrumental methods. Two series of experiments were performed in this study. One was liquid phase experiments occurred in the gas-liquid interface of the reaction vessel by using device in Fig. 1a, the other was gas phase experiments occurred in the ion trap. Fig. 1 (see below) has been remodified in the revised manuscript as suggested.

The uncleared description in the experimental and instrumental methods has been remodified in the revised manuscript. It reads “As depicted in Fig. 1a, the self-constructed experimental setup comprised four primary components: (1) capillary spray array, (2) parallel-plate electric field, (3) reaction vessel, and (4) electric field time-control system. The fundamental principle involves that generation of charged cations at one end through a capillary electrospray array, while at the other end, charged cations are also formed through a parallel electric field, enabling the migration of the cations to the surface of the solution. The cations generated at the electrospray array interact with the solution’s surface, leading to the formation of a demethylation C–C coupling product. This product can subsequently capture H radicals or other anions in the solution, resulting in the formation of stable CCE-related products. Utilizing the experimental apparatus depicted in Fig. 1a, positively charged species, such as cations **RI**, were produced through electrospray array and subsequently reacted with the exposed cations **RI** generated at the liquid-air interface. In detail, using sodiated acetone as an example, the capillary electrospray array, involved inserting all the capillaries (ID 0.19 mm) into a solution of sodiated acetone at one end. The sodiated acetone solution was then charged with high voltage (1-6 kV) via an electrode to generate an array of charged droplets **RI** as depicted in Fig. 1a. The distance from the capillary electrospray array end to the surface of the solution in the reaction vessel was approximately 5 mm. In order to induce the generation of cations **RI** in the reaction vessel, a parallel-plate electric field was constructed. This involved securing a conductive silicon wafer at the base of the reaction vessel and connecting it to a high-voltage power supply, thereby applying a voltage of 1-2 kV. A copper electrode was immersed in the solution and linked to Ground 1 to dissipate the static electricity produced by CCE reaction. The electric field time-control system was employed for the regulation of the high voltage setting of various components such as Al-plate electrode, spray, coated silicon, and copper electrode. (Fig. 1a). In this study, the voltage values were consistent and constant in the spray and Al-plate

electrodes, whereas the voltage in the coated silicon exhibited a pulsed mode. The constant voltage for the spray aims to produce an array of *RI* cations, whereas the constant voltage for Al-plate is intended to create parallel electric fields with the copper electrode connected to Ground 2. The purpose of the parallel-plate electric field between the coated silicon and the copper electrode connected to Ground 2 was to produce *RI* cations. These cations then moved to the surface of the solution and subsequently interacted with *RI* cations generated from the spray. The pulsed voltage applied to the coated silicon was utilized for discharging the static electricity produced from CCE reaction involving two cations. To achieve the appropriate magnitudes of electric fields, the research meticulously regulated the distances between Al-plate electrode, coated silicon, and copper electrode. For instance, the voltage for the spray and Al-plate electrode were set at 6 kV, whereas the voltage for the coated silicon was set at 2 kV. The distance between the Al-plate electrode and the coated silicon is twice that between the coated silicon and the copper electrode. Following the operation of the electric field time-control system, *RI* cations from capillary produce by the capillary electrospray array reacted with *RI* cations present at the liquid-air interface, which were generated from the reaction vessel through a parallel-plate electric field, resulting in the formation of CCE products. The formed CCE products underwent additional analysis using MS and other techniques. **MS for detection CCE products in liquid phase.** All MS experiments related to CCE products formed in the liquid phase were performed using a linear ion trap (LTQ)-XL advantage ion-trap mass spectrometer equipped with a custom-made electrospray ionization (ESI) interface (Thermo Fisher Scientific Inc., San Jose, USA). LTQ mass analyzer was utilized in the positive-ion detection mode, and mass spectra were acquired within the m/z range of 15-400. The formation of $[M + H]^+$ or $[M + Na]^+$ ions was achieved through ESI under the specified typical conditions, which were optimized by adjusting various experimental parameters such as capillary temperature, and tube lens voltage. LTQ mass analyzer was additionally utilized in the negative-ion detection mode, and

mass spectra were acquired within the m/z range of 15-200. The formation of ($M + \text{CH}_3\text{COO}^-$) ions was achieved using the classical ESI-MS method. The optimized conditions were established as follows: the spray voltage was maintained at 3-6 kV, nitrogen gas (purity > 99.999%) was supplied at a pressure of 0.4 MPa, the capillary temperature was set to 320 °C, the capillary voltage was adjusted to 30 V, the tube lens voltage was set at 20 V, and sample solutions were delivered into the ESI source via a syringe at a flow rate of 5 $\mu\text{L min}^{-1}$. The angle formed between the electrospray beam and the MS inlet of LTQ-MS instrument was 150°. MS/MS conditions were as follows: the ion trap pressure was maintained at 1×10^{-5} torr, CE was ranged from 20-30%, and the isolation width was set at 2 Da. The system automatically optimized the other LTQ-XL parameters.”. (see Pages 22-24, Lines 431-489)

Fig. 1 | Schematic illustration of the experimental setup and the possible process of CCE involving two organic cations. **a** Experimental setup for CCE reaction between two RI cations. **b** Mass spectrum of CCE products formed between two sodiated acetone cations. **c** Process of two sodiated acetone cations approached to each other. (Note: RA: Relative abundance; initial stage occurs

when $d_0 > d_1$; repelling and coupling stage occurs when $d_2 < d_0 < d_1$), where d_0 is the distance between the two positive charges located on the carbonyl carbon atoms of the two **RI** cations, d_1 is the effective radius of the reaction cross-section of the two **RI** cations, and d_2 is C–C bond length.

2. The description of the instrumentation and the process were too hard to reconcile. Page 4 contains an intricate mixture of letters ("a, b" distances "d" Figures 1c etc) and the figures do not orientate the reader to the processes. It is confusing.

⇒ We thank the reviewer for pointing out these problems. We have made a consistent description in the instrumentation and the process in the Methods section (See Response 1 above) and the Results and discussion section as suggested. It reads “Fig. 1a depicts a custom-designed reaction apparatus (further described in the Methods section) utilized for CCE reactions (Scheme 1) involving two positively charged organic species (e.g., two sodiated acetone, **RI**) under the influence of a controlled electric field. The apparatus comprised a capillary spray array, parallel-plate electric field, reaction vessel, and an electric field time-control system. Through precise timing manipulation (Fig. 1a), the cation **RI**, produced by electro spraying a 1:1 (v/v) sodiated acetone solution, was propelled by a high voltage (+1-6 kV) to collide with another **RI** cation, generated at the liquid-air interface by applying a positive voltage (+1-2 kV) to the sodiated acetone solution in the reaction vessel (refer to a partial enlargement of Fig. 1a). The electric field intensity can be attained at approximately 10^9 V/m on gas-liquid interface of the cations reaction region ²⁴. The demethylation C–C coupling product, namely sodium 2-methyl-butane-2,3-diolate-3-ium and CH_3^+ , were generated as a result of CCE reaction involving two **RI** cations (Scheme 1).”. (see Pages 5-6, Lines 92-104)

⇒ We have corrected the confusing section in Page 4 and Fig. 1 (see below) according to the suggestion in the revised manuscript. The description of “cation a, b” and “distance d” has been replaced by “two **RI** cations” and “ d_0 ”,

respectively. In response, the content of “Initially, the distance (d) between the positive charges sitting on the carbonyl carbon atoms of the cations a , b was far larger than R_0 (Fig. 1c), the repel effect induced by repulsive Coulomb force (RCF) between the a , b cations was too small (~ 0 eV) to trigger effective interaction between a and b . When d gradually decreased, the RCF between a and b continuously increased and became a more and more dominant force. When $d = R_0$, the a and b cations began to contact on the edge of the reaction cross-section. Due to inertia, a ion still moved towards b , making d continue to decrease. Once $d < R_0$ (Fig. 1d), the repel effect generated by RCF might be significantly larger than the C–C bond energy, which induced the elimination of a small cleavage cation (typically methyl, amino, phenyl cation) and the coupling of the two carbonyl carbon atoms, resulting in positively charged CCE products.” **has been changed into** “Initially, the distance (d_0) between the positively charged particles located on the carbonyl carbon atoms of the two RI cations was significantly greater than the effective radius (d_I) of the reaction cross-section of the two RI cations (Fig. 1c). As a result, the repulsive energy generated by repulsive Coulomb force (RCF) between the two RI cations was insufficiently small (approximately 0 eV is required) to initiate effective interaction between two RI cations. As d_0 gradually decreased, RCF between two RI cations steadily increased and emerged as a dominant force. When $d_0 = d_I$, the two RI cations initiate contact at the edge of the reaction cross-section. Owing to its inertia, RI ion continued to move towards another RI , resulting in a further decrease in d_0 . When $d_0 < d_I$, the repulsive energy produced by RCF may exceed the energy of C–C bond, leading to the expulsion of a small cleavage cation (usually methyl, amino, or phenyl cation) and the connection of the two carbonyl carbon atoms. This process results in the formation of positively charged CCE products (Fig. 1c). Following electric neutralization, CCE products were collected for subsequent characterization. Consequently, RCF between two positively charged organic species (*i.e.*, protonated or sodiated species) served as an easily accessible form of “virtual catalyst” to enhance CCE and

functional group assembly.” in the revised manuscript. (see Pages 7-8, Lines 129-144)

Fig. 1 | Schematic illustration of the experimental setup and the possible process of CCE involving two organic cations. **a** Experimental setup for CCE reaction between two *RI* cations. **b** Mass spectrum of CCE products formed between two sodiated acetone cations. **c** Process of two sodiated acetone cations approached to each other. (Note: RA: Relative abundance; initial stage occurs when $d_0 > d_1$; repelling and coupling stage occurs when $d_2 < d_0 < d_1$), where d_0 is the distance between the two positive charges located on the carbonyl carbon atoms of the two *RI* cations, d_1 is the effective radius of the reaction cross-section of the two *RI* cations, and d_2 is C–C bond length.

3. Supporting information Figure S7 depicts the ions trap - but with a picture of the ion trap that is unclear to me what is going on in that trap - what are the electrodes and what are the purple-shaded areas coming off the electrodes?

⇒ We thank the reviewer for pointing out these problems. We have redrawn the picture of the ion trap in Fig. S13 (see below) to describe what is going on in the trap as suggested. What is going on in the ion trap is mainly including ions

accumulation and isolation (Fig. S13a), ions activation and fragmentation (Fig. S13a). The electrodes were ring electrode and end cap electrode, which were marked in the Fig. S13a. The purple-shaded areas coming off the electrodes were collision gas He, which have been removed. In a response, the content of “The function of ion trap is mainly including ions accumulation and isolation, ions activation and fragmentation. The electrodes are ring electrode and end cap electrode, which are marked in the Supplementary Fig. 13a.” has been added in the supporting information. (see Page 18, Lines 196-198 in Supplementary Material)

Fig. S13. Schematic picture of the process of CCE reaction occurring inside an ion trap. a The schematic illustration about the process of ion trap for gas phase CCE product formation. **b** Process of multistage tandem mass spectrometry experiments in the ion trap.

4. And what is shown in Figure 1a - this a custom designed capillary array reactor?
And where does that fit into the experiment?

⇒ We thank the reviewer for pointing out this problem. We apologize for the

unclear description. The setup of Fig. 1a is designed and applied for collision reaction of two cations, which mainly including capillary spray array, parallel-plate electric field, reaction vessel, electric field time-control system, etc. The function of capillary spray array is to generate abundant cations *RI* used as reactants. Fig. 1 (see below) has been modified in the revised manuscript. In a response, the content of “Fig. 1a depicts a custom-designed reaction apparatus (further described in the Methods section) utilized for CCE reactions (Scheme 1) involving two positively charged organic species (e.g., two sodiated acetone, *RI*) under the influence of a controlled electric field. The apparatus comprised a capillary spray array, parallel-plate electric field, reaction vessel, and an electric field time-control system. Through precise timing manipulation (Fig. 1a), the cation *RI*, produced by electro spraying a 1:1 (v/v) sodiated acetone solution, was propelled by a high voltage (+1-6 kV) to collide with another *RI* cation, generated at the liquid-air interface by applying a positive voltage (+1-2 kV) to the sodiated acetone solution in the reaction vessel (refer to a partial enlargement of Fig. 1a). The electric field intensity can be attained at approximately 10^9 V/m on gas-liquid interface of the cations reaction region²⁴. The demethylation C–C coupling product, namely sodium 2-methyl-butane-2,3-diolate-3-ium and CH_3^+ , were generated as a result of CCE reaction involving two *RI* cations (Scheme 1).” has been added in the revised manuscript. (see Pages 5-6, Lines 92-104)

Fig. 1 | Schematic illustration of the experimental setup and the possible process of CCE involving two organic cations. **a** Experimental setup for CCE reaction between two *RI* cations. **b** Mass spectrum of CCE products formed between two sodiated acetone cations. **c** Process of two sodiated acetone cations approached to each other. (Note: RA: Relative abundance; initial stage occurs when $d_0 > d_1$; repelling and coupling stage occurs when $d_2 < d_0 < d_1$), where d_0 is the distance between the two positive charges located on the carbonyl carbon atoms of the two *RI* cations, d_1 is the effective radius of the reaction cross-section of the two *RI* cations, and d_2 is C–C bond length.

5. I am sorry but I cannot get my head around the experiment - as there is gas-phase ion chemistry but also mentions of an "impact cation" created on a liquid-air interface - is that at the electrospray or in the reactor?

⇒ We thank the reviewer for pointing out these problems. We are sorry for the confused description. An "impact cation" created on a liquid-air interface is generated from the reactor, which is shown in the enlarged section of Fig. 1 (see above). We have done two series of experiments. One was liquid phase experiments occurred in the gas-liquid interface of the reaction vessel by using device in Fig. 1a. Based on the basic principle of the CCE product formed by the

collision reaction between two cations, we designed the reaction device in Fig. 1a to investigate the liquid phase reaction between two sodiated acetone cations. The experimental results showed that the CCE products were formed between the collision of two cations, which were generated from electrospray and reaction vessel, respectively. **The other was gas-phase experiments occurred in the ion trap. Both of the two series of experiments were used to investigate the formation of CCE products between two cations.** For example, in the gas phase CID-MS experiments of sodiated acetone (m/z 81), we also observed the intriguing ions m/z 147, which was characterized as the CCE product between two sodiated acetone cations. The formation mechanism of CCE product was further confirmed by theoretical calculations.

⇒ **In a response, the content of “Fig. 1a depicts a custom-designed reaction apparatus (further described in the Methods section) utilized for CCE reactions (Scheme 1) involving two positively charged organic species (e.g., two sodiated acetone, *RI*) under the influence of a controlled electric field. The apparatus comprised a capillary spray array, parallel-plate electric field, reaction vessel, and an electric field time-control system. Through precise timing manipulation (Fig. 1a), the cation *RI*, produced by electrospraying a 1:1 (v/v) sodiated acetone solution, was propelled by a high voltage (+1-6 kV) to collide with another *RI* cation, generated at the liquid-air interface by applying a positive voltage (+1-2 kV) to the sodiated acetone solution in the reaction vessel (refer to a partial enlargement of Fig. 1a). The electric field intensity can be attained at approximately 10^9 V/m on gas-liquid interface of the cations reaction region²⁴. The demethylation C–C coupling product, namely sodium 2-methyl-butane-2,3-diolate-3-ium and CH_3^+ , were generated as a result of CCE reaction involving two *RI* cations (Scheme 1). Upon the collisions of *RI* cations from spray plumes and the liquid-air interface (as shown in a partial enlargement of Fig. 1a), the mass spectrum indicated the generation of products related to demethylation and C–C coupling during CCE reaction between two *RI* cations. In Fig. 1b, the products associated with C–C**

coupling observed at m/z 149, m/z 167 and m/z 183 corresponded to $(MH+H)^+$, $(MF+H)^+$, and $(MCl+H)^+$, respectively. This correspondence is attributed to the addition of H radical, F^- , and Cl^- anions in the solution within the reaction vessel. The process of formation is depicted in Scheme 2. The small CH_3^+ cation, which is produced during CCE process, can be effectively captured by anions such as F^- , Cl^- , Br^- and CH_3COO^- to yield CH_3F , CH_3Cl , CH_3Br and CH_3COOCH_3 , respectively, in the reaction vessel's solution (Scheme 2). The product signals at m/z 93, m/z 109, m/z 153 and m/z 133, which result from the formation of (CH_3X+Ac^-) adducts, were also identifiable in the negative ion detection mode (Supplementary Fig. 1a). These findings were subsequently validated through collision induced dissociation mass spectrometry (CID-MS) analysis (Supplementary Fig. 1b-1e).” has been added in the revised manuscript. (see Pages 5-7, Lines 92-128) We hope that the newly revised version can give review a good understanding of the reaction process.

6. As we move into the results section, it's unclear what is gas-phase chemistry and what is solution phase chemistry. And Fig 2 is an intricate assignment "codes" and labels that is, in my opinion, too onerous to decipher. Some clear reaction schemes are needed to assist the reader.

⇒ We thank the reviewer for pointing out these problems. We apologize for the confusing description. We have added subheadings and appropriate content to better distinguish the content of gas phase chemistry and liquid phase chemistry. In a response, the corresponding content description has been added in the revise manuscript. It reads “**Demethylation C–C coupling reaction between two sodiated acetone cations in liquid-phase.** Fig. 1a depicts a custom-designed reaction apparatus (further described in the Methods section) utilized for CCE reactions (Scheme 1) involving two positively charged organic species (*e.g.*, two sodiated acetone, **RI**) under the influence of a controlled electric field. The apparatus comprised a capillary spray array, parallel-plate electric field,

reaction vessel, and an electric field time-control system. Through precise timing manipulation (Fig. 1a), the cation **RI**, produced by electro spraying a 1:1 (v/v) sodiated acetone solution, was propelled by a high voltage (+1-6 kV) to collide with another **RI** cation, generated at the liquid-air interface by applying a positive voltage (+1-2 kV) to the sodiated acetone solution in the reaction vessel (refer to a partial enlargement of Fig. 1a). The electric field intensity can be attained at approximately 10^9 V/m on gas-liquid interface of the cations reaction region²⁴. The demethylation C–C coupling product, namely sodium 2-methyl-butane-2,3-diolate-3-ium and CH_3^+ , were generated as a result of CCE reaction involving two **RI** cations (Scheme 1). It is important to note that the products of CCE reactions (Scheme 1) consist of charged species, specifically the methyl cation and the organic cation with a CCE. Under the experimental conditions, the cations were neutralized electronically in two possible ways (Scheme 2) to form stable final products. The first method involved direct grounding to yield a neutral radical, which then captured a hydrogen radical from the solvent environment to produce stable **MH**. The second method involved anion coupling with encountered anions to produce stable **MX** (*i.e.*, **M** = methyl, $\text{R}_2\text{C}(\text{OH}/\text{Na})\text{-C}(\text{OH}/\text{Na})\text{R}_4\text{R}_3$; **X** = F, Cl, Br, I, CH_3COO). Upon the collisions of **RI** cations from spray plumes and the liquid-air interface (as shown in a partial enlargement of Fig. 1a), the mass spectrum indicated the generation of products related to demethylation and C–C coupling during CCE reaction between two **RI** cations. In Fig. 1b, the products associated with C–C coupling observed at m/z 149, m/z 167 and m/z 183 corresponded to $(\text{MH}+\text{H})^+$, $(\text{MF}+\text{H})^+$, and $(\text{MCl}+\text{H})^+$, respectively. This correspondence is attributed to the addition of H radical, F^- , and Cl^- anions in the solution within the reaction vessel. The process of formation is depicted in Scheme 2. The small CH_3^+ cation, which is produced during CCE process, can be effectively captured by anions such as F^- , Cl^- , Br^- and CH_3COO^- to yield CH_3F , CH_3Cl , CH_3Br and $\text{CH}_3\text{COOCH}_3$, respectively, in the reaction vessel's solution (Scheme 2). The product signals at m/z 93, m/z 109, m/z 153 and m/z 133, which

result from the formation of (CH_3X+Ac^-) adducts, were also identifiable in the negative ion detection mode (Supplementary Fig. 1a). These findings were subsequently validated through collision induced dissociation mass spectrometry (CID-MS) analysis (Supplementary Fig. 1b-1e).” **has been added in the revised manuscript. (see Pages 5-7, Lines 82-128).**

⇒ **We have added the reaction scheme in Fig. 2 (see below) as suggested, and we hope that the newly revised Fig. 2 can give you a good understanding of the reaction process.**

Fig. 2 | Reaction and dissociation of sodiated acetone ($R1$, m/z 81) and sodiated acetone- d_6 ($R1-d_6$, m/z 87) in the gas phase. **a Proposed structures for product ions in the dissociation $R1$. **b** Product ions in the MS/MS spectrum of $R1$. **c** MS^3 spectrum of $R1$ (m/z 81 \rightarrow m/z 147 \rightarrow). **d** Product ions in the MS/MS spectrum of $R1-d_6$. **e** MS^3 spectrum of $R1-d_6$ (m/z 87 \rightarrow m/z 156 \rightarrow). (Note: Signal-to-noise ratio was significantly greater than 3. RA denotes Relative abundance.)**

7. Ultimately, I think there might be good science in here - but the current manuscript needs significant/major improvement to clearly describe the experimental methods and the results. For a general chemistry journal, I'm not sure many general readers will be able to understand the processes. As an ion chemist and mass spectrometrists, I cannot follow the experimental workflow to then properly evaluate the claims of the paper. (The Abstract could be significantly improved too - to outline clearly the experimental are gas-phase (right?) and conducted using ion trap mass spectrometry).

⇒ We thank the reviewer for his/her affirmation and comments. We have remodified the uncleared Fig. 1, Fig. 2, Fig. S13 (see Responses 3, 4, 6 above), and made a major improvement to clearly describe the experimental methods and the results as suggested. The details for the experimental methods, please see Response 1 above. The details for the liquid phase experiment results, please see Response 5 above.

⇒ The corresponding content of gas phase ion chemistry has been added in the abstract as suggested. It reads “C–C coupling was commonly observed in both the gas-phase (between cations in an ion trap) and in the liquid phase (between cations in charged droplets). CCE products were characterized using high-resolution mass spectrometry (MS) and isotope-labeling experiments.”. (see Page 2, Lines 29-31) We hope that the newly revised version can give review and general readers a good understanding of the reaction process.

Reviewer #2 (Remarks to the Author):

Comments to the Author

The manuscript by Zhang *et al.* entitled “Demethylation C–C coupling reaction facilitated by the repulsive Coulomb force between two cations” presents a combined theoretical and experimental study in facilitating the C-C coupling via the formation of carbon-based cations and the application of high voltage. The manuscript suffers from several major flaws that prevent me from giving a positive review at this stage.

⇒ **We thank the reviewer for pointing out these problems. We have made detailed revisions in the experimental and calculation section as suggested.** The gas phase ion chemistry and liquid ion chemistry experiments were both performed and confirmed the formation CCE product between the collision of two sodiated acetone cations. The results of CID-MS experiments, isotope experiment, NMR experiments, references experiments, and theoretical calculations, etc. strongly supported the proposed reaction mechanism and our conclusion. The only drawback as the reviewer pointed out is that the energy barrier required for the reaction in our work is very high. **However, this high energy is actually reasonable, because it needs to overcome the coulomb repulsion between the two sodiated acetone cations and further induce the C–C bond dissociation and the other C–C bond formation.** In our reaction system, there was sufficient energy source to facilitate the reaction. For example, in the gas phase, sodiated acetone cations could obtain the collision energy via CID process in the ion trap. It was calculated that the relative collision energy of the two sodiated acetone cations could reach ~ 7.3 eV (Fig. 2b), which was sufficient to promote the formation of CCE product. In the liquid phase, a high electric field ($\sim 10^9$ V/m) at the air-water interface of microdroplets could provide sufficient energy and was responsible for the reaction acceleration (~ 10000 times), which promoted the formation of CCE product. Further, we tried to find other possible TS and CCE product formation path, which also have high energies (see below).

⇒ In a typical CID experiment, most the of dissociation reaction generally requires high energy, because the average internal energy deposited in the precursor ion is much larger than the activation energies required for unimolecular dissociation reaction. For example, it was reported that the energy barrier of the elimination of HCN from CID of guanine radical cations is ~ 4.8 eV (Phys. Chem. Chem. Phys., 2010, 12, 4667-4677). The energy barrier of the elimination of C_2H_4 from CID of $[CH_3CuCH_3]^+$ is ~ 3.5 eV (Acc. Chem. Res. 2015, 48, 329–340). For double charge dissociation, it requires higher energy because of the addition of overcoming the

Coulomb repulsion. For example, the energy for the formation of ions $\text{NH}_3\text{CH}_2\text{CH}_2^+$ (m/z 45) and $[\text{NH}_2\text{CH}_2\text{CH}_2\text{CO}\dots\text{CO}]^+$ (m/z 100) from the dissociation of $[(\text{alanine})_2+\text{H}]^{2+}$ is ~ 7 eV (Nat. Commun. 2020, 11, 3818). These reactions can indeed occur even though the high energy, due to the available sufficient energy source. In our work, two **RI** cations collide with each other with a certain kinetic energy to form a coupling product through **TSI**. This process was also needed to overcome the large Coulomb repulsion. In addition, the sum energy of CCE products (**PA** and CH_3^+) is also very high, reaches ~ 6 eV. Thus, the high energy barrier is reasonable for this CCE reaction. This reaction cannot occur in conventional experiments without extra energy, but can occur in vacuum conditions with collision energy or in ambient conditions with ultra-high electric field acceleration. We hope a clear and well-organized revised manuscript fully address all the concerns raised by the reviewer.

1. I believe it is erroneous to cite the use of a programmed electric field when according to Figure 1a, a very high voltage (between 1 and 9 kV is cited throughout the manuscript) is applied to the aluminium electrode to essentially promote electrochemical reactions via electrospraying. It is also very unclear which voltage is used at which stage as multiple numbers between 1 to 9 kV are cited throughout the manuscript. The high voltage applied (up to 9 kV, which is equivalent to over thousands of kJ/mol in energy) is a very high voltage that can generate an array of side reactions by breaking even strong C-C bonds. The strong bonding energy of the C-C bond of approximately 350 kJ/mol can be easily broken under such high voltage.

Therefore, the authors do not make a convincing argument that it is the electric field that allows the formed carbon-based cations to move close to each other and form a C-C- bond while cleaving the methenium in the process.

⇒ **We thank the reviewer for pointing out these problems. We apologize for our carelessness. The spray voltage and Al-plate voltage we used were 1-6 kV. We have modified the voltage range to be consistent in the revised manuscript.** (Annu. Rev. Anal. Chem. 2010, 3, 231-254; Anal. Chem. 2006, 78, 24, 8549–8555).

In general, the voltage applied to electrospray is ~3-6 kV for single needle to form charged droplets in ESI-MS, which generally formed classical $[M + H]^+$, $[M - H]^-$ ions (J. Am. Chem. Soc. 2023, 145, 29, 15674–15679; PNAS, 2015, 112, 3898-3903; Annu. Rev. Anal. Chem. 2010, 3, 231-254; Anal. Chem. 2006, 78, 24, 8549–8555). In addition, electrochemical redox reactions in ESI source may result in several unexpected ions, such as M^+ , $[M - H]^+$, $[M + H]^-$ ions, in which the corresponding compounds have lower oxidation potential (J. Am. Chem. Soc. 2022, 144, 16184-16190; Analyst, 2011, 136, 4667). In our work, acetone is a stable molecule with higher oxidation potential, which is hardly to be oxidized. When 1-6 kV voltage was applied on the spray of sodiated acetone solution or acetone solution, the base peak ions were $[M + Na]^+$ m/z 81 or $[M + H]^+$ m/z 59 (see Fig. S12 below), the ions related to electrochemical redox reaction or C-C bonds breaking reaction were not observed. Therefore, 1-6 kV voltage on spray was used to generate charged droplets, e.g., $R1$ cations. In a response, the content of “Furthermore, CCE product, corresponding fragments, ions associated with electrochemical redox reactions, and C–C bond cleavage reactions were not detected during the high-voltage spraying of sodiated acetone solution or acetone aqueous solution (Supplementary Fig. 12).” has been added in the revised manuscript. (see Page 10, Lines 187-189)

⇒ In this work, a spray voltage about 1-6 kV was applied to the reaction solution in Fig. 1a and passed through a capillary array to form a spray array of charged microdroplets. Spray and Al-plate both have the same and constant voltage, while coated silicon has pulsed voltage with 1-2 kV. The purpose of constant voltage for spray is to generate $R1$ ion array, while constant voltage for Al-plate is to form parallel electric fields with Ground 2 at the bottom of the reaction vessel. The purpose of pulsed voltage for coated silicon is to generate another $R1$ cations to collide with $R1$ cations formed by spraying. The purpose of pulsed mode for copper electrode (connected to Ground 1) is to release static electricity. The spray and Al-plate were worked continuously.

When the coated silicon worked, the copper electrode is not worked. While when the copper electrode is worked the coated silicon is not worked. Fig. 1 (see below) has been remodified in the revised manuscript. In a response, the content of “Utilizing the experimental apparatus depicted in Fig. 1a, positively charged species, such as cations *RI*, were produced through electro spray array and subsequently reacted with the exposed cations *RI* generated at the liquid-air interface. In detail, using sodiated acetone as an example, the capillary electro spray array, involved inserting all the capillaries (ID 0.19 mm) into a solution of sodiated acetone at one end. The sodiated acetone solution was then charged with high voltage (1-6 kV) via an electrode to generate an array of charged droplets *RI* as depicted in Fig. 1a. The distance from the capillary electro spray array end to the surface of the solution in the reaction vessel was approximately 5 mm. In order to induce the generation of cations *RI* in the reaction vessel, a parallel-plate electric field was constructed. This involved securing a conductive silicon wafer at the base of the reaction vessel and connecting it to a high-voltage power supply, thereby applying a voltage of 1-2 kV. A copper electrode was immersed in the solution and linked to Ground 1 to dissipate the static electricity produced by CCE reaction. The electric field time-control system was employed for the regulation of the high voltage setting of various components such as Al-plate electrode, spray, coated silicon, and copper electrode. (Fig. 1a). In this study, the voltage values were consistent and constant in the spray and Al-plate electrodes, whereas the voltage in the coated silicon exhibited a pulsed mode. The constant voltage for the spray aims to produce an array of *RI* cations, whereas the constant voltage for Al-plate is intended to create parallel electric fields with the copper electrode connected to Ground 2. The purpose of the parallel-plate electric field between the coated silicon and the copper electrode connected to Ground 2 was to produce *RI* cations. These cations then moved to the surface of the solution and subsequently interacted with *RI* cations generated from the spray. The pulsed voltage applied to the coated silicon was utilized for discharging the static electricity produced from CCE

reaction involving two cations. To achieve the appropriate magnitudes of electric fields, the research meticulously regulated the distances between Al-plate electrode, coated silicon, and copper electrode. For instance, the voltage for the spray and Al-plate electrode were set at 6 kV, whereas the voltage for the coated silicon was set at 2 kV. The distance between the Al-plate electrode and the coated silicon is twice that between the coated silicon and the copper electrode.” **has been added in the revised manuscript. (see Pages 22-23, Lines 440-466)**

- ⇒ In the electrospray process of Fig. 1a, although several kV high voltage is applied, the current generated in this process is very small (~ several μA), thus the power consumed is very low (~ several mW, namely ~several mJ/s). Therefore, a voltage of several kV is not equal to several kJ/mol of energy.
- ⇒ We have performed some control experiments (See Figs. S9-12 below) to prove that the formed sodiated acetone cations move close to each other and form a C-C bond while cleaving the methenium in the process. In a response, the content of “As demonstrated in the experiment (Supplementary Table 2), CCE reactions only occurred between two positively charged species. The control experiments were conducted comparing the effects of spray 6 kV and coated silicon 0 kV with spray 6 kV and coated silicon 2 kV. The control experiments demonstrate that no CCE products (Supplementary Fig. 9a and 9d) were detected when only the electrospray was charged without the coated silicon. However, when both the spray and coated silicon were charged, corresponding CCE products (m/z 149 and m/z 171 for sodiated acetone; m/z 105 for protonated acetone) were observed (Supplementary Fig. 9b and 9e). CCE products have also been validated through the corresponding tandem MS experiments (Supplementary Fig. 9c and 9f). The reason for this is that when only the spray is charged, the reactant ions formed are immediately grounded and neutralized without undergoing effective collision. It was observed that conducting two sprays (spray1, 3 kV, spray2, 0 kV vs. spray1, 3 kV, spray2, 2 kV) using online MS analysis yielded consistent results (Supplementary Fig. 10) with those obtained using the apparatus in Fig. 1a

(Supplementary Fig. 9). It was also observed that in case where two different cations were present in both the spray and the reaction vessel, both homo-dicationic and hetero-dicationic reaction products could be observed (Supplementary Fig. 11). Interestingly, the intensity of the hetero-dicationic reaction products appeared to be higher than that of the homo-dicationic reaction products. The issue may stem from a probability problem. Furthermore, CCE product, corresponding fragments, ions associated with electrochemical redox reactions, and C–C bond cleavage reactions were not detected during the high-voltage spraying of sodiated acetone solution or acetone aqueous solution (Supplementary Fig. 12).” **has been added in the revised manuscript. (see Pages 9-10, Lines 170-189)**

⇒ **In addition, if the formation of CCE product is through the ion/molecule reaction (Route 2 in Scheme 3b), carbene intermediate should be generated. However, no targeted nitrile ylide ions (Fig. S26 see below) at m/z 86 (CH_3CN capture $\text{CH}_3\text{HOC:}$) or m/z 108 (CH_3CN capture $\text{CH}_3\text{NaOC:}$) using the experimental setup shown in Fig. 1a, with CH_3CN as the carbene intermediates capturing reagent (*J. Am. Chem. Soc.* 2020, 142, 7836-7844). Therefore, the formation of m/z 147 is likely through the CCE reaction by elimination of a methyl cation with a concerted mechanism, which is in accordance with the DFT calculation results. In a response, the content of “No carbene intermediate (Supplementary Fig. 26) was detected with greater certainty during CCE reactions conducted using the experimental apparatus depicted in Fig. 1a, wherein the solution within the reaction vessel was supplemented with CH_3CN (1-50%, v/v) to potentially react with any carbene intermediate³⁵.” **has been shown in the revised manuscript. (see Page 16, Lines 301-304)****

Fig. 1 | Schematic illustration of the experimental setup and the possible process of CCE involving two organic cations. **a** Experimental setup for CCE reaction between two *RI* cations. **b** Mass spectrum of CCE products formed between two sodiated acetone cations. **c** Process of two sodiated acetone cations approached to each other. (Note: RA: Relative abundance; initial stage occurs when $d_0 > d_1$; repelling and coupling stage occurs when $d_2 < d_0 < d_1$), where d_0 is the distance between the two positive charges located on the carbonyl carbon atoms of the two *RI* cations, d_1 is the effective radius of the reaction cross-section of the two *RI* cations, and d_2 is C–C bond length.

Fig. S12. ESI-MS spectrum of sample solution at the voltage of 6 kV. a Sodiated acetone aqueous solution. **b** Acetone aqueous solution.

Fig. S9. Control experiments for CCE products formation by setup in Fig. 1a. **a** The MS spectrum of CCE reaction of two sodiated acetone cations when the spray charged while the interface not charged. **b** The MS spectrum of CCE reaction of two sodiated acetone cations when the spray and interface both charged. **c** The MS/MS spectrum of CCE product at m/z 149. **d** The MS spectrum of CCE reaction of two protonated acetone cations when the spray charged while the interface not charged. **e** The MS spectrum of CCE reaction of two protonated acetone cations when the spray and interface both charged. **f** The MS/MS spectrum of CCE product at m/z 105.

Fig. S10. Control experiments for CCE products formation by online MS. **a** The MS spectrum of CCE reaction of two sodiated acetone cations when the spray 1 charged while the spray 2 not charged. **b** The MS spectrum of CCE reaction of two sodiated acetone cations when the spray 1 and spray 2 both charged. **c** The MS/MS spectrum of

CCE product at m/z 147.

Fig. S11. Mass spectrum of collision reaction when sodiated acetone and sodiated pentanone mixture solution was introduced to the spray and the interface, respectively.

Fig. S26. Data supporting no capture of carbene intermediate by CH₃CN, demonstrating that the Route 2 was unlikely to generate the CCE product obtained by the setup shown in Fig. 1a. a Schematic illustration on the process of the capture of carbene intermediate by CH₃CN. **b** MS spectrum of CCE reaction of protonated/sodiated acetone by adding CH₃CN to capture carbene intermediate (CH₃HOC:, *MW* 44; CH₃NaOC:, *MW* 66) using the experimental setup shown in Fig. 1a. No targeted ions at *m/z* 86 (CH₃CN capture CH₃HOC:) or *m/z* 108 (CH₃CN capture CH₃NaOC:) are detected. For convenience, further details and related discussions of the figure are listed below.

2. The final products from the carbon-carbon elongation were analysed using ion trap mass spectroscopy. The MS spectra (e.g. Figure 2 as well as multiple MS spectra in supplementary figures) appear to be very messy producing an array of fragmentation products and it is not convincing whether these fragmentation products (see P5 to P9 structures in Figure 2) are not formed in the original experimental setup using the schematic in Figure 1a, especially considering very high voltage was used to generate the proposed product **P4** (see Schematic 2).

⇒ We thank the reviewer for pointing out these problems. We apologized for the confusing representation. Using the schematic in Fig. 1a (the electrospray section), it will not form the fragment ions P5 to P9 shown in Fig. 2b. Electrospray ionization is a soft ionization technique that forms classical ions such as [M+H]⁺/[M+Na]⁺ ions rather than the fragment ions (Angew. Chem. Int. Ed., 2019, 58 (35), 11967-11977), while the fragment ions were produced in the tandem MS techniques such as CID-MS. In our experiments, the formed classical ions are [acetone+Na]⁺ or [acetone+H]⁺, while no fragment ions **P5-P9** in Fig. 2b was observed when spraying sodiated acetone aqueous solution or acetone aqueous solution in ESI-MS experiments (see Fig. S12 below). **In a response, the content of “Furthermore, CCE product, corresponding fragments, ions associated with electrochemical redox reactions, and C–C bond cleavage reactions were not detected during the high-voltage spraying of sodiated acetone solution or acetone aqueous solution (Supplementary Fig. 12).” has been added in the revised manuscript. (see Page 10, Lines 187-189)**

⇒ As the reviewer said, the original Fig. 2 is very messy producing an array of fragmentation products, which were obtained using the ion trap device (see Supplementary Fig. 7) in the gas phase. The collision energy is supplied to isolated cations R1. Study of reaction in the ion trap is to investigate the gas phase ion chemistry and structure elucidation of CCE product. While the study in Fig. 1a is microdroplet liquid ion chemistry. The main formed ions in Fig. 1a are R1 cations, and CCE products, no fragment ions of P5-P9. Therefore, the fragmentation products of P5-P9 in Fig. 2b are not formed in the original experimental setup of Fig. 1a. We have remodified Fig. 2 (see below) and the corresponding content description, which can clear display the generation process of these fragment ions. It reads “Under the specified experimental parameters, the sodiated acetone **RI** (m/z 81) did not yield any fragments or adducts in MS/MS spectrum (Supplementary Fig. 14) when subjected to low CE ($CE < 1.6$ eV). However, it did generate abundant product ions (Fig. 2a and 2b, Supplementary Fig. 14) at m/z 41, m/z 99, and m/z 147 when exposed to relatively high CE ($CE \geq 3.6$ eV). The prominent peak observed at m/z 41 (Fig. 2b) corresponds to the dissociation products of **RI** (m/z 81), indicating a combination of the allyl cation (**PI**) and hydrated sodium cation (**P2**) through the elimination of NaOH and propadiene from the dissociation of **RI**, as illustrated in Fig. 2a, Supplementary Scheme 1a, and Supplementary Fig. 15. The proposed structures of the allyl cation and hydrated sodium cation have been validated through isotope labeling experiments (Fig. 2d, Supplementary Scheme 1b), a kinetic isotope effect of $k_H/k_D = 5$, and theoretical calculations (detailed in the subsequent calculation section). The dominance of the allyl cation (**PI**) and the hydrated sodium cation (**P2**) may be attributed to the stability of the product ions, which are more stable than **P4** carbocation. The detailed information was illustrated in the following calculation results section.”. (see Pages 11-12, Lines 197-216)

Fig. S12. ESI-MS spectrum of sample solution at the voltage of 6 kV. a Sodiated acetone aqueous solution. **b** Acetone aqueous solution.

Fig. 2 | Reaction and dissociation of sodiated acetone (*R1*, m/z 81) and sodiated acetone- d_6 (*R1-d₆*, m/z 87) in the gas phase. a Proposed structures for product ions in the dissociation *R1*. **b** Product ions in the MS/MS spectrum of *R1*. **c** MS³ spectrum of *R1* (m/z 81 \rightarrow m/z 147 \rightarrow). **d** Product ions in the MS/MS spectrum of *R1-d₆*. **e** MS³ spectrum of *R1-d₆* (m/z 87 \rightarrow m/z 156 \rightarrow). (Note: Signal-to-noise ratio was significantly greater than 3. RA denotes Relative abundance.)

3. The DFT calculations presented in Figures 3, S13 and S14 are misleading, especially the investigation of Route 2 (see Scheme 3). The authors are very inconsistent as for Route 2 the demethylation step was not studied properly as I do not understand why two R1 initial reagents were used in plotting the Gibbs free energy profile (Figure S13). What the authors should have done is to systematically study the fragmentation mechanism of the C-C bond in a single carbon-based cation to generate a smaller cation of the R1 substituent and a carbene. All R1 groups used experimentally should have been compared here. From the way the data are presented in S13, the Route 2 was done incorrectly, and this is the reason for very high barriers.

⇒ **We thank the reviewer for pointing out these problems. In original Fig. S13, the purpose of plotting the Gibbs free energy profile with two R1 initial reagents is to compare the energies with that in Fig. 3. Obviously, this misleads the reviewer. Therefore, we have redrawn the corresponding figures (Fig. 3, Figs. S21, 22 see below) based on the reviewer's comments. What we studied in Route 2 was the process of C-C bond dissociation to generate CH_3^+ and carbene in a single sodiated acetone, then the formed carbene continue to interact with one sodiated acetone to form CCE product.** In the first step, one single sodiated acetone decomposes into CH_3^+ and carbene, with a very high dissociation energy of about 8.9 eV (Fig. S22 see below). In the subsequent step, the formed carbene interacts with another sodiated acetone, and finally produces the CCE product, with the barrier (TS4) of about 8.2 eV. This is the ion-molecule reaction step. Thus, the energy for Route 2 via the initial dissociation of one single sodiated acetone and the secondary ion-molecule reaction between carbene and sodiated acetone is at least 8.9 eV, which is much higher than that for Route 1. **In response, the content of “For Route 2, the energy required for the initial dissociation of a single sodiated acetone and the subsequent ion-molecule reaction between carbene and sodiated acetone is at least 8.9 eV (Supplementary Fig. 21), significantly exceeding that for Route 1.” has been added in the revised manuscript. (see Page 15, Lines 286-288)**

Fig. 3 | Results of stationary point and AIMD calculations that support CCE process. **a** Relative energy levels for Route 1 for the generation of *P4a* ion via *TS1* through the demethylation C–C coupling reaction. **b** Snapshots of one reactive trajectory via *TS1*. (Note: Each panel depicts the geometry at the specified reaction time (fs). The internuclear distances (Å) between the two C atoms involved in bond formation, and between the two C atoms in detaching CH_3^+ and *P4a* are indicated by red and blue fonts, respectively.)

Fig. S21. Relative energy levels for route 2 to generate *P4a* by the ion-molecule reaction. The unit of the bond distances is angstroms (Å). Purple Na, red O, gray C,

and white H. **R1**: m/z 81, **R4**: $\text{CH}_3\text{NaOC}\cdot$, **P4a**, **P4b**: m/z 147. For convenience, further details and related discussions of the figure are listed below.

Fig. S22. Intrinsic reaction coordinate (IRC) calculations of Route 1 via *TS1*, and Route 1 via *TS2*, *C1*, *TS3* and *C2* for the formation of product ion *P4* (two configurations including *P4a*, *P4b*) m/z 147 at the B3LYP-gcp-D3/6-31+G(d) level. **a** IRC calculations of *TS1* for the formation of product ion *P4a* at m/z 147. **b** IRC calculations of *TS2* for the formation of product ion *P4b* at m/z 147.

⇒ As suggested, we also calculated the dissociation energies of other compounds in this study and compared all R_1 groups (Fig. S23 below). The results show that the initial dissociation process requires very high energies (8.9~14 eV). Therefore, Route 2 for the initial dissociation of R_1 and further ion-molecule reaction is more difficult to occur than Route 1 for the direct demethylation C–C coupling reaction. In response, the content of “Additionally, the study conducted calculations on the dissociation energies of other compounds with

varying R₁ groups, revealing similarly high energy requirements ranging from 8.9 eV to 14 eV (Supplementary Fig. 23) is indicated to be the lowest compared to other R₁ groups. The findings indicate that the initial dissociation of **R1** through Route 2, followed by the ion-molecule reaction, is less likely to occur compared to Route 1, which involves direct demethylation C–C coupling reaction.” **has been added in the revised manuscript. (see Page 15, Lines 288-293)**

Fig. S23. The dissociation energies for different compounds.

4. In terms of Route 1 (Figure 3), it is very hard to believe that there is only one transition state in such complex carbon-carbon elongation reaction. The AIMD simulations clearly indicate that as the two carbon-based cations move close to each other their potential energy rises to nearly 8 eV (Figure 4a), which is a very high energy to overcome as it corresponds to over 770 kJ/mol! This is an unsurmountable amount

of energy to overcome! The very unconvincing part of the TS1 structure is the fact that the C-C bond is already formed in the transition state and of course the only way to allow for the C-C bond to form is to have the methenium cation liberated!

⇒ We thank the reviewer for pointing out these problems. The reviewer mainly doubts if only one transition state (TS1) connected the reactants and products in Route 1. The TS1 directly optimized by the DFT method gives only one imaginary frequency of about -261.64 cm^{-1} , and 59 real frequencies. The number of frequencies is consistent with that of a direct reaction process. We also checked the output eigenvector of this mode with imaginary mode, as shown in the imaginary frequency mode section of SI (see below), which is the correct direction associated with one C–C bond stretching and the other C–C bond contracting. The IRC plot shown in Fig. S22a (see below) for Route 1 also supports that the reactants are connected with products via TS1. In response, the content of “For Route 1, the key transitions states of *TS1* for *P4a* formation and *TS2* for *P4b* formation were verified through intrinsic reaction coordinate (IRC) calculations (Supplementary Fig. 22), thereby validating the findings presented in Fig. 3a and Supplementary Fig. 20, respectively. The imaginary vibrational frequency of *TS1*, along with the corresponding eigenvector of this vibrational mode, indicated the correct direction associated with the stretching of one C–C bond and the contraction of the other C–C bond (as shown in the imaginary frequency mode section of Supplementary Material).” has been added in the revised manuscript. (see Pages 14-15, Lines 272-286)

⇒ Furthermore, the AIMD snapshots shown in Fig. 3b (see below) and the animation shown in SI support this process. As also indicated in Fig. 4b (see below), the turning point for the internuclear distance between the two carbonyl carbon atoms of sodiated acetones basically occurs at the same time as that for the internuclear distance between the carbonyl carbon and methyl carbon in detaching CH_3^+ and P4a. These results support that the CH_3^+ detaches and the C–C bond forms simultaneously. The potential energies

along this trajectory reach about 8 eV, which is high but reasonable since the TS1 for this pathway is already 7.2 eV. The AIMD calculations were launched for two reactants with additional rovibrational energies and collision energy, and thus the potential energies for this trajectory can go further beyond the transition-state energy. In response, the content of “As depicted in Fig. 4b, the inflection point for the interatomic distance between the two carbonyl carbon atoms of sodiated acetones essentially coincides with that of the interatomic distance between the carbonyl carbon and methyl carbon during the detachment of CH_3^+ and *P4a*. The findings indicate that the detachment of CH_3^+ and the formation of C–C bond occur simultaneously. The potential energies along this trajectory peak at approximately 8 eV, a level that is relatively high but justifiable given that *TS1* for this pathway is already at 7.2 eV. AIMD calculations were initiated for two reactants, incorporating additional rovibrational energies and CE. Consequently, the potential energies for this trajectory may exceed the transition-state energy.” has been added in the revised manuscript. (see Page 18, Lines 350-358)

The imaginary vibrational frequency of TS1, together with the eigenvector of this vibrational mode.

TS1 Frequencies -- -261.6358

Atom	AN	X	Y	Z
1	6	0.04	-0.23	-0.21
2	8	-0.01	-0.07	-0.06
3	6	0.01	-0.02	0.00
4	6	-0.06	0.40	0.34
5	1	-0.07	0.12	-0.09
6	1	0.02	0.27	0.23
7	1	0.02	-0.03	0.06
8	1	0.02	-0.02	0.14

9	1	-0.07	0.31	0.33
10	1	-0.07	0.29	0.17
11	11	0.01	0.01	0.01
12	6	0.03	-0.05	-0.06
13	8	0.03	-0.01	-0.04
14	6	0.04	0.01	-0.03
15	6	-0.06	-0.09	-0.01
16	1	0.09	0.06	-0.07
17	1	-0.05	-0.12	-0.02
18	1	0.02	-0.02	-0.03
19	1	0.00	0.05	0.02
20	1	-0.11	-0.03	0.02
21	1	-0.10	-0.15	-0.01
22	11	0.00	0.01	0.01

Fig. S22. Intrinsic reaction coordinate (IRC) calculations of Route 1 via *TS1*, and Route 1 via *TS2*, *C1*, *TS3* and *C2* for the formation of product ion *P4* (two configurations including *P4a*, *P4b*) m/z 147 at the B3LYP-gcp-D3/6-31+G(d) level. a IRC calculations of *TS1* for the formation of product ion *P4a* at m/z 147. b IRC calculations of *TS2* for the formation of product ion *P4b* at m/z 147.

Fig. 3 | Results of stationary point and AIMD calculations that support CCE process. a Relative energy levels for Route 1 for the generation of *P4a* ion via *TS1* through the demethylation C–C coupling reaction. **b** Snapshots of one reactive trajectory via *TS1*. (Note: Each panel depicts the geometry at the specified reaction time (fs). The internuclear distances (Å) between the two C atoms involved in bond formation, and between the two C atoms in detaching CH_3^+ and *P4a* are indicated by red and blue fonts, respectively.)

Fig. 4 | Potential energy of demethylation C–C coupling reaction using stationary point and AIMD calculations. a Potential energy as a function of the distance between the two positive charges on the carbonyl carbon atoms, with other degrees of freedom fixed at the equilibrium geometry, where d_1 is the effective radius of the reaction cross-section of the two **RI** cations, and d_2 is C–C bond length. **b** Interatomic distances and potential energy as a function of reaction time for one typical trajectory in Fig. 3b. (Note: Green curve represents the trend of the internuclear distance (Å) between the two carbonyl carbon atoms of sodiated acetone and sodiated acetone with bond forming. Blue curve represents the trend of the internuclear distance (Å) between the carbonyl carbon and methyl carbon in detaching CH_3^+ and **P4**. Red curve represents the reaction potential energy. The variation fluctuations in 0-200 fs and 300-400 fs are caused by vibrations of molecules, while the drastic changes in 200-300 fs are due to the occurring of chemical reaction.)

⇒ In addition to Route 1 via TS1, we did find another pathway for C–C coupling via TS2 (Fig. S20 see below) with multiple transition states and intermediates. This pathway proceeds with the initial C–C coupling, and then the rotation of CH_3 group and finally the detachment of CH_3^+ . The energy barrier of TS2 is about 7.2 eV. The IRC plot for this pathway is shown in Fig. S22b (see above). In a response, the content of “In addition to the route 1 through **TS1**, an alternative pathway for C–C coupling through **TS2** (Supplementary Fig. 20) involving multiple transition states and intermediates was also identified. This pathway involves the initial coupling of C–C, followed by the rotation of the CH_3 group, and ultimately the detachment of CH_3^+ . The energy barrier for **TS2** is approximately 7.1 eV.” has been added in the revised manuscript. (see Page 14, Lines 269-272)

Fig. S20. The stationary points and energies on the minimum energy path for additional pathway that support the CCE process. Relative energy levels to generate *P4b* ion via *TS2* by the demethylation C–C coupling reaction. Purple Na, red O, gray C, and white H. *R1*: *m/z* 81, *P4b*: *m/z* 147.

5. Essentially, I am convinced that the authors could not find a transition state between the two carbon-based radicals until they forced the formation of the C-C bond (which is easily done as DFT calculations are driven by finding the closest minimum in the potential energy surface, especially the C-C bond was shortened to suggest that it was close to a minimum), thus allowing for the methenium cation to be liberated. As a result, I do not believe that the DFT calculations presented showcase the correct comparison of two potential mechanisms as shown in Scheme 3.

⇒ We thank the reviewer for pointing out these problems. The TS structures were derived through relaxed potential energy surface (PES) scans employing the density functional theory (DFT) method to generate initial configurations for the transition states (TSs). In this process, a bond length was systematically varied to identify a first-order saddle point, followed by the optimization of the corresponding transition state. Subsequently, the relevant TS structures were explored and refined using either TS or QST2 procedures. The QST2 approach employs a quadratic synchronous transit methodology to approach the quadratic region of the TS, followed by the application of a quasi-Newton algorithm to complete the optimization. Verification of all transition states was ensured by confirming the presence of a single imaginary vibrational

frequency and assessing the reasonableness of the vibrational mode. In addition, IRC calculations were performed on each transition state at the same level of theory. This step further corroborated that the optimized TS structures were genuinely connected to the correct reactants and products through a steepest descent path.

⇒ The TS1 directly optimized by the DFT method gives only one imaginary frequency of about -261.64 cm^{-1} , and 59 real frequencies. The number of frequencies is consistent with that of a direct reaction process. We also checked the output eigenvector of this mode with imaginary mode, as shown in the imaginary frequency mode section of SI (see Response 4 above), which is the correct direction associated with one C–C bond stretching and the other C–C bond contracting. The IRC plot shown in Fig. S22a (see Response 4 above) for Route 1 also supports that the reactants are connected with products via TS1. Therefore, DFT calculations presented showcase the correct comparison of two potential mechanisms as shown in Scheme 3.

6. The way the AIMD simulations were performed was to introduce the bias by starting with the TS1 structure and enforcing the C-C bond formation. I am also not convinced that such hugely positive reaction barrier of over 770 kJ/mol could be overcome unless the methenium cation was encouraged to fragment during the simulation. As the authors pointed out that to imitate the true collision such simulations would be computationally infeasible, but the ones presented can hardly be considered definitive either. Without any input of energy (via significantly increased temperature and inclusion of additional potential energy as regularly done in metadynamics) it is very hard to believe that AIMD simulations could indeed form the C-C bond out of two unoccupied 2p orbitals on the carbon cationic centers! The electrons must come from somewhere and the only convincing argument here is that the CH_3^+ cation is liberated leaving the carbene on one of the cationic centers and this is when the C-C bond can be formed!

⇒ We thank the reviewer for pointing out these problems. The AIMD simulations were run with additional collision energy and rovibrational energies of reactants. This is particularly true because the barrier (TS1) is very high (7.2 eV) and the reaction can only occur with sufficient energies. For any AIMD simulations for a bimolecular reaction, at least the collision energy and zero-point energy (ZPEs) of reactants should be included, and additional rovibrational energies can also be added in terms of efficiency. For a product-like transition state (TS1) of this reaction, which is also called a late-barrier reaction, the initial vibrational excitation should be efficient in enhancing the reactivity. The reason we added additional rovibrational energy is that the AIMD simulations are computationally expensive, in particular due to strong long-range interaction between the two cation reactants. As indicated in Fig. 4b of the time-dependent evolution diagram, the reaction occurs with the simultaneous breakage of one C–C bond and formation of another C–C bond. Therefore, the CH₃⁺ cation does not leave until the C–C bond is established. In response, the content of “AIMD simulations were conducted with supplementary CE and rovibrational energies of the reactants. This is especially accurate due to the high barrier (*TSI*) of 7.2 eV, which necessitates sufficient energies for the reaction to take place. In AIMD simulations of bimolecular reactions, it is essential to incorporate CE and ZPEs of the reactants. Furthermore, for improved efficiency, supplementary rovibrational energies may also be integrated. For a product-like transition state (*TSI*) in this reaction, also known as a late-barrier reaction, it is essential for the initial vibrational excitation to effectively enhance reactivity.” has been added in the revised manuscript. (see Pages 29-30, Lines 612-618)

⇒ The AIMD results supported that the reaction R1+R1→P4 + CH₃⁺ mainly occurs when R1 should first be highly excited in the internal degrees of freedom, due to the efficient energy transfer from the translational energy to the internal excitations of reactants. This is consistent with the experimental

procedure, in which the total energy provided by the electric field for the gas phase collisional activation of R1 was no less than ~5 eV (CE = 22% in the ion trap). Thus, the collision energy (relative translational energy) for the R1 counterparts is large even at a large distance. When the two reactants (cations) get close, the RCF between the two sodium acetones R1 increases substantially (as discussed in Fig. 4), during which the reactants can be first translationally cooled down by the large RCF, *i.e.*, rapid deceleration of the two cations, and remarkable energies are quickly transferred to the excitation of CH₃ group. In this process, as favored by electric field activation and RCF promotion, the formation of P4 from the collision of sodium acetones R1 is energetically enabled.

7. I fail to see the impact of this study in general. The cleavage of the C-C bond can be achieved using a very high voltage and this does not mean that such cleavage represents a practical application. The DFT barriers are way to high (the barriers under 120 kJ/mol are considered acceptable experimentally) to see how the mechanisms presented in Scheme 3 could be achieved under standard laboratory conditions. 9kV of voltage is enough to power a single household! This is an incredibly high amount of energy required to make the proposed carbon-carbon elongation viable.

⇒ We thank the reviewer for pointing out these problems. We are sorry that our original work is not understood and satisfied by the reviewer. In the revised manuscript, we have made a detail correction in the experimental and instrumental the revised manuscript. It reads “As depicted in Fig. 1a, the self-constructed experimental setup comprised four primary components: (1) capillary spray array, (2) parallel-plate electric field, (3) reaction vessel, and (4) electric field time-control system. The fundamental principle involves that generation of charged cations at one end through a capillary electrospray array, while at the other end, charged cations are also formed through a parallel electric field, enabling the migration of the cations to the surface of the solution. The cations generated at the electrospray array interact with the solution’s surface, leading to the formation of

a demethylation C–C coupling product. This product can subsequently capture H radicals or other anions in the solution, resulting in the formation of stable CCE-related products. Utilizing the experimental apparatus depicted in Fig. 1a, positively charged species, such as cations **RI**, were produced through electro spray array and subsequently reacted with the exposed cations **RI** generated at the liquid-air interface. In detail, using sodiated acetone as an example, the capillary electro spray array, involved inserting all the capillaries (ID 0.19 mm) into a solution of sodiated acetone at one end. The sodiated acetone solution was then charged with high voltage (1-6 kV) via an electrode to generate an array of charged droplets **RI** as depicted in Fig. 1a. The distance from the capillary electro spray array end to the surface of the solution in the reaction vessel was approximately 5 mm. In order to induce the generation of cations **RI** in the reaction vessel, a parallel-plate electric field was constructed. This involved securing a conductive silicon wafer at the base of the reaction vessel and connecting it to a high-voltage power supply, thereby applying a voltage of 1-2 kV. A copper electrode was immersed in the solution and linked to Ground 1 to dissipate the static electricity produced by CCE reaction. The electric field time-control system was employed for the regulation of the high voltage setting of various components such as Al-plate electrode, spray, coated silicon, and copper electrode. (Fig. 1a). In this study, the voltage values were consistent and constant in the spray and Al-plate electrodes, whereas the voltage in the coated silicon exhibited a pulsed mode. The constant voltage for the spray aims to produce an array of **RI** cations, whereas the constant voltage for Al-plate is intended to create parallel electric fields with the copper electrode connected to Ground 2. The purpose of the parallel-plate electric field between the coated silicon and the copper electrode connected to Ground 2 was to produce **RI** cations. These cations then moved to the surface of the solution and subsequently interacted with **RI** cations generated from the spray. The pulsed voltage applied to the coated silicon was utilized for discharging the static electricity produced from CCE reaction involving two cations. To achieve the

appropriate magnitudes of electric fields, the research meticulously regulated the distances between Al-plate electrode, coated silicon, and copper electrode. For instance, the voltage for the spray and Al-plate electrode were set at 6 kV, whereas the voltage for the coated silicon was set at 2 kV. The distance between the Al-plate electrode and the coated silicon is twice that between the coated silicon and the copper electrode. Following the operation of the electric field time-control system, **RI** cations from capillary produce by the capillary electrospray array reacted with **RI** cations present at the liquid-air interface, which were generated from the reaction vessel through a parallel-plate electric field, resulting in the formation of CCE products. The formed CCE products underwent additional analysis using MS and other techniques. **MS for detection CCE products in liquid phase.** All MS experiments related to CCE products formed in the liquid phase were performed using a linear ion trap (LTQ)-XL advantage ion-trap mass spectrometer equipped with a custom-made electrospray ionization (ESI) interface (Thermo Fisher Scientific Inc., San Jose, USA). LTQ mass analyzer was utilized in the positive-ion detection mode, and mass spectra were acquired within the m/z range of 15-400. The formation of $[M + H]^+$ or $[M + Na]^+$ ions was achieved through ESI under the specified typical conditions, which were optimized by adjusting various experimental parameters such as capillary temperature, and tube lens voltage. LTQ mass analyzer was additionally utilized in the negative-ion detection mode, and mass spectra were acquired within the m/z range of 15-200. The formation of $(M + CH_3COO^-)$ ions was achieved using the classical ESI-MS method. The optimized conditions were established as follows: the spray voltage was maintained at 3-6 kV, nitrogen gas (purity > 99.999%) was supplied at a pressure of 0.4 MPa, the capillary temperature was set to 320 °C, the capillary voltage was adjusted to 30 V, the tube lens voltage was set at 20 V, and sample solutions were delivered into the ESI source via a syringe at a flow rate of 5 $\mu\text{L min}^{-1}$. The angle formed between the electrospray beam and the MS inlet of LTQ-MS instrument was 150°. MS/MS conditions were as follows: the ion trap pressure was maintained at 1×10^{-5} torr,

CE was ranged from 20-30%, and the isolation width was set at 2 Da. The system automatically optimized the other LTQ-XL parameters.”. (see Pages 22-24, Lines 431-489)

⇒ The reviewer is correct that the energy barriers under 120 kJ/mol are considered experimentally acceptable for a reaction at room temperature without other energy input. The only drawback as the reviewer pointed out is that the energy barrier required for the reaction in our work is very high. However, this high energy is actually reasonable, because it needs to overcome the coulomb repulsion between the two sodiated acetone cations and further induce the C–C bond dissociation and the other C–C bond formation. In our reaction system, there was sufficient energy source to facilitate the reaction. For example, in the gas phase, sodiated acetone cations could obtain the collision energy via CID in the ion trap. It was calculated that the relative collision energy of the two sodiated acetone cations could reach ~ 7.3 eV (Fig. 2b, Supplementary Fig. 14 see below), which was sufficient to promote the formation of CCE product. In the liquid phase, a high electric field ($\sim 10^9$ V/m) at the air-water interface of microdroplets could provide sufficient energy and was responsible for the reaction acceleration (~ 10000 times), which promoted the formation of CCE product. In addition, we tried to find other possible TS and CCE product formation path, which also have high energies (see Response 4 above).

⇒ In the electrospray process of Fig. 1a, although several kV high voltage is applied, the current generated in this process is very small (\sim several μ A), thus the power consumed is very low (\sim several mW, namely \sim several mJ/s). Therefore, a voltage of several kV is not equal to several kJ/mol of energy.

Fig. S14. The CID-MS spectra of sodiated acetone m/z 81 with different collision energies. a Collision energy with 0%. **b** Collision energy with 10%. **c** Collision energy with 20%. **d** Collision energy with 22%. **e** Collision energy with 25%.

Reviewer #3 (Remarks to the Author):

Comments to the Author

The authors in this paper present a study of carbon chain elongation reactions made feasible by the application of an electric field. They have devised a reaction set-up where they show the possible formation of a CCE product from two cationic reactants which are generated by the application of a positive potential to an electrospray and to a liquid-air interface. Upon impact, the repulsive Coulombic force leads to the dissociation of one of the cations and leads to the formation of C-C bond. The evidence of this reaction happening is shown through mass spectrometric studies and the mechanism is corroborated by theoretical calculations.

While the overall results of the paper are interesting, my biggest concern with the paper is that to achieve these results, they apply very high potentials to the system. Most

organic molecules will be ionized at this potential and this could reduce the overall applicability of the results. Additionally, the authors rely completely on mass spectrometric data analysis to prove that they are indeed observing a CCE product. Moreover, Zare, Cooks and others have recently shown many chemical reactions which are made feasible/accelerated in charged droplets, but their works have not been referred to. (eg: 10.1002/anie.201507805)

While the results of the paper are interesting and could be of potential use to develop more CCE chemistry, I think that there are several questions which the authors must address before this can be considered for publication in nature comm.

⇒ **We thank the reviewer for the insightful suggestions and comments. In ESI source, an electrospray plume is produced by applying a potential difference of 3-6 kV, and the electrospray can be considered as an electrochemical cell (Annu. Rev. Anal. Chem. 2010, 3, 231-254; Anal. Chem. 2006, 78, 24, 8549–8555). We have made the appropriate modifications to use a common electrospray voltage (+1-6 kV) according to the suggestions. These potentials can ensure the more acetone molecules to form charged cations while not side products of electrochemical redox reaction. In a response, the content of “Furthermore, CCE product, corresponding fragments, ions associated with electrochemical redox reactions, and C–C bond cleavage reactions were not detected during the high-voltage spraying of sodiated acetone solution or acetone aqueous solution (Supplementary Fig. 12).” has been added in the revised manuscript. (see Page 10, Lines 187-189)**

Fig. S12. ESI-MS spectrum of sample solution at the voltage of 6 kV. a Sodiated acetone aqueous solution. **b** Acetone aqueous solution.

⇒ In addition to MS data, we have also performed NMR, UV techniques to characterize the CCE product (details see below).

⇒ The corresponding works in charged droplets reported by Zare, Cooks and others has been cited in the revised manuscript (see Ref 15-22). In a response, the content of “In recent years, microdroplet chemistry has emerged as an exciting field due to its ability to accelerate chemical reactions by up to 6 orders of magnitude compared to the same reactions in bulk solution¹⁵⁻²⁰. The ultrahigh electric fields (approximately 10^9 V/m) are primarily attributed to the acceleration of the reaction^{21,22}. Under the influence of an electric field, the collisions between ions and molecules/ions can serve as effective mechanisms for initiating to induce chemical reactions. This phenomenon arises from the transfer of energy, momentum, and charge during the collision. This transfer enables the system to surmount existing reaction barriers or generate highly reactive species that can promptly interact with surrounding cluster constituents²³.” has been added in the

revised manuscript. (see Pages 3-4, Lines 56-64)

Major points:

1. The electrochemical set up is not very well described. It is not clear why spray has pulsed voltage while the plates have constant voltage. And how this configuration allows the authors to achieve the right magnitudes of electric fields. Moreover, it will be good to provide a value of the electric field generated at the interface.

⇒ We thank the reviewer for pointing out these problems. We apologize for our carelessness. Spray and Al-plate both have the same and constant voltage, while coated silicon has pulsed voltage. The purpose of constant voltage for spray is to generate R1 ion array, while constant voltage for Al-plate is to form parallel electric fields with Ground 2 at the bottom of the reactor. The purpose of pulsed voltage for coated silicon is to generate another R1 to collide with R1 formed by spraying and release static electricity. Fig. 1 (see below) has been changed in the revised manuscript. In a response, the content of “As depicted in Fig. 1a, the self-constructed experimental setup comprised four primary components: (1) capillary spray array, (2) parallel-plate electric field, (3) reaction vessel, and (4) electric field time-control system. The fundamental principle involves that generation of charged cations at one end through a capillary electro spray array, while at the other end, charged cations are also formed through a parallel electric field, enabling the migration of the cations to the surface of the solution. The cations generated at the electro spray array interact with the solution’s surface, leading to the formation of a demethylation C–C coupling product. This product can subsequently capture H radicals or other anions in the solution, resulting in the formation of stable CCE-related products. Utilizing the experimental apparatus depicted in Fig. 1a, positively charged species, such as cations *RI*, were produced through electro spray array and subsequently reacted with the exposed cations *RI* generated at the liquid-air interface. In detail, using sodiated acetone as an example, the capillary electro spray array, involved inserting all the capillaries (ID 0.19 mm) into a solution of sodiated acetone at one end. The

sodiated acetone solution was then charged with high voltage (1-6 kV) via an electrode to generate an array of charged droplets **RI** as depicted in Fig. 1a. The distance from the capillary electrospray array end to the surface of the solution in the reaction vessel was approximately 5 mm. In order to induce the generation of cations **RI** in the reaction vessel, a parallel-plate electric field was constructed. This involved securing a conductive silicon wafer at the base of the reaction vessel and connecting it to a high-voltage power supply, thereby applying a voltage of 1-2 kV. A copper electrode was immersed in the solution and linked to Ground 1 to dissipate the static electricity produced by CCE reaction. The electric field time-control system was employed for the regulation of the high voltage setting of various components such as Al-plate electrode, spray, coated silicon, and copper electrode. (Fig. 1a). In this study, the voltage values were consistent and constant in the spray and Al-plate electrodes, whereas the voltage in the coated silicon exhibited a pulsed mode. The constant voltage for the spray aims to produce an array of **RI** cations, whereas the constant voltage for Al-plate is intended to create parallel electric fields with the copper electrode connected to Ground 2. The purpose of the parallel-plate electric field between the coated silicon and the copper electrode connected to Ground 2 was to produce **RI** cations. These cations then moved to the surface of the solution and subsequently interacted with **RI** cations generated from the spray. The pulsed voltage applied to the coated silicon was utilized for discharging the static electricity produced from CCE reaction involving two cations. To achieve the appropriate magnitudes of electric fields, the research meticulously regulated the distances between Al-plate electrode, coated silicon, and copper electrode. For instance, the voltage for the spray and Al-plate electrode were set at 6 kV, whereas the voltage for the coated silicon was set at 2 kV. The distance between the Al-plate electrode and the coated silicon is twice that between the coated silicon and the copper electrode. Following the operation of the electric field time-control system, **RI** cations from capillary produce by the capillary electrospray array reacted with **RI** cations present at the liquid-air interface, which

were generated from the reaction vessel through a parallel-plate electric field, resulting in the formation of CCE products. The formed CCE products underwent additional analysis using MS and other techniques.” has been added in the revised manuscript. (see Pages 22-23, Lines 431-471)

⇒ To obtain the right magnitudes of electric fields, we precisely controlled the distances of Al-plate electrode, coated silicon, and copper electrode. For example, when the voltage of spray and Al-plate electrode were settled as 6 kV, while the voltage of coated silicon was settled as 2 kV. The distance between Al-plate electrode and coated silicon is twice of the distance between coated silicon and copper electrode. Due to many references (Proc. Natl. Acad. Sci. U. S. A. 2019, 116, 19294–19298) reported that the ultrahigh electric field ($\sim 10^9$ V/m) generated on air-water interface of a microdroplet. Thus, we referred it directly in the revised manuscript. In a response, the content of “To achieve the appropriate magnitudes of electric fields, the research meticulously regulated the distances between Al-plate electrode, coated silicon, and copper electrode. For instance, the voltage for the spray and Al-plate electrode were set at 6 kV, whereas the voltage for the coated silicon was set at 2 kV. The distance between the Al-plate electrode and the coated silicon is twice that between the coated silicon and the copper electrode.” (see Pages 23, Lines 462-466) and “The electric field intensity can be attained at approximately 10^9 V/m on gas-liquid interface of the cations reaction region ²⁴.” (see Page 6, Lines 100-102) has been added in the revised manuscript.

Fig. 1 | Schematic illustration of the experimental setup and the possible process of CCE involving two organic cations. **a** Experimental setup for CCE reaction between two *RI* cations. **b** Mass spectrum of CCE products formed between two sodiated acetone cations. **c** Process of two sodiated acetone cations approached to each other. (Note: RA: Relative abundance; initial stage occurs when $d_0 > d_1$; repelling and coupling stage occurs when $d_2 < d_0 < d_1$), where d_0 is the distance between the two positive charges located on the carbonyl carbon atoms of the two *RI* cations, d_1 is the effective radius of the reaction cross-section of the two *RI* cations, and d_2 is C–C bond length.

2. It seems to be that there are two set of experiments conducted in this article. The one with the electrochemical setup and another with mass spec using electrospray ionization. Why are the results from the cell not discussed? What are yields of the CCE product from the cell? What other techniques are used to verify the formation of the final product? If no other techniques were used, I suggest that the authors use NMR, UV-Vis or FTIR techniques to show the final product formation as well as report the yields of the product from the experimental setup.

⇒ We thank the reviewer for pointing out these problems. There are two series of experiments in this work, the one is gas-phase ion chemistry in the ion trap,

the other is liquid phase ion chemistry in the cell controlled by electric field. In a response, we have added corresponding discussion about the cell reaction, yields of CCE reaction, the characterization of CCE product by NMR and UV-Vis spectroscopy techniques (see Figs. S7, S8 below). It reads “The yields of CCE products in the liquid phase ranged from <1% to approximately 70%, were calculated based on the MS signal ratios of the product to the reactant. While this approach is frequently employed to estimate reaction yields in MS-based micro-synthetic studies ^{17,25,26} due to its simplicity and effectiveness in assessing microscale products, it is susceptible to significantly bias in actual conversion rates. This is attributed to variations in the ionization efficiencies of the reactants and products. In addition to MS techniques, ultraviolet-visible (UV-Vis) and nuclear magnetic resonance (NMR) spectroscopy were also employed to further characterize CCE product. CCE product formed by two protonated acetone cations and the 2-methylbutane-2,3-diol standard exhibited similar UV-Vis spectra, both of which resulted in the disappearance of the characteristic peak of KMnO₄ (Supplementary Fig. 7). The characteristic peaks observed in ¹H NMR spectrum of CCE product closely corresponded to the peaks in ¹H NMR spectrum of the 2-methylbutane-2,3-diol standard (Supplementary Fig. 8). The characterization results suggest that CCE products can be produced in the liquid phase by employing the apparatus depicted in Fig. 1a.” (see Pages 8-9, Lines 156-169) and “CCE product and corresponding diols standard were analyzed using UV-Vis absorption spectroscopy (UV-1900i, Shimadzu, Japan). Potassium permanganate was reported to undergo reduction by diols in an acidic environment, with the maximum absorption peak at 525 nm ⁴³ (Supplementary Fig. 7). Following the reaction with diols, the peak at 525 nm exhibited a decrease or even disappearance. (see Page 26, Lines 523-527)

Fig. S7. Characterization of CCE product product by UV-Vis spectroscopy.
Control: KMnO_4 ; Standard: 2-methylbutane-2,3-diol.

Fig. S8. ^1H NMR (500 MHz, D_2O) spectrum of 2-methylbutane-2,3-diol and CCE product of protonated acetone ion.

3. The mass spec experiment for the CCE product formation is essentially a gas phase reaction. Therefore, how these fare in the solution phase in the actual experimental setup will be a good comparison.

⇒ We thank the reviewer for pointing out these problems. We have added related content about the results of CCE product formed in liquid phase using the device in Fig. 1a (an offline scale-up device) and compared them with the results of gas phase CCE reaction. It reads “Upon the collisions of *RI* cations from spray plumes and the liquid-air interface (as shown in a partial enlargement of Fig. 1a), the mass spectrum indicated the generation of products related to demethylation and C–C coupling during CCE reaction between two *RI* cations. In Fig. 1b, the products associated with C–C coupling observed at m/z 149, m/z 167 and m/z 183 corresponded to $(MH+H)^+$, $(MF+H)^+$, and $(MCl+H)^+$, respectively. This correspondence is attributed to the addition of H radical, F^- , and Cl^- anions in the solution within the reaction vessel. The process of formation is depicted in Scheme 2. The small CH_3^+ cation, which is produced during CCE process, can be effectively captured by anions such as F^- , Cl^- , Br^- and CH_3COO^- to yield CH_3F , CH_3Cl , CH_3Br and CH_3COOCH_3 , respectively, in the reaction vessel’s solution (Scheme 2). The product signals at m/z 93, m/z 109, m/z 153 and m/z 133, which result from the formation of (CH_3X+Ac^-) adducts, were also identifiable in the negative ion detection mode (Supplementary Fig. 1a). These findings were subsequently validated through collision induced dissociation mass spectrometry (CID-MS) analysis (Supplementary Fig. 1b-1e).” (see Page 7, Lines 115-128), “The yields of CCE products in the liquid phase ranged from <1% to approximately 70%, were calculated based on the MS signal ratios of the product to the reactant. While this approach is frequently employed to estimate reaction yields in MS-based micro-synthetic studies^{17,25,26} due to its simplicity and effectiveness in assessing microscale products, it is susceptible to significantly bias in actual conversion rates. This is attributed to variations in the ionization efficiencies of the reactants and products. In addition to MS techniques, ultraviolet-visible (UV-Vis) and nuclear magnetic resonance (NMR) spectroscopy were also employed to further characterize CCE product. CCE product formed by two protonated acetone cations and the 2-methylbutane-2,3-diol standard exhibited similar UV-Vis

spectra, both of which resulted in the disappearance of the characteristic peak of KMnO_4 (Supplementary Fig. 7). The characteristic peaks observed in ^1H NMR spectrum of CCE product closely corresponded to the peaks in ^1H NMR spectrum of the 2-methylbutane-2,3-diol standard (Supplementary Fig. 8). The characterization results suggest that CCE products can be produced in the liquid phase by employing the apparatus depicted in Fig. 1a.” (see Pages 8-9, Lines 156-169), and “The yields of CCE products in both the gas-phase and liquid phase varied from <1% to approximately 70%. The disparity in yields between gas phase and liquid phase could be associated with the stability of CCE products. For instance, the demethylation C–C coupling products generated in the gas-phase existed in cationic form, whereas those produced in the liquid phase was in neutral form due to the capture of H radicals or other anions in the solution.” (see Page 20, Lines 388-393).

4. It is clear from the MS data that the major product of the reaction is an allylic cation species. The authors have given a proposed mechanism for this product in the SI, but it will also be worth mentioning in the main text. It will be good to discuss the origin of this species and why that dominates instead of the CCE product.

⇒ We thank the reviewer for pointing out this suggestion. You are right that the major product of the reaction is an allylic cation. In a response, we have added the discussion on the major product allylic cation in the revised manuscript. The content of “As confirmed by the tandem MS data (Supplementary Scheme 1, Supplementary Fig. 8), the ions of m/z 41 (the allyl cation, **P1** and/or hydrated sodium cation, **P2**), m/z 99 (the hydrated sodiated acetone, **P3**) were irrelevant to the CCE reaction.” has been changed into “The prominent peak observed at m/z 41 (Fig. 2b) corresponds to the dissociation products of **RI** (m/z 81), indicating a combination of the allyl cation (**P1**) and hydrated sodium cation (**P2**) through the elimination of NaOH and propadiene from the dissociation of **RI**, as illustrated in Fig. 2a, Supplementary Scheme 1a, and Supplementary Fig. 15. The proposed structures of the allyl cation and hydrated sodium cation have been validated

through isotope labeling experiments (Fig. 2d, Supplementary Scheme 1b), a kinetic isotope effect of $k_H/k_D = 5$, and theoretical calculations (detailed in the subsequent calculation section). The dominance of the allyl cation (*P1*) and the hydrated sodium cation (*P2*) may be attributed to the stability of the product ions, which are more stable than *P4* carbocation. The detailed information was illustrated in the following calculation results section.” in the revised manuscript. (see Pages 11-12, Lines 201-216)

5. The authors do not discuss what happens if only the electrospray is charged but not the interface. Do they still form CCE products?

⇒ We thank the reviewer for pointing out this question. No CCE products formed when only the electrospray is charged but not the interface. This result was consistent with that of the gas phase ion reaction described in Supplementary Fig. 9 and Fig. 10. We have performed the control experiments (spray+6 kV, interface +0 kV vs. spray+6 kV, interface +2 kV) by using setup in Fig. 1a. It can be found that no CCE products were observed when the spray is charged but not the interface (Table S2, Fig. S9 below). In addition, we also performed another control experimental (spray-1, +3 kV, spray-2, +0 kV vs. spray-1, +3 kV, spray-2, +2 kV) by using two sprays with online MS detection (Fig. S10 below). The result of using two sprays was consistent with that of using the device in Fig. 1a. In a response, the content of “As demonstrated in the experiment (Supplementary Table 2), CCE reactions only occurred between two positively charged species. The control experiments were conducted comparing the effects of spray 6 kV and coated silicon 0 kV with spray 6 kV and coated silicon 2 kV. The control experiments demonstrate that no CCE products (Supplementary Fig. 9a and 9d) were detected when only the electrospray was charged without the coated silicon. However, when both the spray and coated silicon were charged, corresponding CCE products (m/z 149 and m/z 171 for sodiated acetone; m/z 105 for protonated acetone) were observed (Supplementary Fig. 9b and 9e). CCE products have also been validated through the corresponding

tandem MS experiments (Supplementary Fig. 9c and 9f). The reason for this is that when only the spray is charged, the reactant ions formed are immediately grounded and neutralized without undergoing effective collision. It was observed that conducting two sprays (spray1, 3 kV, spray2, 0 kV vs. spray1, 3 kV, spray2, 2 kV) using online MS analysis yielded consistent results (Supplementary Fig. 10) with those obtained using the apparatus in Fig. 1a (Supplementary Fig. 9).” has been added in the revised manuscript. (see Page 9, Lines 170-182)

Fig. S9. Control experiments for CCE products formation by setup in Fig. 1a. **a** The MS spectrum of CCE reaction of two sodiated acetone cations when the spray charged while the interface not charged. **b** The MS spectrum of CCE reaction of two sodiated acetone cations when the spray and interface both charged. **c** The MS/MS spectrum of CCE product at m/z 149. **d** The MS spectrum of CCE reaction of two protonated acetone cations when the spray charged while the interface not charged. **e** The MS spectrum of CCE reaction of two protonated acetone cations when the spray and interface both charged. **f** The MS/MS spectrum of CCE product at m/z 105.

Fig. S10. Control experiments for CCE products formation by online MS. **a** The MS spectrum of CCE reaction of two sodiated acetone cations when the spray 1 charged while the spray 2 not charged. **b** The MS spectrum of CCE reaction of two sodiated acetone cations when the spray 1 and spray 2 both charged. **c** The MS/MS spectrum of CCE product at m/z 147.

6. If two distinctive cations are generated in the spray and at the interface, do the authors see more of homo-dicationic reaction or hetero-dicationic reaction?

⇒ We thank the reviewer for pointing out this question. We have performed the

experiments by using two distinctive cations in the spray and at the interface as suggested, and found that the products of homo-dicationic reaction and hetero-dicationic reaction all can be observed. The intensity of hetero-dicationic reaction products seems higher than that of homo-dicationic reaction. The reason may be a problem of probability. For example, sodium solution with two mixture samples (acetone and pentanone) was introduced to the spray end and the gas-liquid interface, respectively, to undergo a collision reaction using the device in Fig. 1a. The mass spectrum shows both homo-dicationic reaction products at m/z 149, 171, 205, 227, and hetero-dicationic reaction products at m/z 177, 199 (Figure see below). The intensity of hetero-dicationic reaction products was higher than that of homo-dicationic reaction products. **In a response, the content of “It was also observed that in case where two different cations were present in both the spray and the reaction vessel, both homo-dicationic and hetero-dicationic reaction products could be observed (Supplementary Fig. 11). Interestingly, the intensity of the hetero-dicationic reaction products appeared to be higher than that of the homo-dicationic reaction products. The issue may stem from a probability problem.” has been added in the revised manuscript. (see Pages 9-10, Lines 182-186)**

Fig. S11. Mass spectrum of collision reaction when sodiated acetone and sodiated pentanone mixture solution was introduced to the spray and the interface, respectively.

7. In the SI, some reactions are mentioned which have 76% yield, why do the yields of these reactions vary from less than 1% to 76% should be discussed, ie., what factors control these yields?

⇒ We thank the reviewer for pointing out this question. The yields were roughly calculated by the MS signal ratios of the product to the reactant. This strategy is commonly used for estimating reaction yields in MS based micro-synthetic studies (Chem. Sci., 2011, 2, 501–510; PNAS, 2015, 112(13), 3898-3903; PNAS, 2017, 114(47), 12396-12400) because of simplicity and effectiveness to the microscale products. However, it may suffer from small to large bias to actual conversion because of the difference in the ionization efficiencies of the products and reactants. In a response, the corresponding content has been added in the revised manuscript. It reads “The yields of CCE products in the liquid phase ranged from <1% to approximately 70%, were calculated based on the MS signal ratios of the product to the reactant. While this approach is frequently employed to estimate reaction yields in MS-based micro-synthetic studies^{17,25,26} due to its simplicity and effectiveness in assessing microscale products, it is susceptible to significantly bias in actual conversion rates. This is attributed to variations in the ionization efficiencies of the reactants and products.”. (see Pages 8-9, Lines 156-161)

Minor points

1. The writing of the paper needs to be revised. There are grammatical errors throughout and some sentences are too convoluted to understand. For example, in the methods section, authors have forgotten to remove the guidelines in the start.

⇒ We thank the reviewer for pointing out this question. We have made a detail correction on the grammar and unclear sentences throughout the paper, which are highlighted in the revised manuscript. For example, the content of “In

response, the content of “The stationary point calculations at the B3LYP-gCP-D3/6-31+G(d) level reveal that the product ion of m/z 147 through route 1 (the CCE reaction, Scheme 3a, elimination of a methyl cation with a concerted mechanism, the calculation results shown in Fig. 3a) is advantageous over the route 2 (the ion/molecule reaction, Scheme 3b, a two-step mechanism in which a carbene intermediate firstly generated from a sodiated acetone by direct loss of a methyl cation reacts with another sodiated acetone, the calculation results shown in Supplementary Fig. 13). For route 1, the key transitions state of **TS1** for **P4** formation is confirmed by intrinsic reaction coordinate calculations (Supplementary Fig. 14), validating the results shown in Fig. 3a.” **has been changed into** “A further in-depth theoretical examination of the dissociation or coupling of **RI** has been conducted through DFT calculations. As depicted in Scheme 3, there are two potential pathways for the formation of **P4** product (m/z 147) involving two sodiated acetone cations. Route 1 involves CCE reaction through the elimination of a methyl cation with a concerted mechanism (Scheme 3a). Route 2 involves the ion/molecule reaction through a two-step mechanism, in which a carbene intermediate is initially formed from a sodiated acetone by the direct loss of a methyl cation, and then reacts with another sodiated acetone (Scheme 3b). The stationary point calculations conducted at B3LYP-gCP-D3/6-31+G(d) level indicate that the product ion at m/z 147 through Route 1 (Fig. 3a, Supplementary Fig. 20) is more favorable compared to Route 2 (Supplementary Fig. 21). In addition to the route 1 through **TS1**, an alternative pathway for C–C coupling through **TS2** (Supplementary Fig. 20) involving multiple transition states and intermediates was also identified. This pathway involves the initial coupling of C–C, followed by the rotation of the CH₃ group, and ultimately the detachment of CH₃⁺. The energy barrier for **TS2** is approximately 7.1 eV. For Route 1, the key transitions states of **TS1** for **P4a** formation and **TS2** for **P4b** formation were verified through intrinsic reaction coordinate (IRC) calculations (Supplementary Fig. 22), thereby validating the findings presented in Fig. 3a and Supplementary

Fig. 20, respectively. The imaginary vibrational frequency of *TS1*, along with the corresponding eigenvector of this vibrational mode, indicated the correct direction associated with the stretching of one C–C bond and the contraction of the other C–C bond (as shown in the imaginary frequency mode section of Supplementary Material). For Route 2, the energy required for the initial dissociation of a single sodiated acetone and the subsequent ion-molecule reaction between carbene and sodiated acetone is at least 8.9 eV (Supplementary Fig. 21), significantly exceeding that for Route 1. Additionally, the study conducted calculations on the dissociation energies of other compounds with varying R₁ groups, revealing similarly high energy requirements ranging from 8.9 eV to 14 eV (Supplementary Fig. 23) is indicated to be the lowest compared to other R₁ groups. The findings indicate that the initial dissociation of *RI* through Route 2, followed by the ion-molecule reaction, is less likely to occur compared to Route 1, which involves direct demethylation C–C coupling reaction.” in the revised manuscript. (see Pages 13-15, Lines 255-293)

⇒ The guidelines in the methods section have been removed in the revised manuscript.

2. On page 4, it is not clear why the second solution (at the interface) has acetic acid and salts but no acetone. As the whole paper is discussed with sodiated acetone cation reaction with another sodiated cation, it will be better to use that for both cation a and b examples.

⇒ We thank the reviewer for pointing out this problem. The content of “With proper timing manipulation (Fig. 1b), the cation *a* (e.g., protonated/sodiated acetone) generated by electrospraying of an acetone aqueous solution (1:1, v/v) was accelerated by a high voltage (+1-6 kV) to impact cation *b* (e.g., protonated/sodiated acetone, etc.), which was created on the liquid-air interface by applying a positive voltage (+0.001-2 kV) to the solution containing acetic acid (10 ppm) and equivalent sodium salts of fluorine, chlorine, bromine and iodine (10

ppm).” has been changed into “Through precise timing manipulation (Fig. 1a), the cation **RI**, produced by electro spraying a 1:1 (v/v) sodiated acetone solution, was propelled by a high voltage (+1-6 kV) to collide with another **RI** cation, generated at the liquid-air interface by applying a positive voltage (+1-2 kV) to the sodiated acetone solution in the reaction vessel (refer to a partial enlargement of Fig. 1a). The electric field intensity can be attained at approximately 10^9 V/m on gas-liquid interface of the cations reaction region ²⁴. The demethylation C–C coupling product, namely sodium 2-methyl-butane-2,3-diolate-3-ium and CH_3^+ , were generated as a result of CCE reaction involving two **RI** cations (Scheme 1).”

in the revised manuscript. (see Page 6, Lines 96-104)

3. In the methods section, the voltage range mentioned in different than that on page 4.

⇒ We apologize for our carelessness. We have modified the voltage range to be consistent in the revised manuscript.

4. On page 7, scheme 2 is incorrect. To neutralize a cation, we will need an anion, while X- will serve as an anion, H radical will still leave the final molecule as a cation.

⇒ We have recorrected Scheme 2 according to the suggestion. Please see the scheme below.

Scheme 2 | Direct grounding and anion coupling in solvent to obtain the final product of CCE reactions as depicted in Scheme 1.

5. Figure 2 b and c are mislabelled.

⇒ **We have corrected them.**

6. On page 12, last sentence, specify which section of SI is being referred to.

⇒ **We thank the Reviewer for pointing out this problem. The content of “shown in SI” has been changed into “as depicted in the Supplementary video animation.” in the revised manuscript. (see Page 17, Lines 317-318)**

7. Please redo the caption of figure S3 to make it more succinct.

⇒ **We thank the Reviewer for pointing out this problem. The caption of Fig. S3 has been changed into “MS/MS spectra of the products produced via CCE reactions conducted using the experimental setup shown in Fig. 1. Two protonated acetones (a-c). Protonated butanone and protonated butanone (d-f). Protonated 2-pentanone and protonated 2-pentanone (g-i). Protonated 2-hexanone and protonated 2-hexanone (j-l). Protonated acetamide and protonated acetamide (m-r). Protonated acetic acid and protonated acetic acid (s-u). Protonated acetophenone and protonated acetophenone (v-a1). Protonated acetaldehyde and protonated acetaldehyde (b1-d1). Protonated acetone and protonated urea (e1-g1). Protonated acetone and protonated acetic acid (h1-j1). Protonated urea and protonated urea (k1-m1). Protonated benzaldehyde and protonated benzaldehyde (n1-p1).”.**

Reviewers' Comments:

Reviewer #1:

Remarks to the Author:

The manuscript is significantly improved. Gratitude must go to the reviewers - and their excellent and patient reviewing.

Major mistakes are pointed out. Confusions are carefully communicated. Improvements are constructively suggested.

As a result, the authors' have benefited. And by addressing these comments, the manuscript is understandable.

I think it is still a challenge to follow the experimental workflow - and confusion remains. And I have a few final suggestions.

1. The ion trap (LTQ) pressure is quoted as 1×10^{-5} Torr ... and this is wrong. LTQ Trap pressure under normal operations is ca. 2-3 mTorr (see Thermo Scientific Data Sheets/manual).
2. Also, if the ion ion reactions are in the LTQ ion trap - figure s13 shows a diagram of a Paul-type ion trap - not a linear ion trap of the LTQ. So this looks wrong?
3. All reviewers are confused by the workflow. Fig 1 could still be improved to make it clear that there are two workflows in this Figure (right?). So this reaction vessel needs to be explained better - what is the green? And how is this vessel reaction chemically probed? In that Figure the mass spectrum in part b is from the gas-phase experiment NOT from reaction vessel? These are all too easily confused.

I suggest two clear panels for the figure that make it clear (or split into 2 figures).

a. Solution/Interface Reactor chemistry:

b. Gas-phase Ion-Ion Chemistry:

4.

"The formed CCE products underwent additional analysis using MS and other techniques." Offline? And be explicit please - what are the techniques.

Reviewer #3:

The new revised version of the paper is a substantial improvement on the first version with more clearly defined figures. The authors have conducted more experiments to answer questions raised by me and the other reviewers. I have a few more doubts/questions about the work but they are minor:

1. The authors mention that the electric fields present at the air water interface is what is responsible for overcoming the 7.2 eV barrier. However, the value quoted from Zare and coworkers (10^9 V/m) is electric field present in neutral droplets. However, it is clear from there experiments that unless both the electrosprays/ or the electrospray and surface is charged there is no CCE product observed. Therefore, I disagree with the authors claim that the voltage they are supplying to the electrospray plume and surface is only useful for generating sodiated cations and does not energetically play a role in formation of CCE products. Could the author calculate the amount of electric fields they are generating at the surface, and report that?
2. The UV-Vis spectra shown by the authors is misleading, as it does not go below 400 nm, and what we see is essentially a tail of the spectra. A full spectra from 200-800 nm should be presented.
3. My initial question about hetero vs homo dicationic CCE product was to assert whether both the surface and the droplet have to have the sodiated acetone cation R1 for the reaction, or whether having the reactant in only one medium is sufficient. However, the authors mixed up different cations on both the surface and the plume, so I don't know what the heterodication product being in excess shows in their experiments. I would suggest that if they were to add only sodiated acetone in the plume and sodiated pentanone in the surface and then check what reactions they observed that will help understand the role of the surface beyond being charged.

Reviewer #4:

Remarks to the Author:

The authors assert that Route 1 involves CCE reaction through the elimination of a methyl cation with a concerted mechanism. Nonetheless, the transition state (TS1) as deduced from calculations does not substantiate this claim. All computational findings indicate that TS1 aligns with the dissociation of a methyl cation, with the formation of a C-C bond evident in TS1, characterized by a bond length of 1.58 Å. This contrasts the typical C-C bond length range of 1.8 Å to 2.2 Å observed in transition states for C-C bond formation. The authors' contention that "The TS1 directly optimized by the DFT method gives only one imaginary frequency of about -261.64 cm⁻¹, and 59 real frequencies" merely confirms the transitional nature of TS1 but falls short of validating it as the anticipated concerted transition state. Moreover, the authors' rebuttal stating, "We also checked the output eigenvector of this mode with imaginary mode, as shown in the imaginary frequency mode section of SI (see Response 4 above), which is the correct direction associated with one C-C bond stretching and the other C-C bond contracting," appears inadequate. The imaginary frequency mode distinctly indicates its association with the stretching of one C-C bond, facilitating the elimination of a methyl cation, yet it does not exhibit any vibrational mode indicative of the formation of the anticipated C-C bond. The authors are urged to unambiguously demonstrate the precursor intermediate in the IRC calculation for the TS1 structure and endeavor to identify the transition state for C-C bond formation.

Regarding the AIMD calculations, the study's reliance on a singular reactive trajectory is both insufficient and unconvincing.

The insights provided by Reviewer 2 are particularly pertinent. Despite the authors' extensive response letter, they have not effectively tackled the raised concerns. Furthermore, the manuscript's complexity makes it challenging for readers to grasp, necessitating substantial revisions for clarity and coherence.

Dear reviewers,

We thank you for encouraging and constructive comments. These comments are addressed point-by-point below.

For your convenience, we have listed all the changes made in the revised version on the following pages. Please note that all page numbers, line numbers in this list refer to the *revised manuscript*. We hope that you now find this manuscript acceptable for publication in *Nature Communications*.

Changes made in manuscript in response to referees' comments and questions (All page numbers, line numbers, etc. refer to the *revised version* of the manuscript)

REVIEWER COMMENTS

Reviewer #1 (Remarks to the Author):

Comments to the Author

The manuscript is significantly improved. Gratitude must go to the reviewers - and their excellent and patient reviewing. Major mistakes are pointed out. Confusions are carefully communicated. Improvements are constructively suggested. As a result, the authors' have benefited. And by addressing these comments, the manuscript is understandable. I think it is still a challenge to follow the experimental workflow - and confusion remains. And I have a few final suggestions.

⇒ We thank all the reviewers for their excellent and patient reviewing from the prior round, which greatly improved the quality of the manuscript. We also thank the reviewer for pointing out the new suggestions again. We have done

a detailed revision of the manuscript according to the suggestions.

1. The ion trap (LTQ) pressure is quoted as 1×10^{-5} Torr ... and this is wrong. LTQ Trap pressure under normal operations is ca. 2-3 mTorr (see Thermo Scientific Data Sheets/manual).

⇒ We thank the reviewer for pointing out the question. We have confirmed the working pressure in the vacuum manifold of the ion trap. It was found that the pressure is maintained at about 1×10^{-5} Torr. Please see the following picture. The sentence of “MS/MS conditions were as follows: the ion trap pressure was maintained at 1×10^{-5} torr” has been changed into “MS/MS conditions were as follows: the ion gauge reading in the instrument software was maintained at 1×10^{-5} torr” in the revised manuscript. (see Page 25, Lines 487)

✓ Vacuum OK:	Yes
✓ Ion Gauge Pressure OK:	Yes
✓ Ion Gauge:	On
✓ Ion Gauge (E-5 Torr):	0.99
✓ Convection Pressure OK:	Yes
✓ Convection Gauge (Torr):	1.09
Turbo Pump	
Status:	Running
Life (hours):	98420
Speed (Hz):	799
Power (Watts):	87
Temperature (°C):	57

2. Also, if the ion ion reactions are in the LTQ ion trap - figure s13 shows a diagram of a Paul-type ion trap - not a linear ion trap of the LTQ. So this looks wrong?

⇒ We thank the reviewer for the careful suggestion. The reviewer is correct. The current figure S13 is not a linear ion trap of the LTQ. We have redrawn the figure S13. Please see below.

Fig. S13. Schematic picture of the process of CCE reaction occurring inside an ion trap. **a** The schematic illustration about the process of ion trap for gas phase CCE product formation. **b** Process of multistage tandem mass spectrometry experiments in the ion trap.

3. All reviewers are confused by the workflow. Fig 1 could still be improved to make it clear that there are two workflows in this Figure (right?). So this reaction vessel needs

to be explained better - what is the green? And how is this vessel reaction chemically probed? In that Figure the mass spectrum in part b is from the gas-phase experiment NOT from reaction vessel? These are all too easily confused.

I suggest two clear panels for the figure that make it clear (or split into 2 figures).

a. Solution/Interface Reactor chemistry:

b. Gas-phase Ion-Ion Chemistry:

⇒ We thank the reviewer for the good suggestions. The Figure 1 (see below) and the corresponding context in the text as well has been thoroughly revised by taking all the comments into consideration. The major revisions include the following items:

- ◆ The Figure 1 (including caption) and the text as well have been revised for better readability;
- ◆ The liquid-phase work flow has been separated from the gas-phase one;
- ◆ The key data on the isotope-labeling experiment have been added as Fig. 1d;
- ◆ The original Figure 1d has been combined into Scheme 1 (see below).

⇒ Therefore, we believe we have a much stronger version of the manuscript now. We are waiting for your positive feedback and will be happy to further improved it when necessary. In a response, the content of “While the whole apparatus was electronically grounded, the *RIa* ions (shown as the charged red-brown plume) produced by positive electrospray ionization (ESI, +6 kV) using the spray array were accelerated by +6 kV applied on the Al-plate to impact the *RIb* ions (shown as the small gray balls marked with + sign), which were created on the solution-air interface of acetone solution containing acetic acid (10 ppm) and equivalent sodium salts of fluorine, chlorine (10 ppm) by using the coated Si-plate with +2 kV immersed in the acetone solution (shown as the blue green inside the vessel).” has been added in the revised manuscript. (see Page 6, Lines 96-103)

⇒ Figure 1 (see below) shows the demethylation C–C coupling reaction in the solution/interface reactor chemistry, that is, the reaction occurs between the cations from the spray plume and the cations from the liquid-air interface, while not the reaction occurs in the ion trap. The cations formed by the spray reacted with the cation formed in the reaction vessel at the interface for a period of time, and then the reaction solution was taken out from the vessel for further offline ESI-MS, UV-Vis and NMR analysis. The mass spectra in Fig. 1c, 1d were from the reaction vessel not from the gas-phase experiment. The gas-phase ion-ion chemistry experiments were performed in the ion trap (Fig. 2), while the solution/interface reactor chemistry experiments were performed in the interface of the vessel (Fig. 1a). In a response, the content of “The formed CCE products underwent additional analysis using MS and other techniques.” has been changed into “The formed CCE products underwent additional analysis using offline MS, UV-Vis and NMR techniques.” in the revised manuscript. (see Page 24, Lines 470-471)

Fig. 1 | The liquid-phase CCE reaction occurred between two organic cations *R1a*, *R1b*. **a** Schematic illustration of the experimental apparatus. While the whole apparatus was electronically grounded, the *R1a* ions (shown as the charged red-brown plume) produced by positive electrospray ionization (ESI, +6 kV) using the spray array were accelerated by +6 kV applied on the Al-plate to impact the *R1b* ions (shown as the small gray balls marked with + sign), which were created on the solution-air interface of acetone solution containing acetic acid (10 ppm) and equivalent sodium salts of fluorine, chlorine (10 ppm) by using the coated Si-plate with +2 kV immersed in the acetone solution (shown as the blue green inside the vessel). **b** Diagram of potential variation of each electrode over time. From the starting point (t_0), the voltages on the spray array (+6 kV), Al-plate (+6 kV) and the ground electrode (0 V) maintained constant for over the working time ($t_0 \sim t_4$). However, the voltage on the coated Si-plate (+2 kV for the time slot of $t_0 \sim t_1$, $t_2 \sim t_3$ and 0 V for the time slot of $t_1 \sim t_2$, $t_3 \sim t_4$) and the voltage on the Cu-plate (0 V for the time slot of $t_1 \sim t_2$, $t_3 \sim t_4$ and disconnected from the common grounding point for the time slot of $t_0 \sim t_1$, $t_2 \sim t_3$). **c** The abundant signals of CCE products recorded by electrospray ionization mass spectrometry (ESI-MS). **d** The isotope-labeled CCE products detected by ESI-MS. RA denotes Relative Abundance.

Scheme 1 | Summary of C–C coupling reactions involving two cations *R1a*, *R1b* for CCE. **a** The schematic illustration on the initial stage ($d_0 \gg 2d_1$) of two positively charged acetone cations, *R1a*, *R1b*, approaching each other. **b** The repelling and coupling stage ($d_2 < d_0 < d_1$) of two positively charged acetone cations, *R1a*, *R1b*, approaching to each other. **c** The investigated C–C coupling reactions between two positively charged molecules, *R1a*, *R1b*. Note: d_0 is the

distance between the two positive charges located on the carbonyl carbon atoms of *R1a* and *R1b*. d_1 is the effective radius of the reaction cross-section between *R1a* and *R1b*. d_2 is the averaged length of C–C bond.

4. "The formed CCE products underwent additional analysis using MS and other techniques." Offline? And be explicit please - what are the techniques.

⇒ We thank the reviewer for pointing out the question. The formed CCE products in the solution/interface reactor chemistry underwent additional analysis using MS, UV-Vis and NMR techniques, which were all in offline detection mode. The sentence of "The formed CCE products underwent additional analysis using MS and other techniques." has been changed into "The formed CCE products underwent additional analysis using offline MS, UV-Vis and NMR techniques." in the revised manuscript. (see Page 24, Lines 470-471)

Reviewer #3 (Remarks to the Author):

Comments to the Author

The new revised version of the paper is a substantial improvement on the first version with more clearly defined figures. The authors have conducted more experiments to answer questions raised by me and the other reviewers. I have a few more doubts/questions about the work but they are minor:

⇒ We thank all the reviewers for their excellent and patient reviewing from the prior round. We also thank the reviewer for pointing out the new suggestions again. We have done a detailed revision of the manuscript according to the suggestions.

1. The authors mention that the electric fields present at the air water interface is what is responsible for overcoming the 7.2 eV barrier. However, the value quoted from Zare and coworkers (10^9 V/m) is electric field present in neutral droplets. However, it is clear

from these experiments that unless both the electrosprays/ or the electrospray and surface is charged there is no CCE product observed. Therefore, I disagree with the authors claim that the voltage they are supplying to the electrospray plume and surface is only useful for generating sodiated cations and does not energetically play a role in formation of CCE products. Could the author calculate the amount of electric fields they are generating at the surface, and report that?

⇒ We thank the reviewer for pointing out the new suggestions. The reviewer is correct. The value quoted from Zare and coworkers (10^9 V/m) is electric field present in neutral droplets. However, the droplets in our work are in charged. The voltage supplied to the electrosrapy plume and surface is not only used to generate sodiated cations, but also energetically play a role in formation of CCE products. While an electric field has been implicated at the water interface, there has been no direct measurement in aqueous microdroplets, largely due to the lack of proper measurements tools. Zare and coworkers employed the newly developed stimulated Raman excited fluorescence microscopy to measure the electric field at the water-oil interface of microdroplets, in which the droplets are fixed (*J. Phys. Chem. Lett.* 2020, 11, 17, 7423–7428). As determined by vibrational Stark effect of a nitril-bearing fluorescent probe, the strength of the electric field is found to be on the order of 10^9 V/m. However, in our work, the droplets are charged, which can be attracted by an electric field, and is not easily fixed, so it is difficult to measure the electric field value by using the reference method reported by Zare and coworkers.

⇒ According to the research literature, both the theoretical calculation and the experimental measurement of the electric field at the neutral droplet interface are of the order of 10^9 V/m (*J. Phys. Chem. Lett.* 2020, 11, 17, 7423–7428; *J. Phys. Chem. B* 2019, 123 (15),3348-3358; *J. Phys. Chem. B* 2015, 119 (29), 9114-9122; *J. Phys. Chem. Lett.* 2010, 1 (2), 496-499). For example, Cendagorta and Ichiye reported the formation of water-vapor interface potential as high as 0.4

V with the corresponding electric field at 0.8×10^9 V/m, which originates from the aligned orientation of water dipoles at the interface (J. Phys. Chem. B 2015, 119 (29), 9114-9122). Leung used density functional theory to estimate the water surface potential and the interfacial electric field to be approximately +3.63 V and 10^9 V/m (J. Phys. Chem. Lett. 2010, 1 (2), 496-499). According to the reference results, the water surface potential is related to the interfacial electric field. In our work, due to the application of a voltage of several kilovolts, a large amount of charge was accumulated on the surface of the water droplets, thus the interfacial electric field could be greater than 10^9 V/m. This high electric field plays a key role in the formation of CCE products energetically. At the present stage, we could not calculate the electric field value of the surface of the charged droplets by theoretical or experimental methods. However, it is certain that this value is greater than the electric field value on the surface of the neutral droplets. In our future work plan, we will try to calculate the electric field value at the air water interface in experiments. The detailed study and accurate measurement of the electric field in droplet and its effect on CCE reaction between two cations are beyond the main research scope of this paper.

⇒ In a response, the content of “The electric field intensity can be attained at approximately 10^9 V/m on gas-liquid interface of the cations reaction region ²⁴.” has been changed into “More specifically, an Al-plate applied with a high voltage (+1kV ~ +6 kV) for acceleration of *R1a*, a coated Si-plate applied with a high voltage (+1V ~ +2 kV) for creation of positively charged ions of acetone (e.g., *R1b*) on the liquid-air interface, a Cu-plate for elimination of extra charges which might be accumulated up to 10^9 V m⁻¹ ²⁴ during the reaction duty cycles, see Supplementary Note 1, were used to establish the precisely programmed electric field (Fig. 1b, further described in the Methods section) for CCE reactions (Scheme 1) between the ions of *R1a*, and *R1b*.” in the revised manuscript. (see Page 7, Lines 116-122)

⇒ **The content of** “The value quoted from Zare and coworkers (10^9 V m^{-1}) is electric field present in neutral droplets. However, the droplets in our work are in charged. The voltage supplied to the electrography plume and surface is not only used to generate sodiated cations, but also energetically play a role in formation of CCE products. While an electric field has been implicated at the water interface, there has been no direct measurement in aqueous microdroplets, largely due to the lack of proper measurements tools. Zare and coworkers employed the newly developed stimulated Raman excited fluorescence microscopy to measure the electric field at the water-oil interface of microdroplets, in which the droplets are fixed ¹. As determined by vibrational Stark effect of a nitril-bearing fluorescent probe, the strength of the electric field is found to be on the order of 10^9 V m^{-1} . However, in our work, the droplets are charged, which can be attracted by an electric field, and is not easily fixed, so it is difficult to measure the electric field value by using the reference method reported by Zare and coworkers. According to the research literature, both the theoretical calculation and the experimental measurement of the electric field at the neutral droplet interface are of the order of 10^9 V m^{-1} .¹⁻⁴ For example, Cendagorta and Ichiye reported the formation of water-vapor interface potential as high as 0.4 V with the corresponding electric field at $0.8 \times 10^9 \text{ V m}^{-1}$, which originates from the aligned orientation of water dipoles at the interface.³ Leung used density functional theory to estimate the water surface potential and the interfacial electric field to be approximately +3.63 V and 10^9 V m^{-1} .⁴ According to the reference results, the water surface potential is related to the interfacial electric field. In our work, due to the application of a voltage of several kilovolts, a large amount of charge was accumulated on the surface of the water droplets, thus the interfacial electric field could be greater than 10^9 V m^{-1} . This high electric field plays a key role in the formation of CCE products energetically. At the present stage, we could not calculate the electric field value of the surface of the charged droplets by theoretical or experimental methods. However, it is certain that this value is greater than the electric field value on the surface of the neutral droplets.

In our future work plan, we will try to calculate the electric field value at the air water interface in experiments.” **has been added in the revised Supplementary Information (see Supplementary Note 1 in Pages 4-5).**

2. The UV-Vis spectra shown by the authors is misleading, as it does not go below 400 nm, and what we see is essentially a tail of the spectra. A full spectra from 200-800 nm should be presented.

⇒ **We thank the reviewer for pointing out this question. A full UV-Vis spectra (Supplementary Fig. 7) from 200-800 nm has been provided in the revised Supplementary Material. Please see the Figure below.**

Supplementary Fig. 7. Characterization of CCE product by UV-Vis spectroscopy. Control: KMnO_4 ; Standard: 2-methylbutane-2,3-diol.

3. My initial question about hetero vs homo dicationic CCE product was to assert whether both the surface and the droplet have to have the sodiated acetone cation R1 for the reaction, or whether having the reactant in only one medium is sufficient. However, the authors mixed up different cations on both the surface and the plume, so I don't know what the heterodication product being in excess shows in their experiments. I would suggest that if they were to add only sodiated acetone in the plume and sodiated pentanone in the surface and then check what reactions they observed that will help understand the role of the surface beyond being charged.

⇒ **We apologized for misinterpreting your intentions earlier. We thank you for**

pointing out this question in detail again. We have performed the experiments by using sodiated acetone in the spray and sodiated pentanone in the interface as suggested, and found that only the products of hetero-dicationic reaction can be observed. The mass spectrum shows hetero-dicationic reaction products at m/z 177, 199, while no homo-dicationic CCE products were observed (Supplementary Fig. 11 see below). The online two-spray experiments (Supplementary Fig. 10 see below) also indicate that both the two sprays have to have the sodiated acetone cation R1 for the CCE reaction, and no CCE product was observed when only one spray has the sodiated acetone cation R1. In a response, the content of “It was also observed that in case where two different cations were present in both the spray and the reaction vessel, both homo-dicationic and hetero-dicationic reaction products could be observed (Supplementary Fig. 11). Interestingly, the intensity of the hetero-dicationic reaction products appeared to be higher than that of the homo-dicationic reaction products. The issue may stem from a probability problem.” has been changed into “It was also observed that in case where two different cations were present in the spray and the reaction vessel, the hetero-dicationic reaction products could be observed (Supplementary Fig. 11).” in the revised manuscript. (see Pages 10, Lines 189-191)

Supplementary Figure 11. Mass spectrum of collision reaction when sodiated acetone and sodiated pentanone were introduced to the spray and the interface, respectively.

Supplementary Figure 10. Control experiments for CCE products formation by online MS. a The MS spectrum of CCE reaction of two sodiated acetone cations when the spray 1 charged while the spray 2 not charged. **b** The MS spectrum of CCE reaction of two sodiated acetone cations when the spray 1 and spray 2 both charged. **c** The MS/MS spectrum of CCE product at m/z 147.

Reviewer #4 (Remarks to the Author):

Comments to the Author

1. The authors assert that Route 1 involves CCE reaction through the elimination of a methyl cation with a concerted mechanism. Nonetheless, the transition state (TS1) as deduced from calculations does not substantiate this claim. All computational findings indicate that TS1 aligns with the dissociation of a methyl cation, with the formation of

a C-C bond evident in TS1, characterized by a bond length of 1.58 Å. This contrasts the typical C-C bond length range of 1.8 Å to 2.2 Å observed in transition states for C-C bond formation. The authors' contention that "The TS1 directly optimized by the DFT method gives only one imaginary frequency of about -261.64 cm⁻¹, and 59 real frequencies" merely confirms the transitional nature of TS1 but falls short of validating it as the anticipated concerted transition state. Moreover, the authors' rebuttal stating, "We also checked the output eigenvector of this mode with imaginary mode, as shown in the imaginary frequency mode section of SI (see Response 4 above), which is the correct direction associated with one C-C bond stretching and the other C-C bond contracting," appears inadequate. The imaginary frequency mode distinctly indicates its association with the stretching of one C-C bond, facilitating the elimination of a methyl cation, yet it does not exhibit any vibrational mode indicative of the formation of the anticipated C-C bond. The authors are urged to unambiguously demonstrate the precursor intermediate in the IRC calculation for the TS1 structure and endeavor to identify the transition state for C-C bond formation.

⇒ **We thank the reviewer for bringing this to our attention and have updated the text as suggested. The reviewer is correct. The current calculation results cannot confirm the concerted mechanism. According to the suggestion, we put all our efforts to find other possible transition state structure during the formation of C-C bond. Fortunately, we found a transition state structure (TS1) for the process of C-C bond formation and an intermediate C1 with a very shallow potential well (see Fig. 3a below). Our IRC calculations connecting TS1 and TS2 demonstrate the direct reaction associated with the forming and breaking processes of C-C bonds (see Fig. S22a below). Compared with Route 2, a two-step mechanism (cation dissociation, then ion-molecule reaction) in Scheme 3 (see below), Route 1 can be viewed as a direct mechanism, that is, two cations directly collide and react to produced CCE product. In response, the term "concerted mechanism" has been changed to "direct mechanism" in the revised manuscript, in contrast to "two-step**

mechanism". The content of "Our IRC calculations of *TS1* and *TS2* demonstrate that the correct direction is associated with one C–C bond stretching and another C–C bond contracting (Supplementary Fig. 22a)." has been added in the revised manuscript. (see Pages 15-16, Lines 287-289)

Fig. 3 | Results of stationary point and AIMD calculations that support the CCE process. a Relative energy levels for Route 1 for the generation of *P4a* ion via *TS1* and *TS2* through the demethylation C–C coupling reaction. **b** Snapshots of one reactive trajectory via *TS1* and *TS2*. (Note: Each panel depicts the geometry at the specified reaction time (fs). The internuclear distances (Å) between the two C atoms involved in bond formation, and between the two C atoms in detaching CH₃⁺ and *P4a* are indicated by red and blue fonts, respectively.)

Supplementary Figure 22. Intrinsic reaction coordinate (IRC) calculations of Route 1 via *TS1*, *TS2* and Route 1 via *TS3*, *C1*, *TS4*, *C2* for the formation of product ion *P4* (two configurations including *P4a*, *P4b*) *m/z* 147 at the B3LYP-gcp-D3/6-31+G(d) level. **a** IRC calculations of *TS1* and *TS2* for the formation of product ion *P4a* at *m/z* 147. **b** IRC calculations of *TS3* and *TS4* for the formation of product ion *P4b* at *m/z* 147.

Scheme 3 | Gas-phase production of sodium 2-methyl-butane-2,3-diolate)-3-ium

(*P4*, m/z 147) by collision activation of sodiated acetone (*RI*, m/z 81). **a** Route 1 for demethylation C–C coupling reaction through a direct mechanism. **b** Route 2 for ion/molecule reaction by a two-step mechanism.

⇒ Thanks to the reviewer for this valuable comment. Since the small difference in C–C bond length between the C–C bond formed in the transition state and the corresponding C–C bond of the *P4* product, the vibrational pattern of C–C bond formation in the transition state is not obvious. The content of “The imaginary vibrational frequency of *TS1*, along with the corresponding eigenvector of this vibrational mode, indicated the correct direction associated with the stretching of one C–C bond and the contraction of the other C–C bond.” has been changed into “The imaginary vibrational frequency of *TS1* and *TS2*, along with the corresponding eigenvector of these vibrational modes, indicated the correct direction associated with the stretching of one C–C bond and the contraction of the other C–C bond, respectively.” in the revised manuscript. (see Page 15, Lines 284-286)

2. Regarding the AIMD calculations, the study's reliance on a singular reactive trajectory is both insufficient and unconvincing.

⇒ We thank the reviewer for the comment. Extensive AIMD calculations were carried out, and a total of about 100 reactive trajectories via the direct mechanism were obtained. Actually, more reactive trajectories can be obtained using more computational time and capacity. Just one typical trajectory is shown in the paper to explain the reaction process. In a response, the content of “Fig. 3b displays snapshots of a single trajectory obtained through *TS1*.” has been changed into “Fig. 3b displays snapshots of a typical trajectory obtained through *TS2*.” in the revised manuscript (see Page 17, Line 322) The content of “Many reactive trajectories via the direct mechanism were obtained, while one typical trajectory is shown to explain the reaction process, as depicted in the Supplementary video animation.” has been added in the revised manuscript. (see Pages 29, Lines 593-595)

3. The insights provided by Reviewer 2 are particularly pertinent. Despite the authors' extensive response letter, they have not effectively tackled the raised concerns. Furthermore, the manuscript's complexity makes it challenging for readers to grasp, necessitating substantial revisions for clarity and coherence.

⇒ We thank the Reviewer 2 for the excellent and patient reviewing. We also thank the Reviewer 4 for the insightful and positive suggestions. All of these suggestions greatly improved the quality of the manuscript. The main concern for the Reviewer 2 is that it is very hard to believe that there is only one transition state (TS) in such complex carbon-carbon elongation reaction and the C-C bond is already formed in the TS1. According to the suggestion, we put all our efforts to find other possible TS structure. Fortunately, we have found another transition state structure for C-C coupling with multiple transition states and intermediates (see Fig. S20 below). In addition, we have also found a transition state structure (TS1) for the process of C-C bond formation and intermediate C1 with a very shallow potential well (see Fig. 3a above). This is consistent with the reviewer's suggestions.

Supplementary Figure 20. The stationary points and energies on the minimum energy path for additional pathway that support the CCE process. Relative energy levels to generate *P4b* ion via *TS3* by the demethylation C–C coupling reaction. Purple Na, red O, gray C, and white H. *R1*: *m/z* 81, *P4b*: *m/z* 147.

⇒ The manuscript included abundant experimental data and theoretical

calculation results, in which the experimental results consist of two sets of data, the solution/interface reactor chemistry and the gas-phase ion-ion chemistry. We have now re-edited the manuscript to keep the content more clarity, coherence, understandable and general according to the suggestions (see highlights in the revised manuscript). For example, the confused Fig. 1 has been redrawn (see Fig. 1 below), data in liquid phase section and in gas phase section were presented more clearly, the calculation results involving in the transition state structure (TS1) for the process of C-C bond formation and intermediate C1 with a very small potential well has been added and explained (see Fig. 3 above and Fig. S22 above), the content of calculation part has been modified accordingly.

⇒ The content of “In this work, we present a novel strategy using Coulombic repulsion between two positive charges to induce CCE reactions (Scheme 1) that can occur between two positively charged organic species (*i.e.*, two sodiated acetone, ***R1a***, ***R1b***) in either the liquid or gas phases. Initially, the distance (d_0) between the positive charges sitting on the carbonyl carbon atoms of the cations ***R1a***, ***R1b*** was far larger than d_1 (Scheme 1a), the repel effect induced by repulsive Coulomb force (RCF) between the ***R1a***, ***R1b*** cations was too small (~ 0 eV) to trigger effective interaction between ***R1a*** and ***R1b***. When d_0 gradually decreased, the RCF between ***R1a*** and ***R1b*** continuously increased and became a dominant force. When $d_0 = d_1$, the ***R1a*** and ***R1b*** cations began to contact on the edge of the reaction cross-section. Due to inertia, ***R1a*** ion still moved towards ***R1b***, making d_0 continue to decrease. Once $d_0 < d_1$ (Scheme 1b), the repulsive energy generated by RCF might be significantly larger than the C–C bond energy, which induced the elimination of a small cleavage cation (typically methyl, amino, phenyl cation) and the coupling of the two carbonyl carbon atoms, resulting in positively charged CCE products. After electric neutralization, the CCE products were accumulated for further characterization. Thus, the RCF between two positively charged (*i.e.*, protonated or metal ion complexed) organic species ***R1a*** and ***R1b*** functioned as a

facile type of “*virtual catalyst*” to facilitate CCE and functional group construction (Scheme 1c).” **has been modified in the revised manuscript. (see Pages 4-5, Lines 73-90)**

⇒ **The content of** “A custom-designed reaction apparatus (Fig. 1a) was comprised with a capillary array for electrospraying a 1:1 (v/v) sodiated acetone solution to produce reagent ions (*e.g.*, *RIa*), a reaction vessel assembly and several electrodes to construct a precisely programmed electric field. More specifically, an Al-plate applied with a high voltage (+1kV ~ +6 kV) for acceleration of *RIa*, a coated Si-plate applied with a high voltage (+1V ~ +2 kV) for creation of positively charged ions of acetone (*e.g.*, *RIb*) on the liquid-air interface, a Cu-plate for elimination of extra charges which might be accumulated up to 10^9 V m^{-1} ²⁴ during the reaction duty cycles, see Supplementary Note 1, were used to establish the precisely programmed electric field (Fig. 1b, further described in the Methods section) for CCE reactions (Scheme 1) between the ions of *RIa*, and *RIb*.” **has been modified in the revised manuscript. (see Page 7, Lines 113-122)**

⇒ **The content of** “It is important to note that the primary products of CCE reactions between the ions of *RIa*, and *RIb* (Scheme 1) consist of charged species, specifically the methyl cation (*i.e.*, CH_3^+) and the organic cation with elongated carbon chain (*i.e.*, the sodium 2-methyl-butane-2,3-diolate-3-ium). Under the experimental conditions, the cations were electronically neutralized in two possible ways (Scheme 2) to form stable final products: pathway 1) by direct grounding to yield a neutral radical, which grabs a hydrogen radical from the solvent surrounded to produce stable *MH*; and pathway 2) by cations coupling with encountered anions to produce stable *MX* (*i.e.*, *M* = methyl, $\text{R}_2\text{C}(\text{OH}/\text{Na})\text{-C}(\text{OH}/\text{Na})\text{R}_4\text{R}_3$; *X* = F, Cl, Br, CH_3COO). For example, by subjecting to ESI-MS, the final products of the CCE reaction produced signals at *m/z* 149, *m/z* 167 and *m/z* 183 (Fig. 1c) corresponded to $(\text{MH}+\text{H})^+$, $(\text{MF}+\text{H})^+$, and $(\text{MCl}+\text{H})^+$, respectively. These ion signals were further confirmed by isotope labeling experiments using deuterated acetone, yielding dominant signals at *m/z* 158, *m/z* 176 and *m/z* 192 (Fig. 1d)

corresponded to $(MH-d_9+H)^+$, $(MF-d_9+H)^+$, $(MCl-d_9+H)^+$, respectively. This correspondence was attributed to the addition of H radical (pathway 1 in Scheme 2), F^- , and Cl^- anions (pathway 2 in Scheme 2) in the solution within the reaction vessel. The ESI-MS (Fig. 1c, 1d) and collision-induced dissociation mass spectrometry (CID-MS) (Supplementary Fig. 1a, 1b) results indicated the generation of products related to demethylation and C - C coupling during the CCE reaction between the ions of *R1a*, and *R1b*.” **has been modified in the revised manuscript. (see Pages 7-8, Lines 128-146)**

⇒ **The content of** “As depicted in Fig. 1a, the homemade experimental setup comprised four primary components: (1) capillary array for production of *R1a* ions by electrospray ionization, (2) electrodes including Al-plate, coated Si-plate, Cu-plate for electric field construction, (3) reaction vessel for CCE reaction in liquid-phase, and (4) timing and programmed electric field control system. As shown in Fig. 1a, *R1a* ions were produced by the spray array while the *R1b* ions were created on the liquid-air interface by the high voltage applied to the coated Si-plate covered by the solution inside the reaction vessel. Thus, the solution interface was the major region for liquid-phase CCE reaction. However, the ionic products of liquid phase CCE reaction can subsequently capture H radicals or other anions in the solution, resulting in the formation of stable final products of the CCE reaction. Taking sodiated acetone as an example, the capillary electrospray array was comprised by inserting one end of all the capillaries (inner diameter (ID) 0.19 mm) into a solution of sodiated acetone, which was then biased with high voltage (+1-6 kV) to generate charged droplets containing *R1a* ions (Fig. 1a). All the spray tips were neatly stacked together, with a distance of 5 mm from the liquid surface. In order to create cations *R1b*, a coated Si-plate biased with +1-2 kV was placed under the solution inside the reaction vessel. A Cu-plate was immersed in the solution and linked to Ground to dissipate the static electricity accumulated during the CCE process. A timing-control system was developed to regulate the high voltage settings for various electrodes such as Al-plate, spray array, coated Si-plate, and Cu-plate (Fig.

1a). In this study, the voltage values were equally set and maintained constant for the spray array and Al-plate electrodes, whereas the voltage in the coated Si-plate exhibited a pulsed mode.” has been modified in the revised manuscript. (see Pages 23-24, Lines 446-466)

Fig. 1 | The liquid-phase CCE reaction occurred between two organic cations *R1a*, *R1b*. **a** Schematic illustration of the experimental apparatus. While the whole apparatus was electronically grounded, the *R1a* ions (shown as the charged red-brown plume) produced by positive electrospray ionization (ESI, +6 kV) using the spray array were accelerated by +6 kV applied on the Al-plate to impact the *R1b* ions (shown as the small gray balls marked with + sign), which were created on the solution-air interface of acetone solution containing acetic acid (10 ppm) and equivalent sodium salts of fluorine, chlorine (10 ppm) by using the coated Si-plate with +2 kV immersed in the acetone solution (shown as the blue green inside the vessel). **b** Diagram of potential variation of each electrode over time. From the starting point (t_0), the voltages on the spray array (+6 kV), Al-plate (+6 kV) and the ground electrode (0 V) maintained constant for over the working time (t_0 ~ t_4). However, the voltage on the coated Si-plate (+2 kV for the time slot of t_0 ~ t_1 , t_2 ~ t_3 and 0 V for the time slot of t_1 ~ t_2 , t_3 ~ t_4) and the voltage on the Cu-plate (0 V for the time slot of t_1 ~ t_2 , t_3 ~ t_4 and disconnected from the common grounding point for the time slot of t_0 ~ t_1 , t_2 ~ t_3). **c** The abundant signals of CCE products recorded by electrospray ionization mass spectrometry (ESI-

MS). **d** The isotope-labeled CCE products detected by ESI-MS. RA denotes Relative Abundance.

Reviewers' Comments:

Reviewer #3:

Remarks to the Author:

[Note from the Editor: Reviewer #3 was asked to assess the response given to reviewer #1 who was not able to look over the revision again].

The paper is substantially improved with this new version. However, some more questions/suggestions remain to help me understand the impact of this work:

1. The authors have made the difference between the gas phase and solution phase experiments quite clear in the new version, which is great. In the supplementary figure 10, the authors discuss conducting these same reactions using two electrosprays instead of the spray-planar interface cell. To me, it seems that the two charged spray mass spec data shows higher amounts of adduct product formed and should be a substantially easier setup for anyone to build. What advantages does the multielectrode cell built by authors have over the two electrospray experiments? Authors should discuss the two electrospray experiments in the main paper instead of burying it into the supplementary text.
2. While the authors have put forward some convincing arguments about how it is difficult to gauge the exact value of electric field on moving charged droplets. Calling this experimental setup having precisely programmed electric fields and not being able to provide a value for electric field seems contradictory.
3. While the new Figure 1 is now clearly drawn, I am still lacking an explanation for why is pulsed voltage used for certain electrodes while others are kept at constant voltage?
4. What is the control KMnO₄ spectra showing in the UV-Vis spectra in supplementary information?

Minor:

1. On page 8 the authors use different designations of X, once as neutral species F, Cl, Br and others and another as anionic species F⁻, Cl⁻, Br⁻. The authors need to be consistent with their definition.
2. In scheme 2 the authors end up with neutral RX products but cationic RH products. They should provide an additional pathway for what neutralizes this RH⁺ species.
3. The yield calculations are not easy to understand. Please expand.

Reviewer #4:

Remarks to the Author:

Recommendation: Publish after polishing the manuscript

Dear Reviewers,

Thank you for your decision and constructive comments on our manuscript. We have carefully considered your suggestion and make some changes. We hope that you now find this manuscript acceptable for publication in *Nature Communications*.

The yellow part that has been revised according to your comments. Revision notes, point-to-point, are given as follows:

REVIEWER COMMENTS

Reviewer #3 (Remarks to the Author):

The paper is substantially improved with this new version. However, some more questions/suggestions remain to help me understand the impact of this work:

1. The authors have made the difference between the gas phase and solution phase experiments quite clear in the new version, which is great. In the supplementary figure 10, the authors discuss conducting these same reactions using two electrosprays instead of the spray-planar interface cell. To me, it seems that the two charged spray mass spec data shows higher amounts of adduct product formed and should be a substantially easier setup for anyone to build. What advantages does the multielectrode cell built by authors have over the two electrospray experiments? Authors should discuss the two electrospray experiments in the main paper instead of burying it into the supplementary text.

⇒ We feel great thanks for your professional review work on our manuscript.

The two-channel electrospray data shows more reactant ions [acetone + Na]⁺ adduct than the spray-planar interface cell. It is because the reaction solution in the spray-planar interface cell was diluted by a factor of 100 before injecting into MS detection. The content “The quantity of acetone reactant utilized was 24 mL, and the reaction took place over a duration of 30 min using the apparatus depicted in Fig. 1a. Subsequently, the reaction solution was diluted by a factor of 100 and subjected to ESI-MS analysis.” was described in the original Method section-Yield calculation, which is now moved to the Method section-Experimental setup and process. (see Page 26, Lines 496-499).

⇒ **Two-channel electrospray is indeed an easy device to implement. However, 1) the cation-cation collision in the two-channel electrospray is not a forward collision, but an oblique collision at a certain angle. Due to the dispersion of three-dimensional space and the influence of atomizing gas flow, the collision probability is low, and the corresponding yield of CCE product ions is also low (~ 0.01% for sodiated/protonated acetone). The use of spray array device can better improve the yield of CCE product ions (~ 14% for sodiated/protonated acetone). 2) For the two-channel electrospray device, the lack of product collection device makes it difficult to collect the forming CCE product for subsequent analysis. While the use of spray array device can facilitate the collection of products for subsequent characterization analysis. The design of a spray array-planar interface cell aimed to promote the development of mass spectrometry preparation towards the bulk-scale synthesis. 3) Due to the low yield of the product, the difficulty of product collection, and the waste of the substrates, resulting in the low utilization rate of the substrates. Therefore, we designed a spray array-planar interface cell device to study the CCE reaction between two cations.**

⇒ **According to the suggestion, Fig. S10 was moved to Fig. 1 in the main text (See Fig. 1 below). The discussion on the advantages of multielectrode cell over the two-channel electrospray device has been added in the revised manuscript. It reads “Similarly, ionic CCE products at m/z 147 can also be observed with a low intensity by using online MS with two-channel charged sprays (spray1, +3 kV, spray2, +2 kV) (Fig. 1e). The characterization of m/z 147 was confirmed by tandem MS experiment (Supplementary Fig. 10). No ionic CCE product ion was observed by using online MS with two-channel sprays, when only one-channel spray was charged (spray1, +3 kV, spray2, 0 kV) (Fig. 1f). These experiment results were similar with those obtained using the apparatus in Fig. 1a (Supplementary Fig. 9). Two-channel electrospray is in effect an easy apparatus to implement. However, on one hand, the cation-cation collision in the two-channel electrospray is not a**

forward collision, but an oblique collision at a certain angle. Due to the dispersion of three-dimensional space and the influence of atomizing gas flow, the forward collision probability is low, and the corresponding yield of CCE product ions is also low (~ 0.01% for sodiated/protonated acetone). The use of spray array apparatus can better improve the yield of CCE product ions (~ 14% for sodiated/protonated acetone). On the other hand, for the two-channel electrospray apparatus, the lack of product collection apparatus makes it difficult to collect the forming CCE product for subsequent analysis. While the use of spray array apparatus can facilitate the collection of products for subsequent characterization analysis. The design of a spray array-planar interface cell aims to promote the development of mass spectrometry preparation towards the bulk-scale synthesis. In addition, the low yield of the product, the difficulty of product collection, and the waste of the substrates result in the low utilization rate of the substrates. Therefore, a spray array-planar interface cell apparatus was designed to study the CCE reaction between two cations.”. (see Page 10-11, Lines 188-207)

Fig. 1 | The liquid-phase CCE reaction occurred between two organic cations *R1a*, *R1b*.
a Schematic illustration of the experimental apparatus. **b** Diagram of potential variation of each electrode over time. **c** The abundant signals of CCE products recorded by electrospray ionization mass spectrometry (ESI-MS). **d** The isotope-labeled CCE products detected by ESI-MS. **e** The online-MS spectrum of CCE reaction between two sodiated acetone cations using a two-channel electrospray apparatus when spray 1 and spray 2 were both charged. **f** The online-MS spectrum of CCE reaction between two sodiated acetone cations using a two-channel electrospray apparatus when spray 1 was charged while the spray 2 was not charged. RA denotes Relative Abundance.

2. While the authors have put forward some convincing arguments about how it is difficult to gauge the exact value of electric field on moving charged droplets. Calling this experimental setup having precisely programmed electric fields and not being able

to provide a value for electric field seems contradictory.

⇒ **We thank the reviewer for the suggestion. We agree with you that “Calling this experimental setup having precisely programmed electric fields and not being able to provide a value for electric field seems contradictory”. We originally wanted to emphasize precisely sequence control of the operating voltages of the different electrodes. We apologized for the incorrect language description, and thank you very much for helping us improve the quality of our draft. According to the suggestion, we modified throughout the text, the corresponding content of “precisely programmed electric fields” has been changed into “precisely sequence-controlled voltage system” in the revised manuscript. (see Page 5, Lines 93-94; Page 23, Lines 440-441; Page 24, Line 446; Page 25, Line 494)**

3. While the new Figure 1 is now clearly drawn, I am still lacking an explanation for why is pulsed voltage used for certain electrodes while others are kept at constant voltage?

⇒ **Thank you again for your positive comments to improve the quality of our manuscript. When the cations formed by the spray array collides with the cations formed at the solution surface of the Si-plate, a large amount of static electricity is generated due to the accumulation of charge, which needs to be released or eliminated. The static electricity generated during the reaction is released to the ground by adding a Cu-plate electrode to the solution. The Si-plate and Cu-plate are in the same solution, and their purpose is to form cations and release static electricity, respectively, resulting in their inability to work at the same time. Therefore, Si-plate and Cu-plate work alternatively in a pulse mode, while the spray array, Al-plate, and Ground maintain a constant voltage.**

⇒ **To make the description in Figure 1 (see above) clearer, the content of “When the cations formed by the spray array collides with the cations formed at the**

solution surface of the Si-plate, a large amount of static electricity is generated due to the accumulation of charge, which needs to be released or eliminated. The static electricity generated during the reaction is released to the ground by adding a Cu-plate electrode to the solution. The Si-plate and Cu-plate are in the same solution, and their purpose is to form cations and release static electricity, respectively, resulting in their inability to work at the same time. Therefore, Si-plate and Cu-plate work alternatively in a pulse mode, while the spray array, Al-plate, and Ground maintain a constant voltage.” **has been added in the revised manuscript. (see Page 25, Lines 484-491)**

4. What is the control KMnO₄ spectra showing in the UV-Vis spectra in supplementary information?

⇒ **Thanks for the insightful question. The control KMnO₄ spectra was the blue-line spectrum in Supplementary Figure 7 (see below), which show the characteristic spectra in the range of 500-600 nm and a maximum absorption characteristic peak at 525 nm. The content of “The CCE product formed by two protonated acetone cations and the 2-methylbutane-2,3-diol standard exhibited similar UV-Vis spectra, both of which resulted in the disappearance of the characteristic peak of KMnO₄ (Supplementary Fig. 7).” has been changed into “2-Methylbutane-2,3-diol standard and the CCE product formed by two protonated acetone cations exhibited similar UV-Vis spectra. Both of them resulted in the decrease or even disappearance of the characteristic absorption peak of KMnO₄. The characteristic spectra of KMnO₄ were in the range of 500 ~ 600 nm, and the maximum absorption peak is 525 nm (Supplementary Fig. 7).” in the revised manuscript. (see Page 10, Lines 171-175)**

Supplementary Figure 7. Characterization of CCE product by UV-Vis spectroscopy. Control: KMnO_4 ; Standard: 2-methylbutane-2,3-diol.

Minor:

1. On page 8 the authors use different designations of X, once as neutral species F, CL, Br and others and another as anionic species F⁻, Cl⁻, Br⁻. The authors need to be consistent with their definition.

⇒ We were really sorry for our careless mistakes. Thank you for your reminder.

The designations of X have been changed to be consistent in all texts. The origin content of “which result from the formation of ($\text{CH}_3\text{X}^+\text{Ac}^-$, $\text{X} = \text{F}^-$, Cl^- , Br^- and CH_3COO^-) adducts.” has been changed into “which resulted from the formation of [$\text{CH}_3\text{X}^+\text{Ac}^-$] ($\text{X} = \text{F}$, Cl , Br and CH_3COO) adducts” in the revised manuscript. (see Page 9, Lines 149-150)

2. In scheme 2 the authors end up with neutral RX products but cationic RH products. They should provide an additional pathway for what neutralizes this RH^+ species.

⇒ Thank you for pointing this out. The additional pathway for neutralizing RH^+ species has been added in the revised Scheme 2 (see below) according to the suggestion. (see Page 8, Lines 123-125) In addition, the origin content of “Under the experimental conditions, the cations were electronically neutralized in

two possible ways (Scheme 2) to form stable final products: pathway 1) by direct grounding to yield a neutral radical, which grabs a hydrogen radical from the solvent surrounded to produce stable MH ; and pathway 2) by cations coupling with encountered anions to produce stable MX (*i.e.*, M = methyl, $R_2C(OH/Na)-C(OH/Na)R_4R_3$; X = F, Cl, Br, CH_3COO).” **has been modified as** “Under the experimental conditions, the cations were electronically neutralized in two possible pathways (Scheme 2) to form stable final products. The pathway 1 was conducted by capturing a hydrogen radical from the solvent to yield a radical cation species, which was further directly grounded and neutralized to produce stable MH and R_1H (*i.e.*, M = $R_2C(OH/Na)-C(OH/Na)R_4R_3$; R_1 = methyl). The pathway 2 was conducted by cations coupling with encountered anions to produce stable MX and R_1X (*i.e.*, M = $R_2C(OH/Na)-C(OH/Na)R_4R_3$; R_1 = methyl; X = F, Cl, Br, CH_3COO).” **in the revised manuscript. (see Page 8, Lines 129-135)**

Scheme 2 | Direct grounding and anion coupling in solvent to obtain the final product of CCE reactions as depicted in Scheme 1. $M=R_2C(OH/Na)-C(OH/Na)R_4R_3$

3. The yield calculations are not easy to understand. Please expand.

⇒ **Thank you for your nice comments on our manuscript. According to your suggestions, we have supplemented the content of the yield calculation in detail in the revised text. Due to space constraints, detailed yield calculations were moved to the Supplementary Information. It reads “The method for yield calculation in this work was derived from the rough estimates of the yields of the microdroplet synthesis reactions reported by Cooks^{25,44} group and Zare^{17,26} group,**

which was obtained by measuring the conversion ratio: the ratio of the intensity between the product (P) and the sum of the intensities of the reactant (R), and product (P), e.g., $[P]/([R]+[P])$. In this work, the yields were determined as the ratio of the sum of the intensity of C–C coupling product ions to the sum of the intensity of reactant ions and the intensity of C–C coupling product ions.

For the gas-phase reaction in the ion trap, taking CCE reaction between two sodiated acetone cations as an example, the detailed calculation process is outlined as follows: the yield of CCE reaction for sodiated acetone was determined by calculating the ratio of the intensity of CCE product ions at m/z 147 [$\text{CH}_3\text{C}^+(\text{ONa})\text{-C}(\text{ONa})(\text{CH}_3)_2$] to the sum of the intensity of reactant ions at m/z 81 [$\text{CH}_3\text{C}^+(\text{ONa})\text{CH}_3$] and C–C coupling product ions at m/z 147 (Fig. 2a). The signal intensity of m/z 81 and m/z 147 were 8.5 and 2.5, respectively. Consequently, the yield for CCE reaction between two sodiated acetone cations in the gas-phase was calculated as approximately 25%.

For the liquid phase reaction in the Fig. 1a, taking CCE reaction between protonated/sodiated acetone as an example, the detailed calculation process is outlined as follows: the yield of CCE reaction for protonated/sodiated acetone was determined by calculating the ratio of the sum of the intensity of CCE product ions at m/z 103 [$\text{CH}_3\text{C}^+(\text{OH})\text{-C}(\text{OH})(\text{CH}_3)_2$], m/z 105 [$\text{CH}_3\text{CH}(\text{OH})\text{-C}(\text{OH})(\text{CH}_3)_2 + \text{H}^+$], m/z 147 [$\text{CH}_3\text{C}^+(\text{ONa})\text{-C}(\text{ONa})(\text{CH}_3)_2$], m/z 149 [$\text{CH}_3\text{CH}(\text{ONa})\text{-C}(\text{ONa})(\text{CH}_3)_2 + \text{H}^+$], m/z 163 [$\text{CH}_3(\text{CH}_3\text{COO})\text{C}(\text{OH})\text{-C}(\text{OH})(\text{CH}_3)_2 + \text{H}^+$], and m/z 207 [$\text{CH}_3(\text{CH}_3\text{COO})\text{C}(\text{ONa})\text{-C}(\text{ONa})(\text{CH}_3)_2 + \text{H}^+$] to the sum of the intensity of reactant ions at m/z 59 [$\text{CH}_3\text{C}^+(\text{OH})\text{CH}_3$], m/z 81 [$\text{CH}_3\text{C}^+(\text{ONa})\text{CH}_3$] and CCE product ions at m/z 103, m/z 105, m/z 147, m/z 149, m/z 163, and m/z 207 (Supplementary Fig. 5a). The signal intensity of m/z 59, m/z 81, m/z 103, m/z 105, m/z 147, m/z 149, m/z 163, and m/z 207 were 7.38×10^5 , 1.85×10^5 , 1.72×10^4 , 1.71×10^4 , 1.92×10^4 , 1.99×10^4 , 1.32×10^4 , 7.00×10^4 , respectively.

Consequently, CCE yield for protonated/sodiated acetone was calculated as approximately 14%.”. (see Pages 6-7 in the Supplementary Information)

Reviewer #4 (Remarks to the Author):

Recommendation: Publish after polishing the manuscript

⇒ We feel great thanks for your professional review work on our manuscript. We have now worked on both language and readability and have also polished the manuscript. We really hope that the language level has been substantially improved.

Reviewers' Comments:

Reviewer #3:

Remarks to the Author:

I am happy with the answers provided by the authors to my questions and implemented changes to the manuscript. I recommend publication.

REVIEWER COMMENTS

Reviewer #3 (Remarks to the Author):

I am happy with the answers provided by the authors to my questions and implemented changes to the manuscript. I recommend publication.

⇒ **We are happy to hear this good news from you and feel great thanks for your professional review work on our manuscript.**